*Nat Cell Biol.* Author manuscript; available in PMC 2023 January 24.

# LAP1 supports nuclear adaptability during constrained melanoma cell migration and invasion

Yaiza Jung-Garcia[1,2,3,4], Oscar Maiques[2,4], Joanne Monger[2], Irene Rodriguez-Hernandez[2,4], Bruce Fanshawe[4], Marie-Charlotte Domart[5], Matthew J Renshaw[6], Rosa M Marti[7], Xavier Matias-Guiu[8], Lucy M Collinson[5], Victoria Sanz-Moreno[2,4,*], Jeremy G Carlton[1,3,*]

[1]Organelle Dynamics Laboratory, The Francis Crick Institute, 1 Midland Rd, London NW1 1AT, United Kingdom

[2]Sanz-Moreno Group, Centre for the Tumour Microenvironment, Barts Cancer Institute, Queen Mary University of London, John Vane Science Building, Charterhouse Square, London EC1M 6BQ, United Kingdom

[3]Comprehensive Cancer Centre, School of Cancer and Pharmaceutical Sciences, King's College London, London, SE1 1UL, United Kingdom

[4]Randall Division of Cell and Molecular Biophysics, King's College London, London SE1 1UL, United Kingdom

[5]Electron Microscopy Science Technology Platform, The Francis Crick Institute, 1 Midland Road, London NW1 1AT, UK

[6]Advanced Light Microscopy Science Technology Platform, The Francis Crick Institute, 1 Midland Road, London NW1 1AT, UK

[7]Department of Dermatology, Hospital Universitari Arnau de Vilanova, University of Lleida, IRB Lleida, CIBERONC, 25198 Lleida, Spain

[8]Department of Pathology and Molecular Genetics, Hospital Universitari Arnau de Vilanova, University of Lleida, IRB Lleida, CIBERONC, 25198 Lleida, Spain

## Abstract

Metastasis involves dissemination of cancer cells away from a primary tumour and colonisation at distal sites. During this process, the mechanical properties of the nucleus must be tuned since

Correspondence to: Victoria Sanz-Moreno; Jeremy G Carlton.

Correspondance: jeremy.carlton@kcl.ac.uk, v.sanz-moreno@qmul.ac.uk.
*These authors contributed equally
*These authors jointly supervised this work

**Author Contributions**
J.C.G. and V.S.-M. were principal investigators, designed the research, supervised experiments and wrote the paper; Y.J.-G. designed the research, performed the experiments, analysed the data, wrote the paper; O.M. performed immunohistochemistry and digital pathology and supervised mouse work I.R.-H. assisted with the transcriptomic analyses and qPCR; J.M. and B.F. performed mouse work; M.-C.D. and L.M.C. performed and supervised the CLEM experiments; M.R. trained Y.J.-G. in FRAP and supervised the experiments; R.M.M. and X.M.-G provided the human tissue.

**Competing Interests**
The authors declare no competing interests.

they pose a challenge to the negotiation of physical constraints imposed by the microenvironment and tissue structure. We discovered increased expression of the inner nuclear membrane protein LAP1 in metastatic melanoma cells, at the invasive front of human primary melanoma tumours and in metastases. Human cells express two LAP1 isoforms (LAP1B and LAP1C), which differ in their amino terminus. Using *in vitro* and *in vivo* models that recapitulate human melanoma progression, we found that expression of the shorter isoform, LAP1C, supports nuclear envelope blebbing, constrained migration and invasion by allowing a weaker coupling between the nuclear envelope and the nuclear lamina. We propose that LAP1 renders the nucleus highly adaptable and contributes to melanoma aggressiveness.

## Keywords

Nuclear envelope; bleb; migration; actomyosin contractility; LAP1

## Introduction

Metastatic spread accounts for the majority of cancer-related deaths[1, 2] and there is an urgent need to understand how metastatic potential is acquired. Metastatic melanoma is the leading cause of death for skin cancers[3, 4]. Melanoma cells can switch between different collective and individual migratory modes, can degrade the extracellular matrix, and can reprogram cells in the tumour microenvironment to favour cancer cell survival, migration, and invasion[5–12]. Repeated physical constraints, for example traversing tissue constrictions or the vascular endothelium, are a major barrier to metastatic spread[13, 14]. While the cytoplasm can accommodate large deformations, the mechanical properties of the nucleus make translocation of this organelle the rate-limiting step during constrained migration[15]. Nuclear mechanical properties are regulated through interactions between the cytoskeleton, integral nuclear envelope (NE) proteins, the nuclear lamina and chromatin[16–18]. Disruption of the nuclear lamina or peripheral heterochromatin as well as unrestrained tensional or compressive force can cause NE membranes to rupture[19–26], leaving genomic DNA exposed to damaging agents in the cytoplasm and prone to persistent damage due to the mis-localisation of repair factors[27, 28]. NE ruptures occur typically at NE blebs, where intranuclear pressure can cause the outflow of nucleoplasm and sometimes chromatin[22, 25, 29]. As for plasma membrane blebs[30], actomyosin contractility regulates NE bleb dynamics[31, 32]. Whilst plasma membrane blebs can facilitate bleb-based migration[33–37], whether NE blebs contribute to cellular migratory programmes is unclear. The involvement of nucleo-cytoskeletal connections in NE bleb dynamics suggests that studying how these structures are regulated might shed light into how tumour cells withstand high-level mechanical stress. Here, we challenged melanoma cells derived from primary tumours and those derived from isogenic metastatic lesions to multiple migratory constraints and 3D invasion in collagen to identify proteins that enable cells to negotiate these challenges. Using a combination of transcriptomics, digital pathology, molecular cell biology and advanced microscopy we discovered that elevated expression of the inner nuclear membrane (INM) protein lamin-associated polypeptide 1 (LAP1) supports the ability of metastatic cells to overcome repetitive physical challenges through a migratory programme involving NE blebbing.

# Results

## Metastatic melanoma cells can negotiate repeated constraints

We designed a multi-round transwell migration assay using pores with a subnuclear diameter[14] (Fig. 1a) to understand the differential abilities of a pair of isogenic melanoma cell lines derived from the same patient (WM983A, derived from the primary tumour; WM983B, derived from a metastatic lesion) stably expressing nucleus-localised GFP (GFP-NLS) to negotiate repeated constraints. We found that during the first round of migration, whilst decreasing pore size impaired migration, WM983B cells could negotiate 8-μm and 5-μm pores better than WM983A cells (Fig. 1b and Extended Data Fig. 1a) and up to 10% nuclei displayed at least one NE bleb before and after pore transit (Fig. 1c and Extended Data Fig. 1b-d). On a second round of migration using sequentially 8-μm and 5-μm transwells, we found that WM983B cells, but not WM983A cells, could negotiate this second challenge with a similar efficiency to the first challenge and displayed enhanced NE blebbing (Fig. 1d-f and Extended Data Fig. 1e-g). We confirmed that WM983A and WM983B cells retained this migratory behaviour after a third round of migration using sequentially 8-μm, 8-μm and 5-μm pores, but NE blebbing was not further enriched (Extended Data Fig. 1h,i). Migration through constraints did not affect cell viability (Extended Data Fig. 2a,b) and neither the morphological features of apoptosis nor active Caspase 3 were present before or after a second round of migration (Extended Data Fig. 2c), suggesting that repetitive constraints do not activate a cell death programme.

WM983B cells display enhanced Rho-ROCK1/2-driven Myosin II (MLC2) activity compared to WM983A cells[6, 7, 9] (Extended Data Fig. 2d). Actomyosin contractility promotes migration under confinement[35, 36], NE blebbing and rupturing[25, 26, 31]. We reasoned that higher Rho-ROCK1/2-driven MLC2 activity and higher actomyosin contractility could contribute to both generation of NE blebs and enhanced migration of WM983B cells over WM983A cells. We treated cells with the ROCK1/2 inhibitor (ROCKi) GSK269962A and challenged them to one-round and two-round transwell assays. We confirmed that MLC2 activity was reduced after ROCK1/2 inhibition (Extended Data Fig. 2d,e) and found that ROCK1/2 inhibition reduced blebbing, but did not reduce nuclear translocation of WM983B cells during the first round of migration (Fig. 1g,h). However, ROCK1/2 inhibition markedly impaired nuclear translocation and reduced NE blebbing during the second round through 8-μm (Extended Data Fig. 2f,g) and 5-μm pores (Fig. 1i,j). We found that after passage through the first challenge, WM983B, but not WM983A, cells displayed higher MLC2 activity (Fig. 1k), suggesting that passage through the first constraint activates a Rho-ROCK1/2-dependent migration programme specifically in metastatic melanoma cells.

## The nuclear envelope of metastatic melanoma cells is dynamic

We hypothesised that melanoma cells that had previously negotiated multiple constraints during metastasis *in vivo* may have retained a nuclear mechanical memory. In unconfined cells, we found that 30% of WM983B cells displayed NE blebs compared to only 5% of WM983A cells (Fig. 2a). WM983B cells displayed more NE blebs per nucleus and higher bleb phenotypical variability (Fig. 2b,c). We observed NE blebs only in 1% of melanocytes

(Extended Data Fig. 3a,b). When grown on collagen I, WM983B cells were more rounded and harboured higher MLC2 phosphorylation levels than WM983A cells (Extended Data Fig. 3c-3e and 3i). We obtained similar results with another melanoma cell line pair (highly metastatic and highly amoeboid A375M2 cells and less metastatic and less amoeboid A375P cells[6, 9, 38] (Extended Data Fig. 3f-h and 3i). Under these conditions, 20% of WM983B cells displayed NE blebs compared to 10% of WM983A cells and 40% of A375M2 cells showed NE blebs compared to 10% of A375P cells (Extended Data Fig. 3j-k and 3l).

NE blebs form at regions of the NE with compromised structural integrity[19, 20, 22, 24, 25]. We found that about 90% of NE blebs present weak B-type lamin staining but persistent Lamin A/C staining (Fig. 2d,e). Chromatin was found in NE blebs and whilst blebs were positive for markers of double-strand DNA breaks (Extended Data Fig. 4a,b), overall levels of DNA damage in the culture were not increased (Extended Data Fig.4c). We found using correlative light and electron microscopy (CLEM) that NE blebs could either be intact or ruptured where nuclear membranes peeled away from the underlying lamina of the bleb with concomitant leakage of GFP-NLS into the cytosol (Fig. 2f and Supplementary Video 1-4). WM983B cells exhibited more ruptured NE blebs than WM983A cells (Fig. 2g). NE ruptures were abrogated by ROCK inhibition (Fig. 2h), implicating actomyosin contractility in their biogenesis and rupture. Cells reseal their ruptured NE blebs via ESCRT-III-dependent repair[24, 25]. We found that the average NE repair time was 10 minutes in both WM983A and WM983B cells (Fig. 2i), however, the rupture rate for WM983B cells was higher than in WM983A cells, with up to one event per hour (Fig. 2j, Extended Data Fig. 4d,e and Supplementary Videos 5 and 6). These results suggest that metastatic melanoma cells have a higher background level of NE instability.

## *TOR1AIP1* is upregulated in metastatic melanoma cells

As the degree of NE blebbing and migratory ability correlated with melanoma progression (Fig.1 and Extended Data Fig.1,3), we compared transcriptomes of A375M2 cells with A375P cells[5, 7, 9, 10] using Gene Set Enrichment Analysis (GSEA) and focusing on genes encoding nuclear proteins (Extended Data Fig. 5a). 63% of the gene sets containing genes encoding nuclear proteins were upregulated in A375M2 cells relative to A375P cells (FDR<5%) compared with only 1% in A375P cells relative to A375M2 cells (Extended Data Fig. 5b). About one third of upregulated gene sets in A375M2 cells were related to the nuclear membrane and organelle organisation (Extended Data Fig. 5b and Supplementary Tables 1-12). Leading-Edge Analysis[39] allowed us to identify a cluster of overlapping upregulated genes across NE gene sets (Extended Data Fig. 5c) and a final list of 7 candidate genes showing statistically significant upregulation was compiled (Extended Data Fig. 5e). Comparing transcriptomes of A375M2 cells to A375M2 cells treated with actomyosin contractility inhibitors (ROCK1/2 inhibitors H1152 or Y27632, or myosin II inhibitor blebbistatin) revealed similar gene expression changes (Extended Data Fig. 5d,e). Candidate gene upregulation in A375M2 cells compared to A375P cells was confirmed by RT-qPCR. Whilst all the genes were upregulated in A375M2 cells, *OSBPL8*, *SUMO1* and *TOR1AIP1* achieved statistical significance (Fig. 3a). We confirmed statistically significant upregulation of *TOR1AIP1* by RT-qPCR in WM983B cells compared to WM983A cells (Fig. 3b).

Expression of these genes was analysed in two publicly available datasets (Philadelphia and Mannheim[40]) containing transcriptomic profiles of melanoma cell lines compared to melanocytes. In both cases, upregulation of our candidate genes was observed in melanoma lines (Fig. 3c). Furthermore, using publicly available patient datasets (Kabbarah[41], Riker[42] and Xu[43]), we found that *TOR1AIP1* mRNA levels were consistently upregulated in human samples obtained from metastatic melanoma lesions compared to primary melanoma (Fig. 3d). These data suggest that expression of *TOR1AIP1* is upregulated during melanoma progression.

## LAP1 enables repeated constrained migration and invasion

Human cells express two isoforms of the protein encoded by *TOR1AIP1*, LAP1, that differ in the length of their amino terminus (NT); a long isoform, LAP1B, and a shorter isoform, LAP1C, generated by use of an alternative translation initiation codon at position 122[44, 45] (Fig. 4a). Consistent with our transcriptomic analyses, we found that both LAP1B and LAP1C were upregulated in metastatic melanoma cells relative to melanocytes and primary melanoma cells (Fig. 4b,c).

We took a loss of function approach to understand whether LAP1 contributes to NE blebbing and enhanced migration of WM983B cells. Whilst LAP1 isoforms were relatively resistant to siRNA depletion, we could reduce expression of LAP1 isoforms in WM983B cells, approximating levels observed in WM983A cells (Fig. 4d and Extended Data Fig. 6a). We performed two-round transwell migration assays and discovered that reducing LAP1 levels in WM983B cells suppressed both second-round migration efficiency and NE blebbing (Fig. 4e,f), without affecting cell viability (Extended Data Fig. 6b). We confirmed these results using two independent LAP1 siRNAs (Extended Data Fig. 6c-f) and found that reduced LAP1 expression levels decreased the ability of WM983B cells to invade into 3D collagen I matrices (Fig. 4g,h). Overall, these data suggest that as well as supporting NE-bleb generation, LAP1 allows metastatic melanoma cells to negotiate migratory constraints in the microenvironment.

## LAP1B and LAP1C are differentially tethered to nuclear lamins

We next set out to understand if LAP1 isoforms play different roles in NE blebbing and migration based on the distinct length of their NTs. We carried out a solubilisation assay, and in agreement with others[44, 46, 47] found that LAP1C was readily released from the nucleus of melanoma cells, whereas LAP1B was only released when the nuclear lamina was solubilised (Extended Data Fig. 7a). The NT of LAP1 interacts with Lamins[46] and two distinct lamin-binding regions in LAP1's NT have been described: 1-72 (present in the unique region of LAP1B) and 184-337 (present in both LAP1B and LAP1C)[48] (Extended Data Fig. 7b). Solubilising INM proteins for immunoprecipitation while retaining native interactions is challenging. We instead employed a mitochondrial retargeting assay and co-expressed mEmerald-Lamin A/C or mEmerald-Lamin B1 with LAP1 N-termini fused to HA and the mitochondrial targeting sequence from Monoamine Oxidase[49]. We found that mitochondria displaying HA-LAP1B$^{NT}$ or HA-LAP1B$^{1-121}$, but not HA-LAP1C$^{NT}$ were recruited to the nuclear periphery in cells expressing mEmerald-Lamin B1 or mEmerald-Lamin A/C (Extended Data Figure 7c,d). We next examined the ability of mitochondria

displaying HA-LAP1 N-termini to differentially recruit mEmerald-Lamins. We found that mitochondria displaying HA-LAP1B[NT] and HA-LAP1B[1-121] but not HA-LAP1C[NT], could recruit mEmerald-Lamin B1, but not mEmerald-Lamin A/C (Extended Data Figure 7c,d), suggesting that the unique NT of LAP1B encodes a dominant lamin-binding domain displaying preference for B-type lamins. We observed that LAP1B-mRuby3, like mEmerald-Lamin B1, was largely excluded from NE blebs, whereas both LAP1C-mRuby3 and mEmerald-Lamin A/C were readily detectable in NE blebs (Extended Data Fig. 8a,b). In addition, NE blebs with LAP1C also contained nucleoplasm, chromatin, and Emerin but not nuclear pore complexes (NPCs) (Extended Data Fig. 8c,d). We used RNAi to deplete individual nuclear lamins and found that endogenous LAP1 localised to NE blebs in the absence of Lamin B1 or Lamin B2, but not in the absence of Lamin A/C (Extended Data Fig. 8e-g). Lastly, using Fluorescence Recovery After Photobleaching (FRAP), we found that LAP1C-mRuby3 was more mobile than LAP1B-GFP both at the main NE and in NE blebs (Extended Data Fig. 8h-j and Supplementary Videos 7-9), in agreement with previous studies in other systems[48]. We suggest that LAP1C can move more freely in the INM than LAP1B and can populate NE blebs in a Lamin A/C-dependent manner.

## LAP1C supports constrained migration and invasion

We next wondered if LAP1 isoforms made a differential contribution to constrained cell migration and invasion. We generated WM983A cells stably expressing GFP-NLS and both LAP1 isoforms (LAP1-mRuby3), LAP1B-mRuby3 or LAP1C-mRuby3 (Extended Data Fig. 9a) and performed two-round transwell migration assays. We found that expression of either LAP1-mRuby3 or LAP1C-mRuby3, but not expression of LAP1B-mRuby3, increased both NE blebbing and migration efficiency of WM983A cells (Fig. 5a,b). We confirmed these results using A375P cells (Extended Data Fig. 9b,c). We found that expression of LAP1-mRuby3 and LAP1C-mRuby3, but not LAP1B-mRuby3, promoted invasion of WM983A cells into 3D collagen I matrices (Fig. 5c,d). We concluded that expression of LAP1C permits NE blebbing and facilitates constrained migration and invasion.

We investigated two factors that could influence the opposite roles of LAP1 isoforms: NE/lamina tethering and LAP1's interplay with the ER-luminal AAA-ATPase, Torsin-1A[50–52]. We generated versions of LAP1B lacking either the dominant lamin-binding domain (LAP1B[1-72]-mRuby3) or the chromatin binding region (CBR, LAP1B[CBR]-mRuby3). To enhance NE/lamina tethering, we created versions of LAP1C fused to the whole (LBR[NT]-LAP1C-mRuby3) or part (LBR[TRS]-LAP1C-mRuby3) of the Lamin B Receptor's (LBR) NT. We also created versions of LAP1 lacking the arginine finger thought to be required for Torsin-1A activation (LAP1B[R563G]-mRuby3 and LAP1C[R441G]-mRuby3)[50, 51] (Fig. 5e). Expression of LAP1B[1-72]-mRuby3 or LAP1B[R563G]-mRuby3 enhanced NE blebbing and migration of both WM983A and A375P cells in two-round transwell assays (Fig. 5f,g and Extended Data Fig 9d,e), although expression of LAP1B[CBR]-mRuby3 only increased NE blebbing and migration of WM983A cells (Fig. 5f,g and Extended Data Fig 9d,e). We confirmed that expression of RNAi-resistant versions of LAP1B[1-72]-mRuby3 or LAP1B[CBR]-mRuby3 in a background of LAP1-depletion allowed WM983A cells to generate NE blebs and migrate efficiently through a second constraint (Extended Data Fig. 9f,g). In line with our transwell migration results, we found that expression

of LAP1C-mRuby3, LAP1B $^{1-72}$-mRuby3 or LAP1B$^{R563G}$-mRuby3 enhanced invasion of WM983A cells into collagen I (Fig. 5h,i). Conversely, expression of LBR$^{NT}$-LAP1C-mRuby3 or LBR$^{TRS}$-LAP1C-mRuby3 did not enhance NE blebbing and constrained migration of WM983A cells (Fig. 5j,k) and restricted their invasion into collagen I (Fig. 5l,m). Interestingly, LAP1C$^{R441G}$-mRuby3 behaved similarly to LAP1C-mRuby3 in its ability to support NE blebbing and constrained migration suggesting that the strength of the NE/lamina tether is important in controlling Torsin activation (Extended Data Fig. 9h,i). We suggest that the strong NE/lamina tethering provided by LAP1B and its activation of Torsin-1A do not allow NE blebbing and constrained migration, whereas the weaker NE/lamina tethering provided by LAP1C allows NE blebbing and lets cells negotiate physical constraints.

## LAP1C promotes invasion *in vivo*

We hypothesized that LAP1 could contribute to the metastatic cascade by supporting local invasion into the dermis. We used orthotopic melanoma models where WM983A, WM983B, A375P or A375M2 cells were injected into the dermis of NOD Xenograft Gamma (NXG) and allowed to grow and invade the local tissue. In these models, we observe three defined regions: tumour body (TB), proximal invasive front (PIF) and distal invasive front (DIF)[9]. We found that the metastatic lines invaded into the dermis more than their less- or non-metastatic counterparts. Moreover, A375M2 were not only more invasive but also grew faster *in vivo*, highlighting the aggressiveness of this model (Fig. 6a-d and Extended Data Fig. 10a-f). Using digital pathology, we scored LAP1 intensity from 0 (very low) to 3 (very high) in individual tumour cell nuclei and observed that LAP1 expression was higher at the PIF compared to the TB and at the DIF compared to the PIF of these tumours (Fig. 6e-h). Moreover, across the PIF and DIF, WM983B and A375M2 tumours presented a higher proportion of cancer cells expressing very high levels of LAP1 compared to WM983A and A375P respectively (Fig. 6f,h). Tumours grown in severe combined immunodeficient (SCID) mice after subcutaneous injection retained a similar LAP1 expression pattern (Extended Data Fig. 10g-j), indicating that LAP1 is associated with aggressive features even when tumours grow outside the dermal environment.

Next, A375P cells bearing versions of LAP1-mRuby3 were challenged for their invasive growth potential after intradermal injection and compared to A375M2, our model of aggressive disease. Expression of LAP1C-mRuby3, LAP1B $^{1-72}$-mRuby3 or LAP1B$^{R563G}$-mRuby3 in A375P cells increased local invasion (Fig. 6i,j and Extended Data Fig. 10k,l), relative to A375P cells stably expressing LAP1B-mRuby3, as seen previously in our *in vitro* 3D invasion models (Fig. 5h,i). Increased invasion was accompanied by increased tumour growth in A375P LAP1B $^{1-72}$-mRuby3 or LAP1C-mRuby3 but not in LAP1B$^{R563G}$-mRuby3 expressing tumours (Fig. 6k). Interestingly, as we had seen *in vitro*, expression of LAP1B-mRuby3 did not increase local invasion *in vivo*, while growth was comparable to that observed for A375M2 tumours (Fig. 6i-k and Extended Data Fig. 10k,l). No differences in proliferation were observed *in vitro* (Extended Data Fig. 10m), suggesting that these cancer cells establish different interactions with the tumour microenvironment for their differential growth *in vivo*. Overall, our data shows that LAP1C supports tumour invasion both *in vitro* and *in vivo*.

**LAP1 levels increase in human melanoma progression**

Finally, we assessed LAP1 expression in tissue microarrays from two human melanoma patient cohorts (cohort A including 19 primary tumours and 14 metastases and cohort B with a total of 29 primary tumours and their matched metastases[9]) (Supplementary Tables 13 and 14). We found increased LAP1 expression at the IF compared to the TB of primary tumours and in metastatic lesions compared to primary tumours (Fig. 7a-f). We observed that tumour cells showing very high levels of LAP1 were enriched in metastases compared to primary melanomas (Fig.7 c,f). Importantly, we found that high LAP1 expression in the IF was associated with shorter disease-free survival (Fig. 7g), indicating that LAP1 levels are linked to worse prognosis. These results suggest that LAP1 could be a prognostic marker in melanoma.

## Discussion

Recent work recognises the importance of the cell nucleus in mechanosensing during constrained migration[53, 54]. Here, we identify the INM protein LAP1 as a master regulator of NE remodelling during melanoma progression.

Unlike plasma membrane blebs[30], NE blebs do not retract and can burst repeatedly and be resealed, allowing the dissipation of intranuclear pressure[19, 24, 25]. During constrained migration, NE blebs typically form at the leading edge of the nucleus through a combination of elevated intranuclear pressure, Nesprin-mediated pulling and actomyosin contractility, suggesting that a combination of external and internal forces acting on pliant membranes controls their biogenesis[32]. We found that the degree of NE blebbing and both migratory and invasive ability of melanoma cells correlated positively with the expression of LAP1. Of the two LAP1 isoforms encoded by *TOR1AIP1*, the short isoform (LAP1C) could be localised to and could support the biogenesis of NE blebs, suggesting a bleb-permissive LAP1C-phenotype may predominate in invasive melanoma with elevated levels of this protein. The biogenesis of these NE blebs was linked to the ability of cells to negotiate migratory constraints and was in-part linked to the strength of the NE-lamina interaction. Unlike LAP1C, expression of LAP1B did not enhance NE bleb formation or the ability of cells to negotiate physical constraints, unless its strong N-terminal lamin-binding domain or its ability to activate Torsin-1A was removed. LAP1B's N-terminal lamin-binding domain displayed a preference for Lamin B, and LAP1B and LAP1C's distribution in relation to NE blebs followed that of Lamin B and Lamin A/C respectively. These data suggest LAP1B may function to relay the less-deformable Lamin B environment to the activation state of Torsin-1A. LAP1C on the other hand, with its weaker NE/lamina tether, supports NE blebbing, migration and invasion both *in vitro* and *in vivo* suggesting that competition between isoforms may regulate the local activity of Torsin-1A in the intermembrane space. We speculate that LAP1C recruitment to structurally compromised regions of the NE or to NE blebs could allow rapid NE bleb expansion facilitating the observed detachment of the nuclear membranes from herniated chromatin (Fig. 2f) and the release of actomyosin-driven intranuclear pressure[31] (Fig. 7h).

In the light of our results, we propose that by fine tuning expression and localisation of LAP1 isoforms at the NE, cells may alter their array of nucleo-cytoskeletal connections

and Torsin-activation to modulate nuclear plasticity and allow their negotiation of physical constraints. We found that LAP1 could be a prognostic marker in human melanoma and propose that a better understanding of LAP1 function would not only provide a route to prevent metastatic dissemination but also may guide research on normal or pathological cell states featured by perturbed nuclear membranes.

## Methods

### Compliance

This research complies with all relevant ethical regulations; all animals were maintained under specific pathogen-free conditions and handled in accordance with the Institutional Committees on Animal Welfare of the UK Home Office (The Home Office Animals Scientific Procedures Act, 1986). All animal experiments were approved by the Ethical Review Process Committees at Barts Cancer Institute or King's College London and carried out under license from the Home Office, UK. For human tumour material, tumour samples were processed by IRB Lleida (PT17/0015/0027) and HUB-ICO-IDIBELL (PT17/0015/0024) Biobanks integrated in the Spanish National Biobank Network and Xarxa de Bancs de Tumors de Catalunya following standard operating procedures with approval of their Ethics and Scientific Committee and were collected with specific informed consent, in accordance with the Helsinki declaration. Compensation was not provided.

### Cell culture

The human melanoma cells WM1361 and WM1366 (CVCL_6789) were from Professor Richard Marais (Cancer Research UK Manchester Institute); the human melanoma cells WM793B (CVCL_8787), WM983A (CVCL_6808), WM983B (CVCL_6809) and WM88 (CVCL_6805) were purchased from the Wistar Collection at Coriell Cell Repository; the human melanoma cells A375P (CVCL_6233) and A375M2 (CVCL_C0RP) were from Dr. Richard Hynes (HHMI, MIT); the human primary melanocytes M206 and M443 were a kind gift from Dr. Benilde Jiménez (Universidad Autónoma de Madrid and Instituto de Investigaciones Biomédicas CSIC-UAM, Spain) and were isolated from foreskins obtained with informed written consent from healthy donors and under approval of the Institutional Review Board of Hospital Infantil Universitario Niño Jesus (Madrid, Spain); 293T (CVCL_0063) and CAL-51 (CVCL_1110) were from The Francis Crick Institute Cell Services STP. WM1361 and WM793B were cultured in Roswell Park Memorial Institute (RPMI, Gibco) medium supplemented with 10% of foetal bovine serum (FBS) and 1% (v/v) penicillin and streptomycin (PenStrep, Gibco); 293T, CAL-51, WM1366, WM983A, WM983B, WM88, A375P and A375M2 were cultured in Dulbecco's modified Eagle's media (DMEM, Gibco) supplemented with 10% FBS and 1% (v/v) PenStrep; M206 and M443 were cultured in MGM-4 basal growth medium (Lonza) supplemented with the Melanocyte Growth Medium-4 BulletKit (MGM-4, Lonza). Cells were grown at 37°C and 5% or 10% $CO_2$. Cells were kept in culture to a maximum of three or four passages.

### Plasmids

pTRIP-SFFV-EGFP-NLS and the coding sequences of mEmerald-Lamin A/C and mEmerald-Lamin B1 were from Addgene, plasmids 86677, 54138, 54140. pLVX-N-GFP

was a kind gift from Dr Michael Way (The Francis Crick Institute, UK) and was modified to express mEmerald-Lamin A/C and mEmerald-Lamin B1 by exchanging *SnaBI/BamHI* fragments. The coding sequence for LAP1, LAP1B carrying the deletion of the chromatin binding region (LAP1B $^{CBR}$) and LBR$^{TRS}$ and LBR$^{NT}$ were synthesised by Integrated DNA Technologies. Point mutations in LAP1 (M122A), LAP1B (LAP1B$^R$, LAP1B $^{1-72R}$, LAP1B$^{R563G}$) or LAP1C (LAP1C$^{R441G}$), deletion of lamin-binding residues 1-72 ( 1-72) in LAP1B and addition of part (LBR$^{TRS}$-LAP1C) or whole (LBR$^{NT}$-LAP1C) of LBR's NT in LAP1C were generated using standard PCR and gene synthesis. Of note, The M122A mutation in LAP1 was used to inactivate expression of LAP1C and ensure LAP1B alone was expressed. All PCR primers are in Supplementary Table 15. LAP1, LAP1B and LAP1C mutants were cloned into pCMS28-*EcoRI-NotI-XhoI*-Linker-mRuby3, a modified version of pCMS28[55] containing the 50-amino acid flexible linker from the Localisation And Purification (LAP) tag)[56]. LAP1 (M122A) was also cloned into pNG72-*EcoRI-NotI-XhoI*-Linker-GFP. LAP1B$^{NT}$, LAP1B$^{1-121}$ or LAP1C$^{NT}$ were cloned *EcoRI*/*NotI* into pCR3.1. A cDNA for the HA-tagged mitochondrial targeting sequence from monoamine oxygenase[49] was synthesised and cloned into the *NotI/XbaI* sites of these vectors.

## Transient transfection

CAL-51 or WM983B cells were seeded at a density of $8x10^4$ cells in 4-well or 24-well plates and transfected with 0.5 μg of vector using Lipofectamine 3000 (Invitrogen). Media was changed 6 hours post-transfection. At 48-hours post-transfection, CAL-51 cells were fixed with 4% formaldehyde (FA) for 15 minutes at room temperature (RT).

## Generation of stable cell lines

HEK293T cells were seeded at a density of $4x10^5$ cells/ml in 6-well plates. For lentiviral production, cells were transfected with 1.5 μg of lentiviral vector, 0.5 μg of pVSVG and 2 μg of HIV-1 pCMVd8.91 using Lipofectamine 3000. For retroviral production, cells were transfected with 1.5 μg of retroviral vector, 0.5 μg of pHIT-VSVG and 2 μg of MLV-GagPol using Lipofectamine 3000. Media was changed 6 hours post-transfection. Viral supernatants were collected 48 hours post-transfection, spun down, filtered (0.2 μm) and used to transduce melanoma cells. Antibiotic selection as required was started 48 hours post-infection.

## siRNA transfection

Melanoma cells were seeded at a density of $4x10^4$ cells/ml in 24-well plates or $2.5x10^5$ in 6-well plates and transfected one hour after seeding. Cells were transfected with 20 nM siGenome Smart Pool or On-Targetplus LAP1 siRNA oligonucleotides using Lipofectamine RNAimax (Invitrogen). siRNA oligonucleotides were from Dharmacon and are listed in Supplementary Table 16. Non-targeting siRNA was used as a control. Transwell migration and invasion assays were carried out 48-hours post-transfection.

## Transwell migration assays

Cells were starved in serum-free DMEM overnight and seeded at a density of $1.65x10^5$ cells/ml per insert in 24-well plates or $2.335x10^6$ cell/ml per insert in 6-well plates. DMEM

10% FBS was used as chemoattractant. Cells were allowed to migrate 16 hours in 24-well plates and 24 hours in 6-well plates. For multi-round transwell assays, cells were collected after one round (24 hours) from inserts in 6-well plates and seeded on inserts in 24-well plates. Cells were fixed with 4% FA for 15 minutes at RT. Transwell inserts were from Corning.

### Inhibitor treatments

ROCKi GSK269962A (Axon MedChem) was used at 1 μM and Staurosporin (Cell Guidance Systems) was used at 1 μM. In transwell assays, the inhibitor was added to the chemoattractant.

### Quantitative real-time PCR (qPCR)

Melanoma cells were seeded at a density of $1x10^5$ cells/ml in 12-well plates 24 hours prior to the experiment. RNA was extracted using Trizol (Life Technologies) following manufacturer's instructions. RNA was treated with DNA-free™ DNA Removal Kit (Life Technologies) and RNA purity was determined with a ND-1000 Nanodrop (Thermo Fisher Scientific). qPCRs were performed using 100 ng RNA, QuantiTect primer assays and Brilliant III SYBR Green QRT-PCR Kit (Agilent Technologies) in a ViiA 7 Real-Time PCR System (Thermo Fisher Scientific). GAPDH was used as loading control. The following qPCR primers from Qiagen were used: RANBP2 (QT00035378), TPR (QT00046242), OSBPL8 (QT00067102), SUMO1 (QT00014280), NUP50 (QT00081669), ZMPSTE24 (QT00025627), TOR1AIP1 (QT00070147).

### Western blotting

Melanoma cells were seeded at a density of $6x10^5$ cells/ml in 12-well plates and lysed the next day with LDS buffer 1X (Life Technologies). Lysates were denatured at 95°C for 5 minutes and sonicated. SDS-PAGE and western blotting was performed using standard procedures. Membranes were visualised and bands quantified using an Odyssey Fc (LI-COR). Loading controls were run on the same blot. Individual 680 and 800 channels were co-visualised in greyscale on the same image. Primary antibodies were: LAP1 (1:1000; #21459-1-AP), Lamin A/C (1:1000, #10298-1-AP), Lamin B1 (1:1000, #12987-1-AP), Lamin B2 (1:1000, #10895-1-AP) from Proteintech; pThr18/Ser19-MLC2 (1:750, #3674), MLC2 (1:750, #3672) from Cell Signalling Technology; GAPDH (1:10,000, #MAB374) from Merck. Secondary antibodies were: IRDye 680RD goat anti-rabbit IgG (1:10,000, #925-68071) and IRDye 800RD goat anti-mouse IgG (1:10,000, #925-32210) from LI-COR.

### LAP1 solubilisation assay

The assays performed were adapted from[44]. Protein solubilisation from melanoma cell lysates was carried out using the following extraction buffers supplemented with 1 mM DTT, 10 mM NaF, 10 mM β-glycerophosphate, 1 mM $Na_3VO_4$, 0.3 mM PMSF and cOmplete EDTA-free protease inhibitor tablet (Roche): 50 mM Tris-HCl pH 7.5 (hypotonic); 50 mM Tris-HCl pH 7.5, 1% Triton-X-100 (hypotonic + detergent); 100 mM Tris-HCl pH 7.5, 1% Triton-X-100 (low salt + detergent); 500 mM Tris-HCl pH 7.5, 1% Triton-X-100 (high salt + detergent). Protein lysates were incubated for 20 minutes on ice

and then centrifuged at 14,000 x g for 15 minutes. The supernatants were combined with 4X LDS buffer and the pellets solubilised with 1X LDS buffer.

### Cell culture on thick layers of collagen type I

Collagen I matrices were prepared using FibriCol (CellSystems) at 1.7 mg/ml. Collagen was left 4 hours to polymerise and melanoma cells were seeded on top at a density of $5 \times 10^3$ in 96-well plates. Medium was changed the next day to DMEM 1% FBS. After 24 hours, cells were fixed with 20% FA for 15 minutes at RT. For nuclear staining, cells were fixed with 2% FA for 30 minutes at RT.

### 3D invasion assays

Collagen I was prepared using FibriCol at 2.3 mg/ml. Melanoma cells were suspended in serum-free DMEM to a final concentration of $1.5 \times 10^4$ cells per 100 µl of collagen. Cells were centrifuged at 1,800 rpm and 4°C for 8 minutes, resuspended in collagen and seeded on glass bottom 96-well plates (Ibidi). Plates were centrifuged at 900 rpm for 5 minutes to get all the cells at the bottom. Collagen was left 4 hours to polymerise and DMEM 10% FBS was added on top. Cells were allowed to invade for 24 hours. Cells in collagen were fixed with 20% FA overnight at 4°C. The 3D invasion index was obtained dividing the number of invading cells at 50 µm by the total number of cells.

### Immunofluorescence staining

Cells in coverslips were fixed with 4% FA for 15 minutes at RT, permeabilised with 0.3%Triton-X-100 for 20 minutes, blocked with 4% bovine serum albumin (BSA) for 30 minutes and incubated sequentially with primary antibody in 4% BSA for two hours at RT and secondary antibody in 4% BSA for one hour at RT. DNA was stained with DAPI 1:1000 or Hoechst 1:5000 in PBS (Gibco). For DNA damage staining, cells were washed in PBS pre-fixation and incubated with CSK buffer (10 mM Pipes, pH 6.8, 100 mM NaCl, 300 mM sucrose, 3 mM $MgCl_2$, 10mM B-glycerol phosphate, 50mM NaF, 1mM EDTA, 1mM EGTA, 5mM $Na_2VO_3$, 0.5% Triton-X-100) for three minutes and again for 1 minute on ice. Cells were fixed with 4% FA for 20 minutes on ice, blocked with 10% goat serum for one hour at RT and incubated sequentially with primary antibody in 1% goat serum overnight at 4°C and with secondary antibody in 1% goat serum for one hour at RT. For nuclear staining in collagen I, cells were permeabilised with 0.5% Triton-X-100 for 30 minutes, blocked with 4% BSA overnight at 4°C, incubated with primary antibody in 4% BSA overnight at 4°C and with secondary antibody in 4% BSA for two hours at RT. Primary antibodies were: Lamin A/C (1:200, #10298-1-AP) from Proteintech or (1:200; #mab3538) from Millipore; Lamin B1 (1:200, #12987-1-AP), Lamin B2 (1:200, #10895-1-AP), LAP1 (1:200; #21459-1-AP) and Emerin (1:1000, #10351-1-AP) from Proteintech; pSer19-MLC2 (1:200, #3671) from Cell Signalling); HA.11 (1:500, #901503) from BioLegend; Gamma-H2AX (1:600, #05-636) from Merck; 53BP1 (1:600, #NB100305) from Novus Biologicals; Mab414 (1:200, #ab24609) from Abcam. Secondary antibodies were Alexa Fluor 488 and Alexa Fluor 555 (1:1000) from Life Technologies and raised in donkeys against the corresponding species. F-actin was stained using Alexa Fluor 546-phalloidin from Life Technologies and DNA with Hoechst 33342 from Invitrogen.

## Confocal fluorescence microscopy and image analysis

Images were acquired with a Dragonfly 200 high speed spinning disc confocal microscope (Andor). Imaging was carried out with 20X dry or 60X oil objectives. Z-stacks of 1-5 μm step-size distance were acquired from fixed cells in 2D, transwells and collagen I. A Z-range of 50 μm was used to assess 3D invasion into collagen I. Live cell imaging was performed at 37°C and 5% $CO_2$ in an environmental chamber. Movies were taken at 2-minutes intervals for 1-15 hours. Image analysis was carried out using ImageJ. Fluorescence signal intensities were quantified from pixel intensity in single cells relative to the areas of interest.

## Fluorescence recovery after photobleaching

Cells were seeded at a density of $8x10^4$ in 4-well chamber slides (Ibidi) 24 hours prior to the experiment. A LSM 880 Carl Zeiss microscope was used for imaging. Cells were kept at 37°C and 5% $CO_2$ in a sealed chamber. Cells were imaged with a 63 x 1.4 NA objective. First, 5 pre-bleached values were acquired and then squares of 1.3 X 1.3 (1.69 $\mu m^2$) at the main NE and at NE blebs were bleached with 50% laser intensity. Fluorescence recovery was measured every second for 200 cycles. The FRAP data was curated subtracting background and normalised.

## Correlative light and electron microscopy

Melanoma cells were seeded at a density of $2x10^5$ in 35 mm gridded glass-bottom dishes (MatTek) 24 hours prior to the experiment. Live cells were imaged until nuclear envelope rupture was spotted, at which point cells were fixed adding 8% (v/v) formaldehyde (Taab Laboratory Equipment Ltd, Aldermaston, UK) in 0.2 M phosphate buffer (PB) pH 7.4 to the cell culture medium (1:1) for 15 minutes. Cells were mapped using brightfield light microscopy to determine their position on the grid and tile scans were generated. Processing details for the EM can be found in the Supplementary Information.

## In vivo experiments

SCID (NOD.CB17-Prkdc[scid]/NcrCrl) mice were obtained from Charles River and NXG (NOD-*Prkdc[scid]-Il2rg[tm1]/Rj*) mice from Janvier-Labs. Mice were female 6-8 weeks old for all experiments. The QMUL Biological Services holding facility maintained a 7-hour light/ dark cycle an ambient temperature of 19-22°C and humidity of 50-60%. Tumour volume ($mm^3$) = length x width x height x 0.52. No randomisation was performed in both subcutaneous and intradermal mouse experiments. None of the studies required any treatment since tumour inoculation. Tumours were calipered and animals were sacrificed at the same time at the end point of the experiment. During injections mice were anaesthetised with isoflurane. The maximal tumour size permitted in the animal license is 1.5 $cm^3$ for a single tumour. In the case of two tumours (right/left flank) the combination of both tumour volumes could not reach 1.5 $cm^3$ either. We confirm that none of the experiments performed exceed the maximal tumour size allowed in our animal license. Subcutaneous tumours: 2 x $10^6$ cells (A375P, A375M2, WM983A and WM983B) in a volume of 50 μL in PBS -/- were subcutaneously injected into the flank of SCID mice. All four cell lines were EGFP positive and n = 8 mice per group were used for A375P, A375M2, WM983A and n = 4 mice were used for WM983B (28 mice total)[6]. A375P and A375M2 tumours were grown for 33 days

while WM983A and WM983B were grown for 55 days. Intradermal injections (wild-type study): 2 x 10$^5$ cells (WM983A, WM983B, A375P, A375M2) in a volume of 50 μL in PBS -/- were intradermally injected into the flank of NXG mice. Two injections per mouse were performed, obtaining tumours in two flanks. n = 8 mice/group were used (32 mice total). A total of 16 tumours were obtained for each group (64 tumours total). Note: For A375P intradermal injections two time points were considered (24 and 36 days) to examine changes in local invasion at early and late timepoints. WM983A and WM983B-derived tumours were grown for 42 days. Intradermal injections (LAP1 mutants): 2 x 10$^5$ cells (A375M2-GFP-NLS, A375P GFP-NLS LAP1B-mRuby3, A375P GFP-NLS LAP1C-mRuby3, A375P GFP-NLS LAP1B $^{1-72}$-mRuby3, A375P GFP-NLS LAP1B$^{R563G}$-mRuby3) were injected in a volume of 30 μL in PBS-/- through intradermal injection into NXG mice. One injection per mouse was performed in the flank. n = 10 mice/group were used. Total 50 mice. For information on sample sizes, power calculation, randomisation and blinding, please see the Supplementary Information.

## Immunohistochemistry

Whole sections from intradermal and subcutaneous tumours (see in vivo experiment section) and two tissue microarrays including a total of 48 primary melanoma tumours and 43 metastases, were used. Details of patient tumours used to generate the microarrays are described in Supplementary Tables 13 and 14. Patients diagnosed with melanoma were recruited from the Dermatology departments at Hospital Arnau de Vilanova (Lleida, Spain) and Hospital de Bellvitge (Barcelona, Spain) during the period 1992 to 2014. No selection on age or gender were made. All cases were diagnosed in the respective Pathology Departments as melanoma according to the latest AJCC criteria. The invasive front (IF) from patient or xenograft tumours were delimited as the tumour areas showing at least 50% contact with the matrix, as previously described[5, 8–10]. Each biopsy was represented by two cores (1 mm diameter) from the tumour body (TB) and two cores from the IF areas. All samples were formalin-fixed paraffin-embedded tissue. Processing stages are described fully in the Supplementary Information.

## Gene enrichment analysis

Normalised gene expression microarray data was obtained from GSE23764[10]. A375M2 were compared to A375P and to A375M2 treated 24 hours with contractility inhibitors (ROCK inhibitors H1152 or Y27632 or myosin inhibitor blebbistatin). A catalogue of nuclear gene sets was downloaded, and analyses were carried out using GSEA software (http://www.broadinstitute.org/gsea/index.jsp). Permutations were set to 1000, permutation type to gene-set and t-test was established for ranking. GO gene sets were classified according to GO ontologies. Upregulated gene sets were filtered based on FDR<5% and p-value<0.05.

## Statistics and reproducibility

Unpaired two-tailed t-test, one-way ANOVA and two-way ANOVA followed by Tukey's or Sidak's multiple comparisons tests as indicated in the figure legends were performed using GraphPad Prism (GraphPad Software, Inc). Survival curve estimation from human samples based on the Kaplan–Meier method test was performed using SPSS Statistics (IBM). All

results were obtained from at least three independent experiments unless otherwise stated. Data is plotted as boxplots with the mean, 25th and 75th quartiles indicated by the boxes and maxima and minima indicated by the whiskers, or bar charts with the mean ± standard error of the mean (SEM) from the indicated number of independent experiments. The mean of each independent experiment is depicted as a point on the plot. The threshold for statistical significance was set to a p-value of less than 0.05. Power calculations, randomisation and blinding for *in vivo* experiments are described in the online Supplemental Methods. Excluding these *in vivo* experiments, no statistical method was used to predetermine sample size, no data was excluded from the analysis, the experiments were not randomised and the investigators were not blinded to allocation during experiments and outcome assessment.

## Extended Data

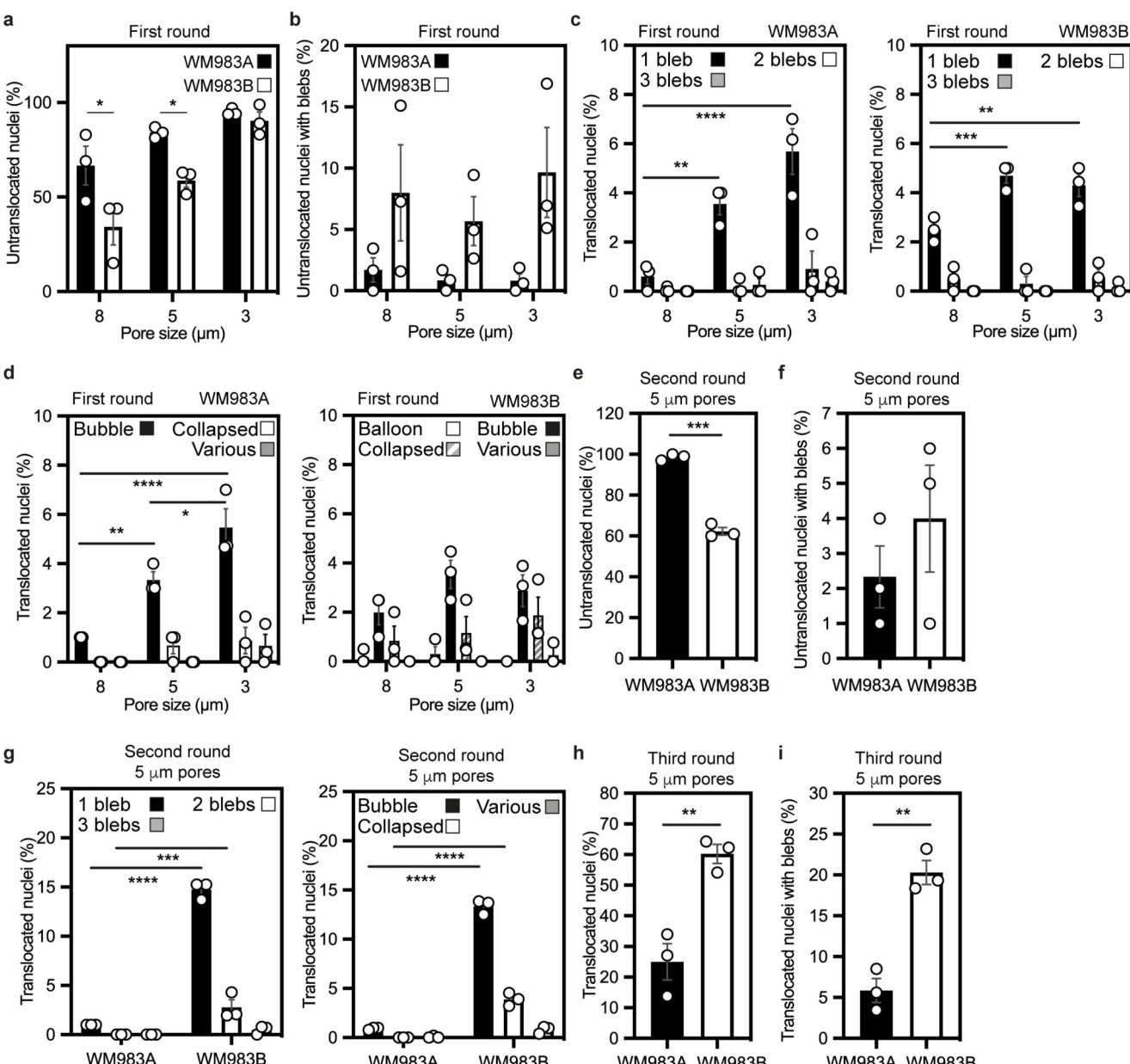

**Extended Data Figure 1. Multi-round transwell migration assays with melanoma cells**
**(a,b)** Percentage of primary melanoma WM983A and metastatic melanoma WM983B cells
that did not translocate their nuclei **(a)** and displayed nuclear envelope blebs **(b)** after
one round of migration in transwells of various pore sizes. **(c,d)** Percentage of WM983A
and WM983B cells that translocated their nuclei and displayed nuclear envelope blebs
classified according to bleb number (c) and bleb shape (d) after one round of migration
in transwells of various pore sizes. n= 1758 and 1728 cells, respectively. **(e,f)** Percentage
of WM983A and WM983B cells that did not translocate their nuclei **(e)** and displayed
nuclear envelope blebs **(f)** after a second round of transwell migration through 5-μm pores.

**(g)** Percentage of WM983A and WM983B cells that translocated their nuclei and displayed nuclear envelope blebs classified according to bleb number (left) and bleb shape (right) after a second round of transwell migration through 5-μm pores. n= 192 and 313 cells, respectively. **(h,i)** Percentage of WM983A and WM983B cells that translocated their nuclei **(h)** and displayed nuclear envelope blebs **(i)** after a third round of transwell migration through 5-μm pores. n= 536 and 606 cells, respectively. Bar charts show the mean and error bars represent S.E.M. from N = 3 independent experiments. P-values calculated by two-way ANOVA (a,b,c,d,g), and unpaired two-tailed t-test (e,f,h,i); *p<0.05, **p<0.01, ***p<0.001, ****p<0.0001. Numerical data and exact p-values are available in the Source Data.

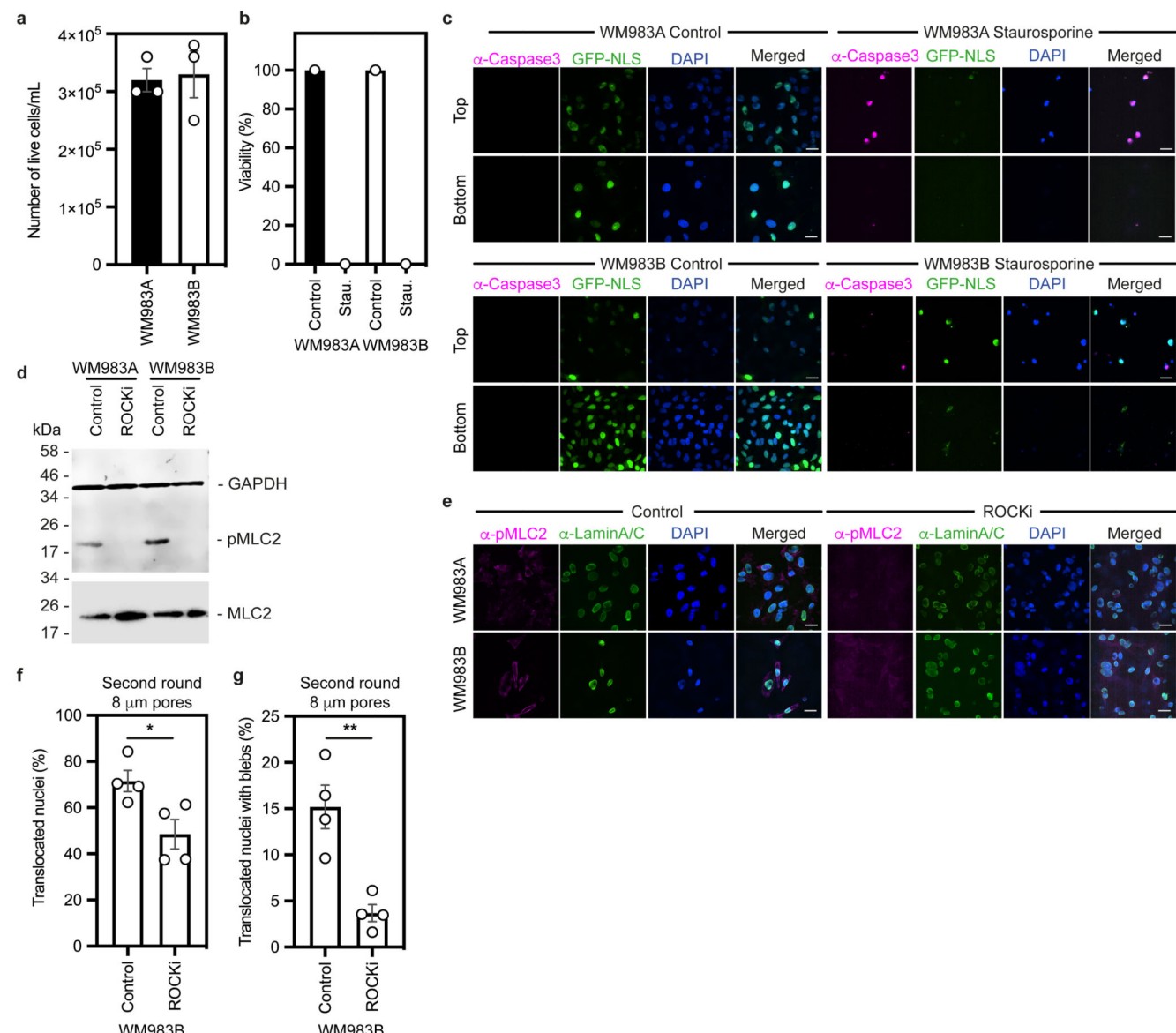

**Extended Data Figure 2. Cell viability and actomyosin contractility in multi-round transwell migration assays with melanoma cells**

(a) Percentage of alive primary melanoma WM983A cells and metastatic melanoma WM983B cells after a first round of migration in transwells through 8-μm pores (b) Percentage of alive WM983A and WM983B cells after a second round of migration in transwells through 5-μm pores upon staurosporin treatment. n= 180 and 213 cells, respectively (N=1). (c) Representative pictures of WM983A and WM983B cells expressing GFP-NLS (green) and stained for caspase-3 (magenta) and DNA (blue) after a second round of migration in transwells through 5-μm pores in the presence or absence of staurosporin. Scale bars, 30 μm. (d) Resolved cell lysates of WM983A and WM983B cells after treatment with ROCK inhibitor (ROCKi) GSK269962A were examined by western blotting with antisera raised against GAPDH, MLC2 and pMLC2. (e) Representative pictures of WM983A and WM983B cells treated with ROCKi and stained for pMLC2 (magenta), lamin A/C (green) and DNA (blue) after a first round of transwell migration through 8-μm pores. Scale bars, 30 μm. (f,g) Percentage of WM983B cells that translocated their nuclei (f) and displayed nuclear envelope blebs (g) after a second round of transwell migration through 8-μm pores upon ROCK inhibitor (ROCKi) GSK269962A treatment. n= 787 and 691 cells, respectively. Bar charts show means and error bars represent S.E.M. from N = 3 independent experiements, unless otherwise stated. P-values calculated by unpaired two-tailed t-test (a,f,g). *p<0.05, **p<0.01. Unprocessed western blots, numerical data and exact p-values are available in the Source Data.

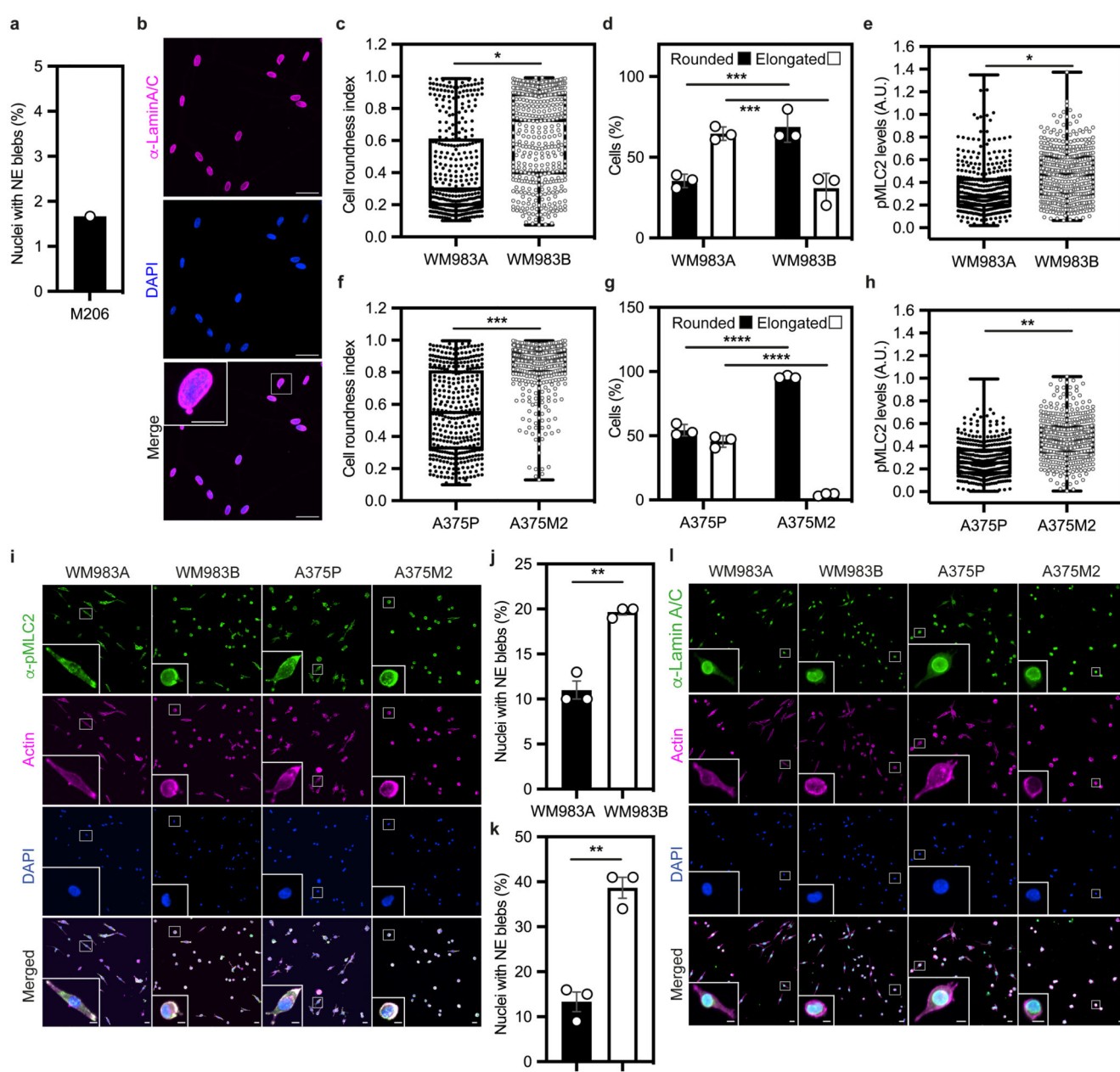

**Extended Data Figure 3. Cell and nuclear morphology changes in melanoma progression**
**(a)** Percentage of nuclei with nuclear envelope blebs in M206 melanocytes. n= 120 cells.
N=1. **(b)** Representative image of M206 melanocytes stained for lamin A/C (magenta) and
DNA (blue). Scale bars, 30 μm. The magnification shows a M206 nucleus with a nuclear
envelope bleb. Scale bar, 10 μm. **(c-e)** Cell roundness index **(c)**, percentage of elongated and
rounded cells **(d)** and pMLC2 levels **(e)** in primary melanoma WM983A cells and metastatic
melanoma WM983B cells grown on collagen I. n= 375 and 425 cells, respectively. **(f-h)**
Cell roundness index **(f)**, percentage of elongated and rounded cells **(g)** and pMLC2 levels
**(h)** in less metastatic melanoma A375P cells and highly metastatic melanoma A375M2
cells grown on collagen I. n= 385 and 381 cells, respectively. **(i)** Representative images

of WM983A, WM983B, A375P and A375M2 cells stained for pMLC2 (green), actin (magenta) and DNA (blue) on collagen I. Scale bars, 30 μm. The magnifications show representative cells and in the magnification the scale bars are 10 μm. **(j)** Percentage of nuclei with nuclear envelope blebs in WM983A and WM983B cells grown on collagen I. n= 337 and 373 cells, respectively. **(k)** Percentage of nuclei with nuclear envelope blebs in A375P and A375M2 cells grown on collagen I. n= 335 and 331 cells, respectively. **(l)** Representative images of WM983A, WM983B, A375P and A375M2 cells stained for lamin A/C (green), actin (magenta) and DNA (blue) on collagen I. Scale bars 30 μm. In **a,d,g,j,k** Bar charts show the mean and error bars represent SEM from N = 3 independent experiments. In **c,e,f,h** horizontal lines show the median and whiskers show minimum and maximum range of values. p values calculated by two-way ANOVA (d,g) and unpaired two-tailed t-test (c,e,f,h,j,k); **$p<0.01$, ***$p<0.001$, ****$p<0.0001$. Numerical data and exact p-values are available in the Source Data.

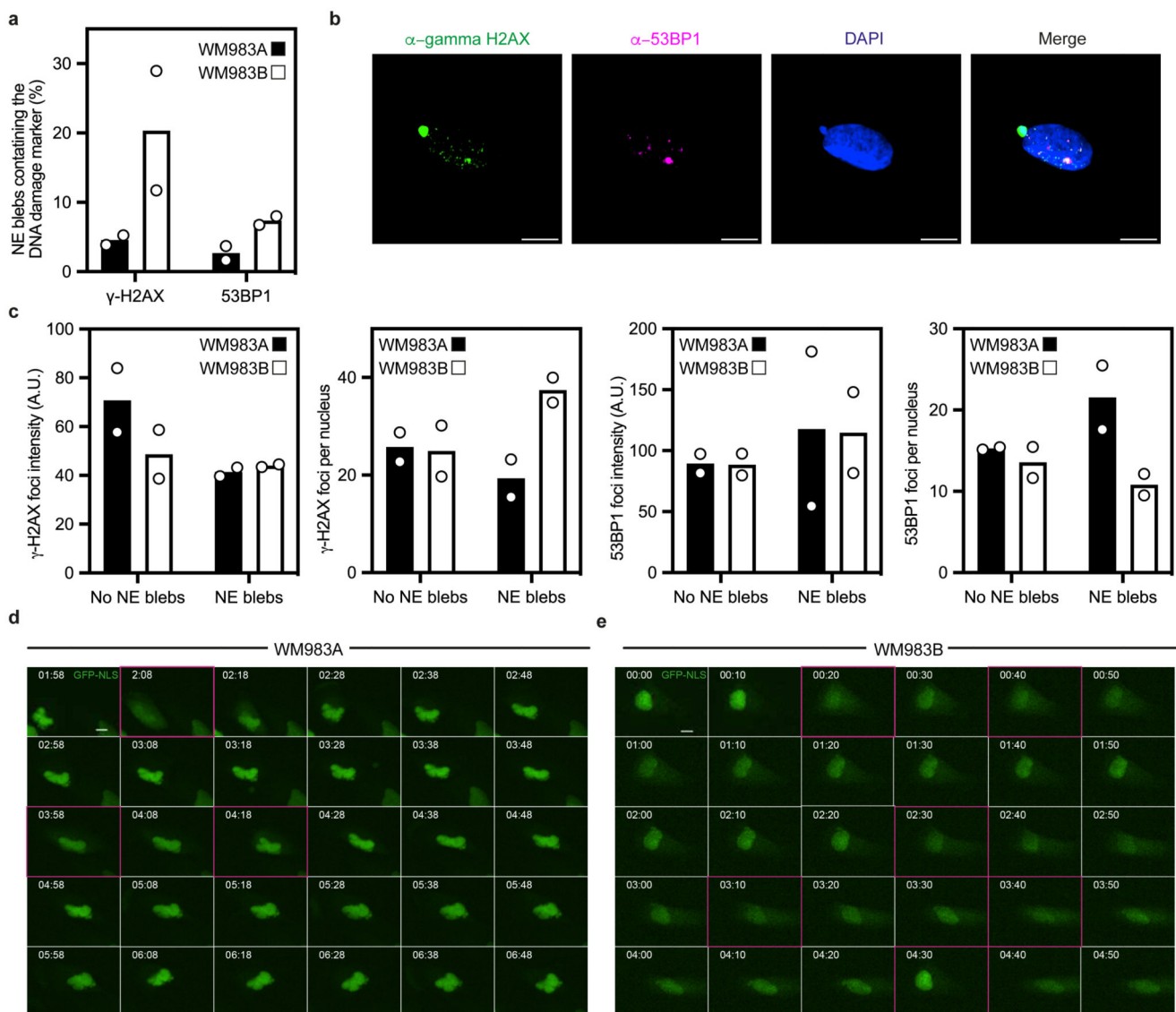

**Extended Data Figure 4. Instability of nuclear envelope blebs in melanoma cells**
**(a)** Percentage of nuclear envelope blebs positive for DNA damage response markers
(53BP1 and γ-H2AX) in primary melanoma WM983A cells and metastatic melanoma
WM983B cells. N=2. **(b)** Representative pictures of a WM983B nucleus stained for γ-
H2AX (green), 53BP1 (magenta) and DNA (blue). Scale bars, 10 μm. **(c)** Relative γ-H2AX
and 53BP1 fluorescence intensity levels (left) and average γ-H2AX and 53BP1 foci per
nucleus (right) in WM983A and WM983B cells with or without nuclear envelope blebs.
n= 246 and 283 cells, respectively. **(d,e)** Representative image sequence from repetitive,
transient nuclear envelope rupture events in WM983A cells **(d)** or WM983B cells **(e)** stably
expressing GFP-NLS and imaged over 5 hours. Images showing the first frame of a nuclear
envelope rupture in (d) or (e) are surrounded by a magenta square. Scale bars, 10 μm.
Numerical data are available in the Source Data.

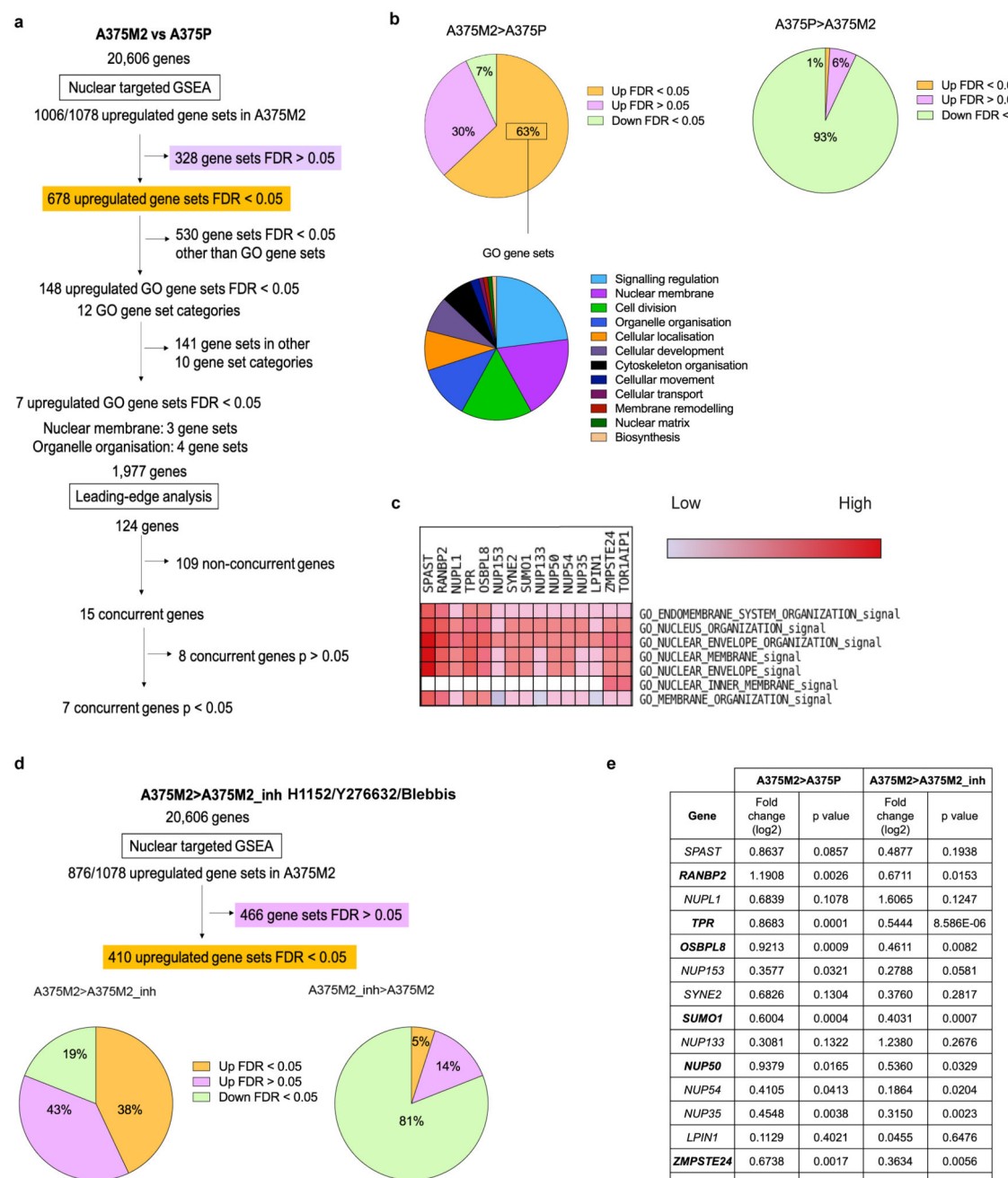

**Extended Data Figure 5. Transcriptomics reveals *TOR1AIP1* upregulation in metastatic melanoma cells**

**(a)** Flow diagram of GSEA carried out comparing highly amoeboid and highly metastatic melanoma A375M2 cells with less amoeboid and less metastatic melanoma A375P cells from data of gene expression microarray analysis. **(b)** The upper pie charts show the distribution of upregulated and downregulated nuclear gene sets in A375M2 cells (left) and A375P cells (right). The lower pie chart shows the 12 most enriched GO gene set categories in A375M2. **(c)** Section of heatmap of the leading-edge analysis showing a cluster

of upregulated nuclear envelope genes in A375M2 cells. **(d)** Flow diagram of the GSEA carried out comparing A375M2 cells with A375M2 cells treated with contractility inhibitors (ROCK inhibitors H1152 or Y27632 or myosin inhibitor blebbistatin) (A375M2_inh). The graphs show the distribution of upregulated and downregulated nuclear gene sets in A375M2 cells (left) and A375M2_inh (right). **(e)** Upregulated nuclear envelope genes selected from leading-edge analysis with their log2 fold change in expression and statistical significance in A375M2 compared to A375P cells and in A375M2 compared to A375M2_inh. Statistically significant upregulated genes appear in bold in the table. P-values calculated by two-tailed unpaired t-test.

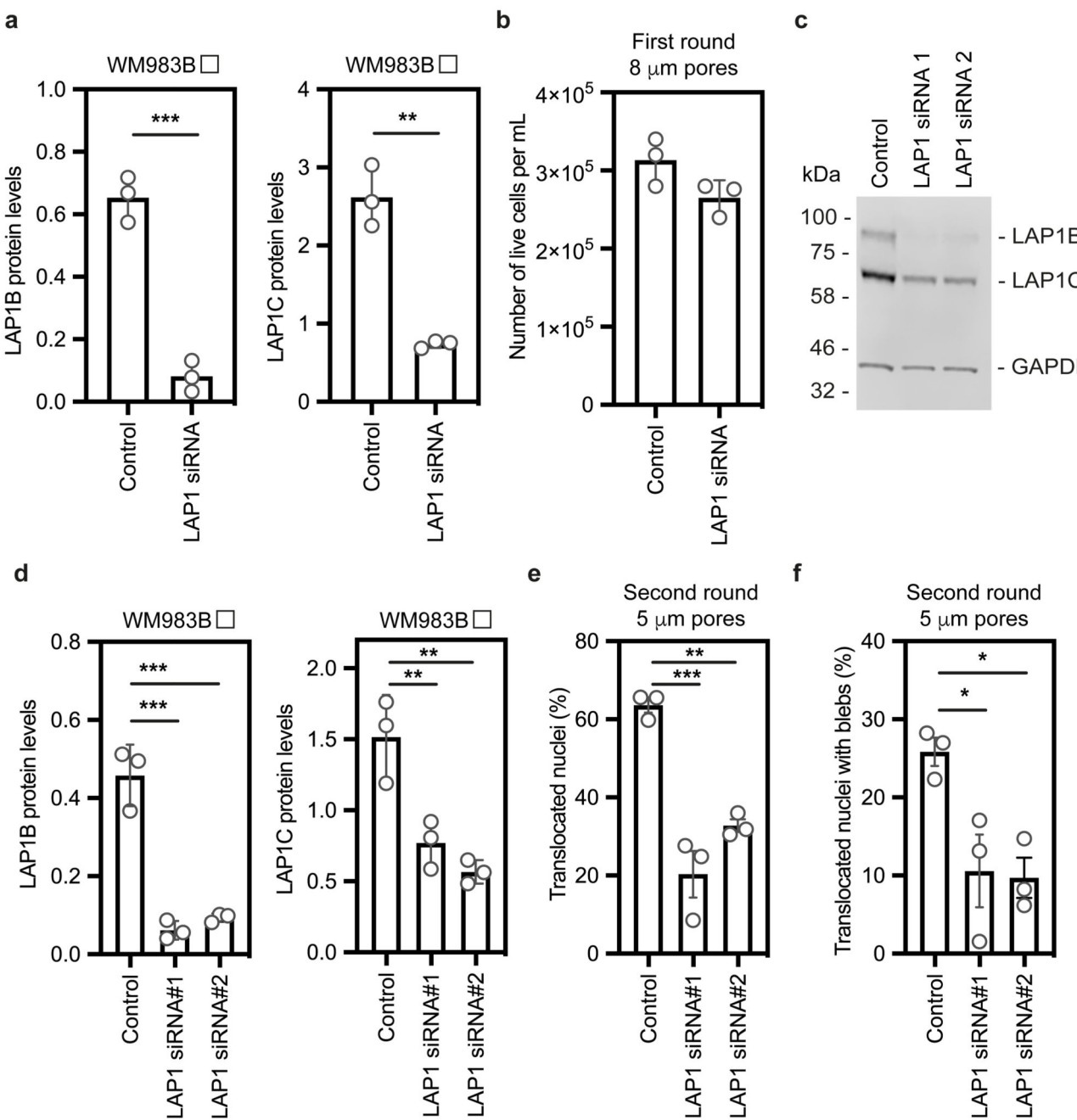

**Extended Data Figure 6. Impact of LAP1 expression levels on metastatic melanoma cells**
**(a)** Quantification of LAP1B (left) and LAP1C (right) expression in metastatic melanoma WM983B cells after 48 hours of treatment with a siGENOME SMARTpool targeting LAP1. **(b)** Percentage of alive WM983B cells after a first round of transwell migration upon LAP1 depletion with a siGENOME SMARTpool. **(c)** Representative immunoblot for LAP1 expression levels in WM983B cells after 48 hours of LAP1 depletion with ON-TARGETplus individual siRNAs. Western blot was probed with anti-GAPDH and anti-LAP1 antisera **(d)** Quantification of LAP1B (left) and LAP1C (right) expression in

WM983B cells after 48 hours of LAP1 depletion with ON-TARGETplus individual siRNAs as in (c). **(e,f)** Percentage of WM983B cells that translocated their nuclei **(e)** and displayed nuclear envelope blebs **(f)** after a second round of transwell migration through 5-μm pores upon LAP1 depletion with ON-TARGETplus individual siRNAs. n= 772, 632 and 590 cells, respectively. Bar charts show the mean and error bars represent S.E.M. from N = 3 independent experiments. P-values calculated by one-way ANOVA (d,e,f) and unpaired two-tailed t-test (a,b); *p<0.05, **p<0.01, ***p<0.001, ****p<0.0001. Unprocessed western blots, numerical data and exact p-values are available in the Source Data.

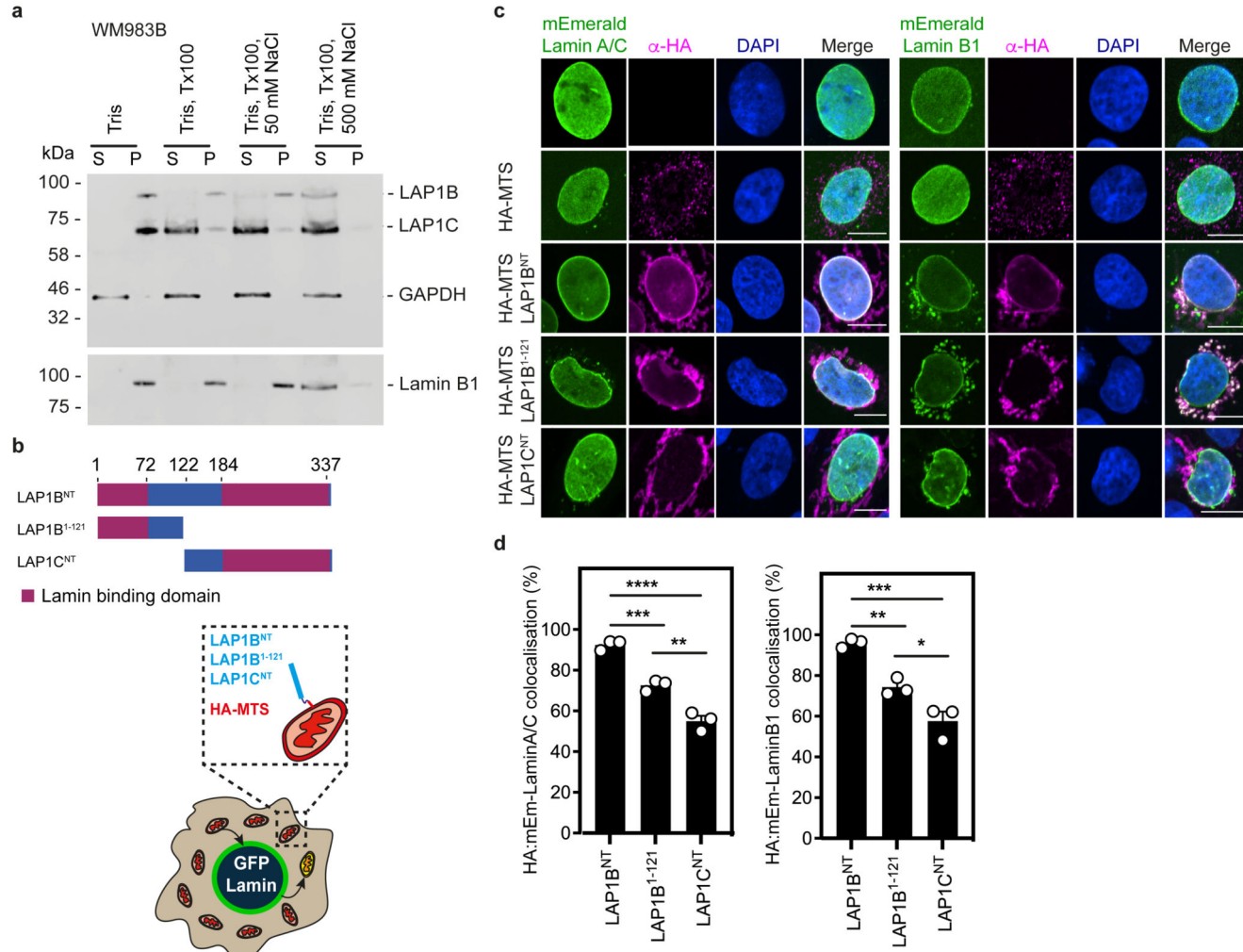

**Extended Data Figure 7. LAP1B and LAP1C are differentially tethered to nuclear lamins**
**(a)** Resolved soluble (S) and insoluble (P) fractions from WM983B cells subject to the indicated extractions were examined by western blotting with antisera raised against LAP1, GAPDH or Lamin B1. Western blot representative of N = 2 independent experiments. **(b)** Schematic of the different N-terminal domains of LAP1 isoforms and cartoon illustrating the mitochondrial retargeting assay. **(c)** Representative images of mitochondrial retargeting assay in cells expressing mEmerald-Lamin A/C (left) or mEmerald-Lamin B1 (right)

and stained for HA (magenta) and DNA (blue). Scale bars, 10 μm. **(d)** Percentage of cells showing co-localisation of mEmerald-Lamin A/C (left, n= 210, 211 and 190 cells, respectively) or mEmerald-Lamin B1 (right, n= 211, 199 and 183 cells, respectively) with LAP1B$^{NT}$/ LAP1B$^{1-121}$/LAP1C$^{NT}$. Bar charts show the mean and error bars represent S.E.M. from N = 3 independent experiments. P-values calculated by one-way ANOVA (d); *p<0.05, **p<0.01, ***p<0.001, ****p<0.0001. Unprocessed western blots and numerical data are available in the Source Data.

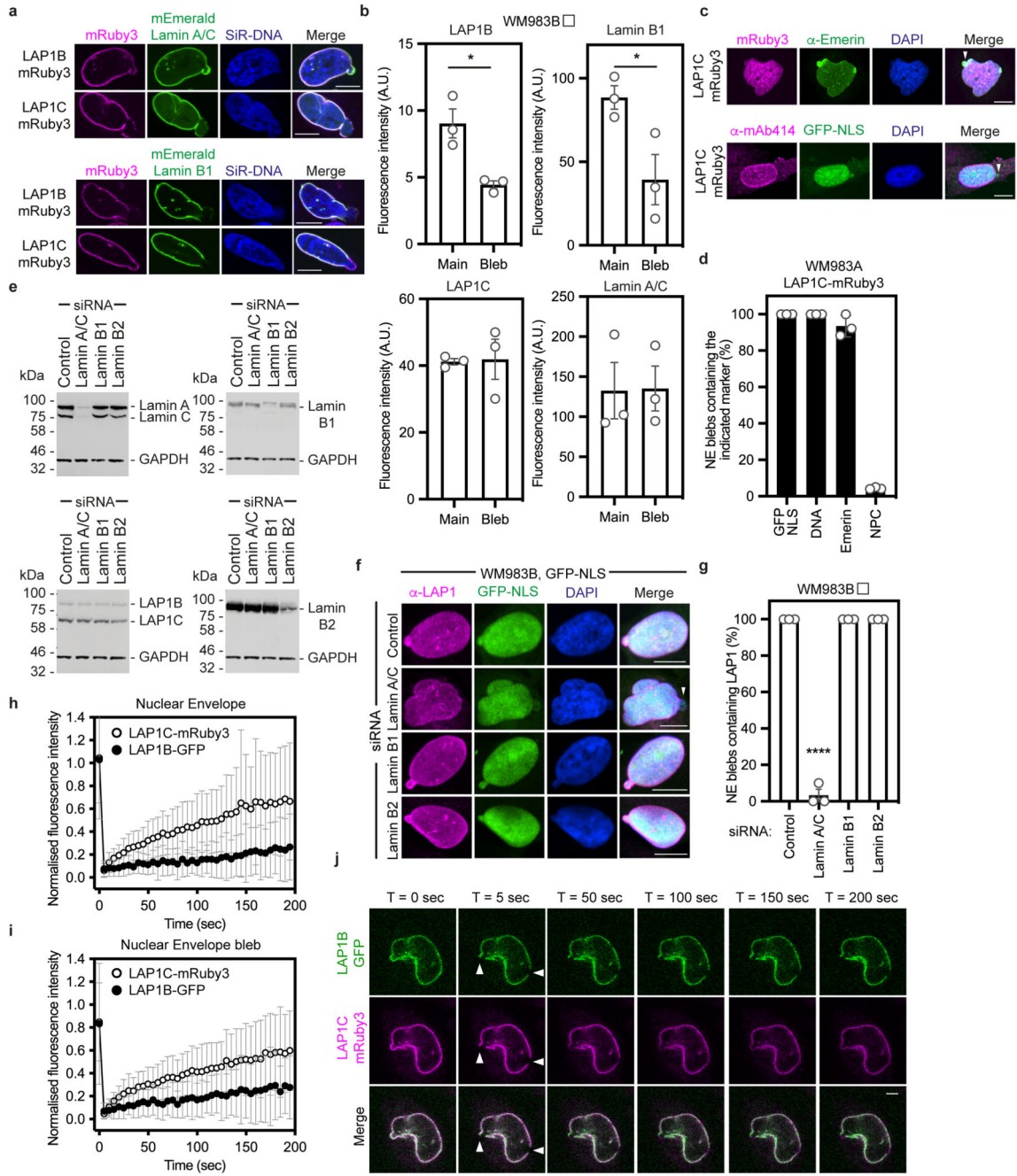

**Extended Data Figure 8. Lamin-binding properties of LAP1 isoforms**

**(a)** WM983B cells stably expressing LAP1B-mRuby3 or LAP1C-mRuby3, transfected with mEmerald-lamin A/C or mEmerald-lamin B1, stained with SiR-DNA and displaying nuclear envelope blebs. Scale bars, 10 μm. **(b)** Levels of LAP1B-mRuby3, LAP1C-mRuby3, mEmerald-Lamin A/C and mEmerald-Lamin B1 in the main nucleus and in NE blebs of WM983B cells stably expressing the indicated LAP1-mRuby proteins. n= 43, 63, 30 and 42, respectively. **(c)** WM983A cells stably expressing LAP1C-mRuby3 and stained for Emerin and DNA (top) or NPCs (bottom). NE blebs indicated by arrowhead. Scale bars, 10 μm **(d)** Quantification of data from (c). n= 671 and 605 for anti-Emerin and anti-NPC staining, respectively. **(e)** Representative immunoblots for lamin A/C, LAP1, lamin B1 and lamin B2 expression levels in WM983B cells after 72 hours of siRNA depletion of lamin A/C, lamin B1 or lamin B2 with siGENOME SMARTpools. **(f)** Representative pictures of WM983B stably expressing GFP-NLS (green) and stained for endogenous LAP1 (magenta) and DNA (blue) and transfected with the indicated siRNA. NE blebs without LAP1 staining indicated by arrowhead. Scale bars, 10 μm. **(g)** Percentage of nuclear envelope blebs containing LAP1 upon depletion of lamin A/C, lamin B1 or lamin B2 with siGENOME SMARTpools in WM983B cells. **(h-j)** FRAP analysis of LAP1 isoforms at the main nuclear envelope **(h)** and at nuclear envelope blebs **(i)** in metastatic melanoma WM983B cells stably co-expressing LAP1B-GFP and LAP1C-mRuby3. n= 26 cells. **(j)** Representative images from h and i. Scale bars, 5 μm. White arrowheads indicate bleaching at the main nuclear envelope and at the nuclear envelope bleb. In **b,d,g**, graphs show the mean and error bars represent S.E.M. from N = 3 independent experiments. In **i,j** graphs show the mean and error bars represent S.D. P-values calculated by one-way ANOVA (g); *p<0.05, ****p<0.0001. Numerical data and exact p-values are available in the Source Data.

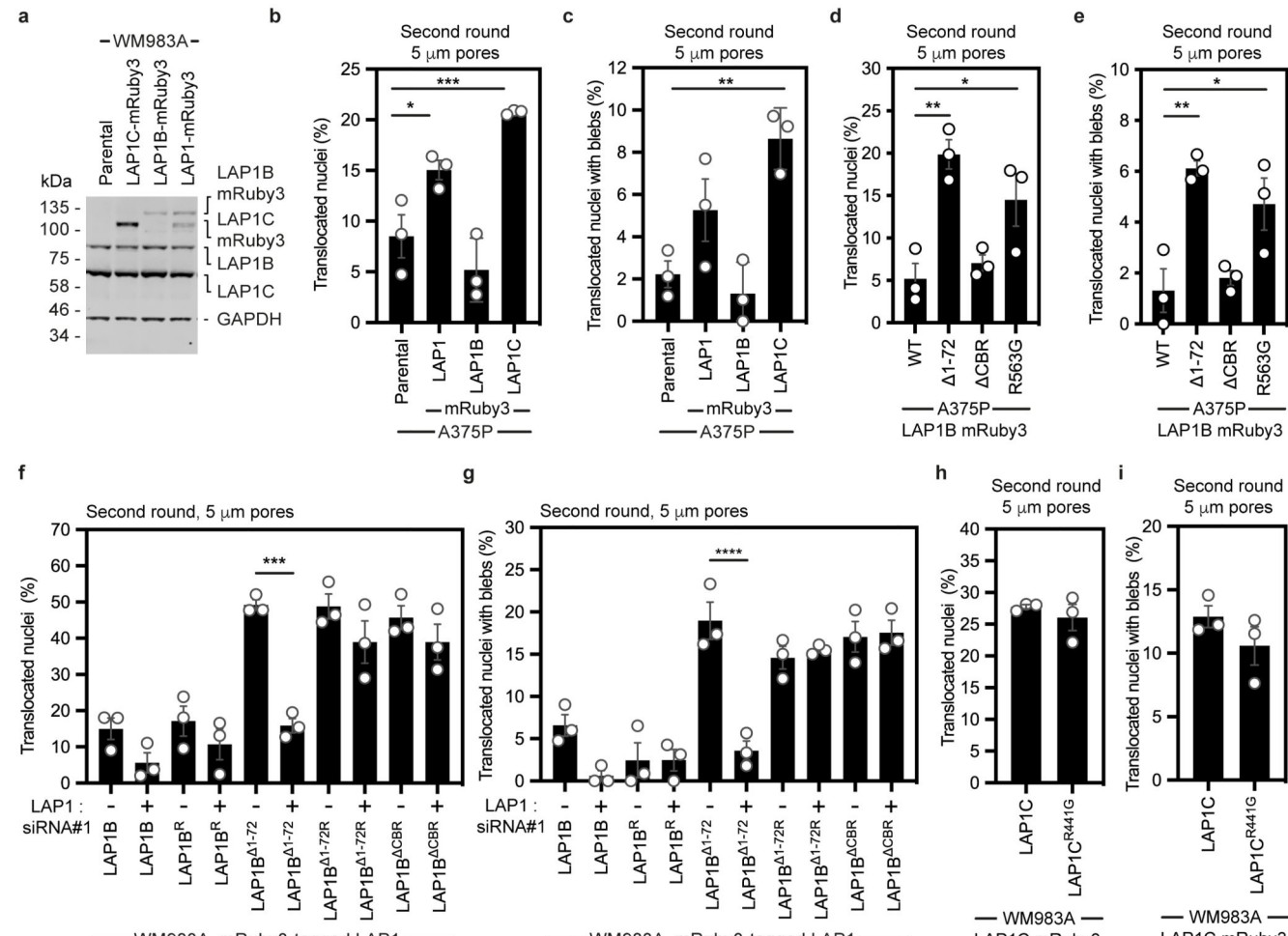

**Extended Data Figure 9. Effects of LAP1 overexpression *in vitro***
(a) Representative immunoblot for LAP1 expression levels in primary melanoma WM983A cells stably expressing GFP-NLS and LAP1C-mRuby3, LAP1B-mRuby3 or LAP1-mRuby3, related to Figure 5a-5d. (b,c) Percentage of less metastatic A375P cells stably expressing GFP-NLS, A375P stably expressing GFP-NLS and LAP1-mRuby3, LAP1B-mRuby3 or LAP1C-mRuby3 that translocated their nuclei (b) and displayed nuclear envelope blebs (c) after a second round of transwell migration through 5-μm pores. n= 374, 316, 344 and 439 cells, respectively. (d,e) Percentage of metastatic melanoma A375P cells stably expressing GFP-NLS and LAP1B-mRuby3, LAP1B $^{\Delta 1-72}$-mRuby3, LAP1B $^{\Delta CBR}$-mRuby3 or LAP1B$^{R563G}$-mRuby3 that translocated their nuclei (d) and displayed nuclear envelope blebs (e) after a second round of transwell migration through 5-μm pores. n= 344, 412, 273, and 317 cells, respectively. (f,g) Percentage of primary melanoma WM983A cells stably expressing GFP-NLS and LAP1B-mRuby3 or LAP1B $^{\Delta 1-72}$-mRuby, or siRNA resistant LAP1B-mRuby3 (LAP1B$^R$-mRuby3), LAP1B $^{\Delta 1-72}$-mRuby3 (LAP1B $^{\Delta 1-72R}$-mRuby3) or LAP1B $^{\Delta CBR}$-mRuby3 that translocated their nuclei (f) and displayed nuclear envelope blebs (g) after a second round of transwell migration through 5-μm pores upon treatment with LAP1 siRNA#1. n= 322, 269, 305, 357, 368, 363, 400, 369, 372 and 375 cells, respectively.

Note, OnTarget LAP1 siRNA#1 targets a sequence present within the CBR of LAP1. (**h,i**) Percentage of WM983A cells stably expressing GFP-NLS and LAP1C-mRuby3 or LAP1C$^{R441G}$-mRuby3 that translocated their nuclei (**h**) and displayed nuclear envelope blebs (**i**) after a second round of transwell migration through 5-μm pores. n= 418 and 388 cells, respectively. Graphs show the mean and error bars represent S.E.M. from N = 3 independent experiments. P-values calculated by one-way ANOVA (b-g) and two-tailed t-test (h, i; n.s.), *p<0.05, **p<0.01, ***p<0.001, ****p<0.0001. Unprocessed western blots, numerical data and exact p-values are available in the Source Data.

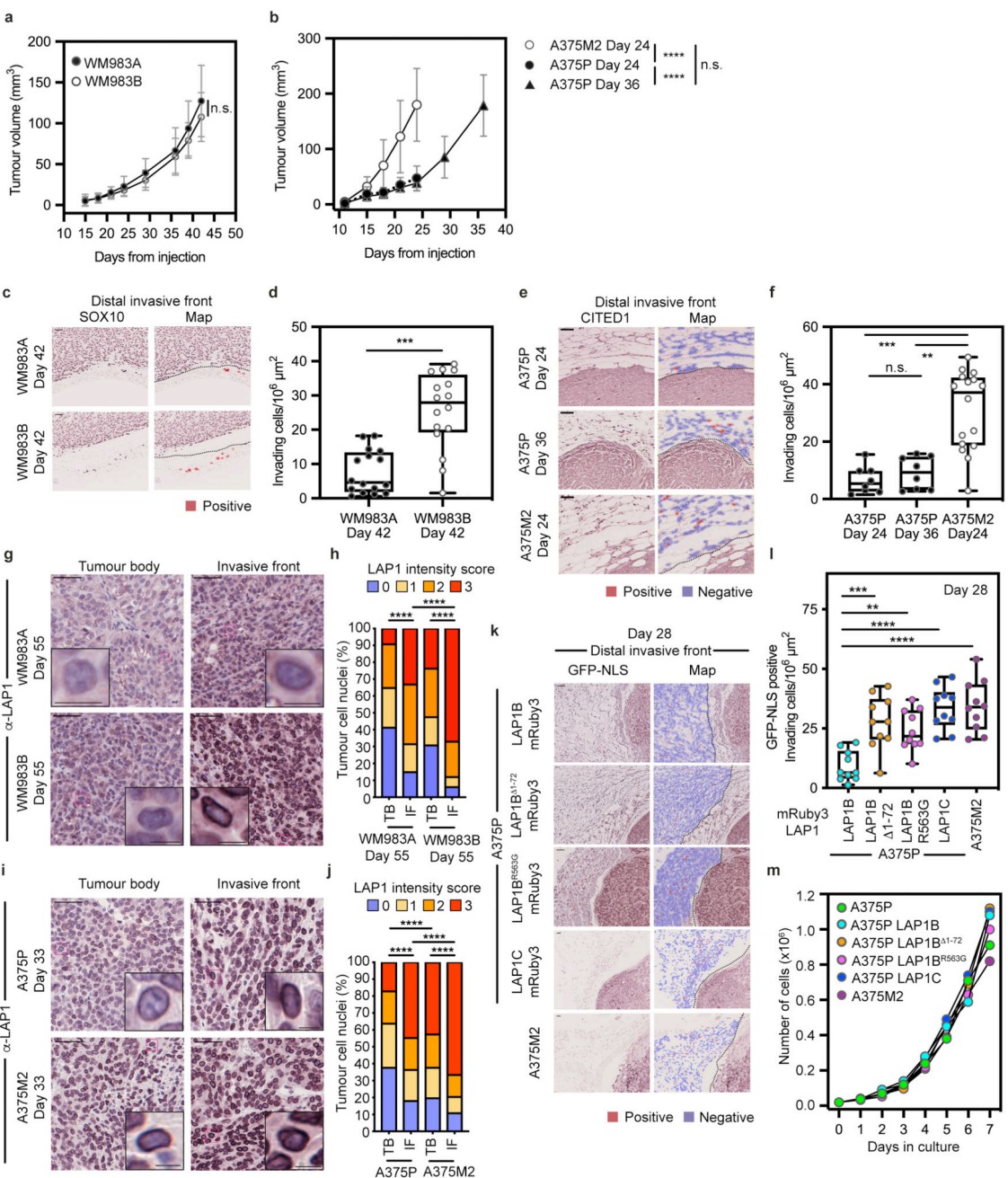

**Extended Data Figure 10. Effects of LAP1 overexpression *in vivo***
(**a, b**) Growth curves of WM983A, WM983B, A375P or A375M2 tumours grown intradermally in NXG mice. A375P endpoints considered at same-time and same-volume as A375M2. (**c**) Representative images and QuPath mark-up (map, red) of SOX10 staining and (**d**) quantification of dermal invasion in Day 42 WM983A and WM983B tumours grown intradermally in NXG mice. Dashed lines represent the boundary between proximal invasive front (PIF) and the distal invasive front (DIF). n= 8 and 8 mice, respectively. (**e**) Representative images and QuPath mark-up (map, red) of CITED1 staining and (**f**)

quantification of dermal invading cells in Day 24 and Day 36 A375P tumours and Day 24 A375M2 tumours grown in NXG mice. n= 4, 4 and 8 mice, respectively. **(g-j)** Representative images (g,i) and percentage of tumour cell nuclei according to LAP1 intensity score (h,j) in the tumour body (TB) and invasive front (IF) of Day 55 WM983A (n = 8) and WM983B (n = 4) tumours (g,h) or in TB and IF of Day 33 A375P tumours (n = 8) and Day 33 A375M2 tumours (n = 8) grown subcutaneously in SCID mice. Significant p values shown for LAP1 score '3'. **(k)** Representative images and QuPath mark-up of GFP-NLS staining (map, red), **(l)** quantification of dermal invasion in Day 28 intradermal tumours (n= 10 in all conditions) and **(m)** *in-vitro* proliferation curves of A375P cells stably expressing GFP-NLS LAP1B-mRuby3, GFP-NLS LAP1B $^{1-72}$-mRuby3, GFP-NLS LAP1B$^{R563G}$-mRuby3, GFP-NLS LAP1C-mRuby3 and A375M2 GFP-NLS cells. n = 10 tumours per condition. In **d, f, l,** horizontal lines show the median and whiskers show minimum and maximum. P-values calculated by one-way ANOVA (f,l), two-way ANOVA (h,j), and unpaired t-test (d). **p<0.01, ***p<0.001, ****p<0.0001. Numerical data and exact p-values are available in the Source Data. In all cases, scale bars are 50 μm (10 μm in enlargements).

## Supplementary Material

Refer to Web version on PubMed Central for supplementary material.

## Acknowledgments

J.C.G. is a Wellcome Trust Senior Research Fellow (206346/Z/17/Z). V.S.-M. is a Cancer Research UK (CRUK) Senior fellow and the V.S.M. lab was supported by (CRUK) C33043/A24478 and Barts Charity. I.R.-H. was supported in the V.S.-M. lab by a Fundación Alfonso Martin Escudero fellowship and Marie Sklodowska-Curie Action, grant agreement No 659022. Y.J.-G. received a Crick-KCL PhD studentship. B.F. received a King's Health Partners PhD studentship. R.M.M was funded by ISCIII/FEDER "Una manera de hacer Europa" FIS-PI1500711 and PI18/00573; R.M.M and X.M.-G were funded by CIBERONC CB16/12/0023. We thank Dr. Eva Crosas-Molist and Dr. Jose L. Orgaz (Barts Cancer Institute, UK) for their supervision with cell biology experiments. We thank Professor Richard Marais (Cancer Research UK Manchester Institute) for the melanoma cells provided and Dr. Benilde Jiménez (Universidad Autónoma de Madrid and Instituto de Investigaciones Biomédicas CSIC-UAM, Spain) for the melanocytes. We thank Professor Kairbaan Hodivala-Dilke and Professor John Marshall (Queen Mary University of London) for help with animal experiments. This work was supported in part by the Francis Crick Institute which receives its core funding from Cancer Research UK (FC001002, FC001999), the UK Medical Research Council (FC001002, FC001999), and the Wellcome Trust (FC001002, FC001999). For the purpose of Open Access, the author has applied a CC BY public copyright licence to any Author Accepted Manuscript version arising from this submission.

## Data Availability

Gene expression data from cells were obtained from publicly available datasets and normalized as previously described [40, 57]. Data from four melanocyte datasets from (GSE4570), (GSE4840) (refs. [40,58,59) and data from melanoma cell lines (Philadelphia cohort GSE4841 (29 samples) and Mannheim cohort GSE4843 (37 samples)) were obtained from ref. [40]. Heat maps for gene expression in cells were generated using MeV_4_9_0 software (http://mev.tm4.org/). Gene expression data from human samples were derived from three publicly available datasets from ref. [42] (Riker GSE7553 (14 primary and 40 metastatic melanomas)) ref. [41], (Kabbarah GSE46517 (31 primary and 73 metastatic melanomas)) and ref. [43] (Xu GSE8401 (31 primary and 52 metastatic melanomas)). Source data are provided with this paper. All other data supporting the findings of this study

are available from the corresponding authors on reasonable request. Biological materials created in this study can be obtained from the corresponding authors on reasonable request.

## Code Availability

No code beyond freely available ImageJ tools was used in this study.

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

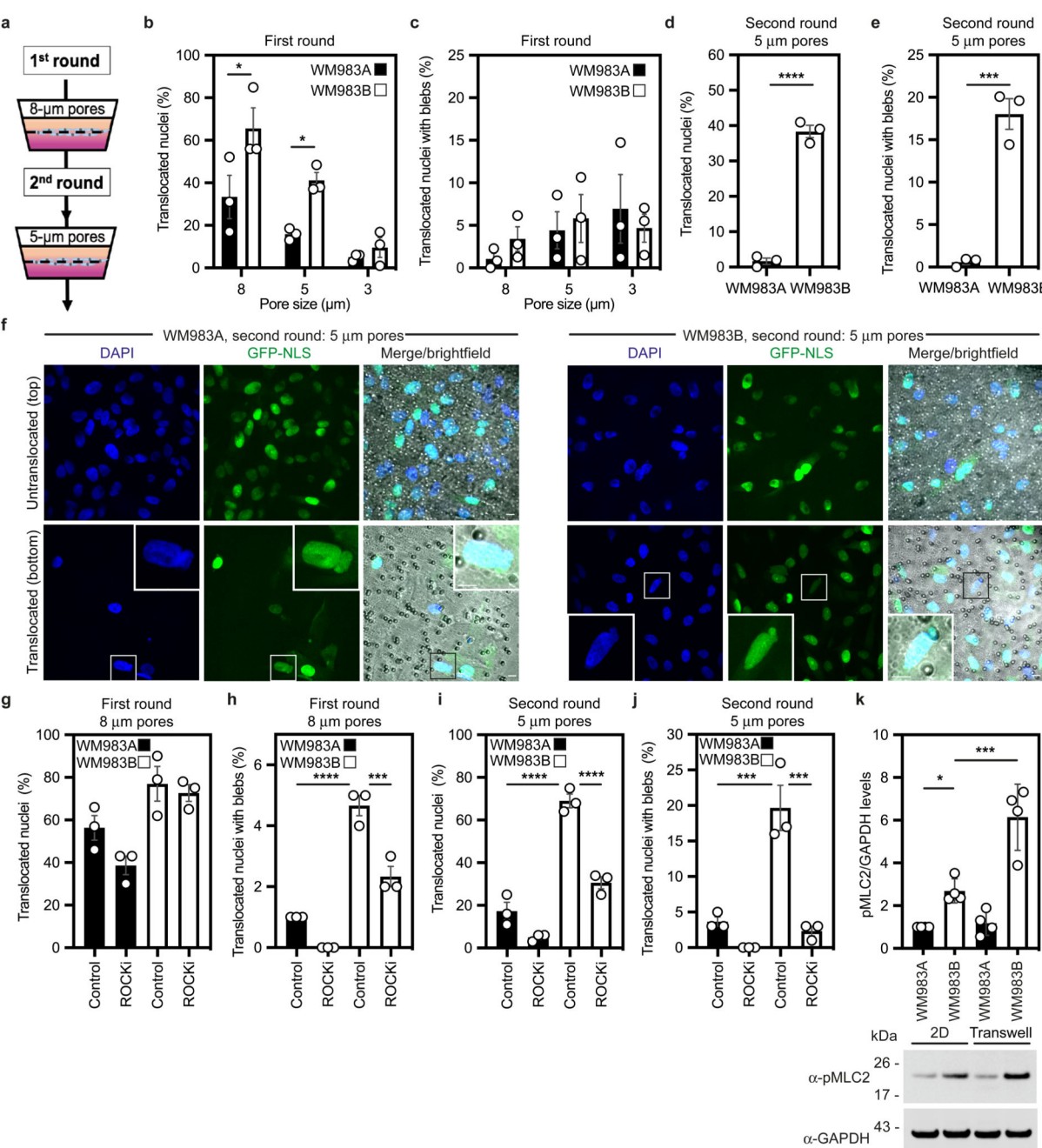

**Figure 1. Metastatic melanoma cells can negotiate repeated constraints**
(**a**) Schematic of transwell assays. Briefly, cells were challenged to migrate through transwells once or collected after a round of migration and challenged again. The arrows indicate the direction for chemotactic migration. (**b,c**) Percentage of primary melanoma WM983A cells and metastatic melanoma WM983B cells stably expressing GFP-NLS that translocated their nuclei (**b**) and displayed nuclear envelope blebs (**c**) after one round of migration in transwells of various pore sizes. n= 1758 and 1728 cells, respectively. (**d,e**) Percentage of WM983A and WM983B cells stably expressing GFP-NLS that translocated

their nuclei **(d)** and displayed nuclear envelope blebs **(e)** after a second round of transwell migration through 5-μm pores. n= 192 and 313 cells, respectively. **(f)** Representative images of WM983A and WM983B cells stably expressing GFP-NLS and stained for DNA after a second round of transwell migration through 5-μm pores. Transwell pores were visualised using transmitted light. Magnifications show nuclei that translocated and displayed nuclear envelope blebs indicated by white arrow heads. Scale bars, 30 μm and 10 μm magnifications. **(g,h)** Percentage of WM983A and WM983B cells stably expressing GFP-NLS that translocated their nuclei **(g)** and displayed nuclear envelope blebs **(h)** after one round of transwell migration through 8-μm pores upon ROCK inhibitor (ROCKi, GSK269962A) treatment. n= 1689 and 1757 cells, respectively. **(i,j)** Percentage of WM983A and WM983B cells stably expressing GFP-NLS that translocated their nuclei **(i)** and displayed nuclear envelope blebs **(j)** after a second round of transwell migration through 5-μm pores upon ROCKi treatment. n= 1119 and 1487 cells, respectively. **(k)** Resolved lysates from WM983A and WM983B cells grown in 2D or collected after overnight passage through 8 μm Transwell pores were examined by western blotting with anti-pMLC and anti-GAPDH antisera, and quantified by densitometry (above). Bar charts show the mean and error bars represent S.E.M. from N = 3 individual experiments. In k, error bars represent the S.D. from N = 4 independent experiments. P-values calculated by one-way ANOVA (g-k), two-way ANOVA (b,c), and two-tailed unpaired t-test (d,e); *p<0.05, ***p<0.001, ****p<0.0001. Numerical data and exact p-values are available in the Source Data.

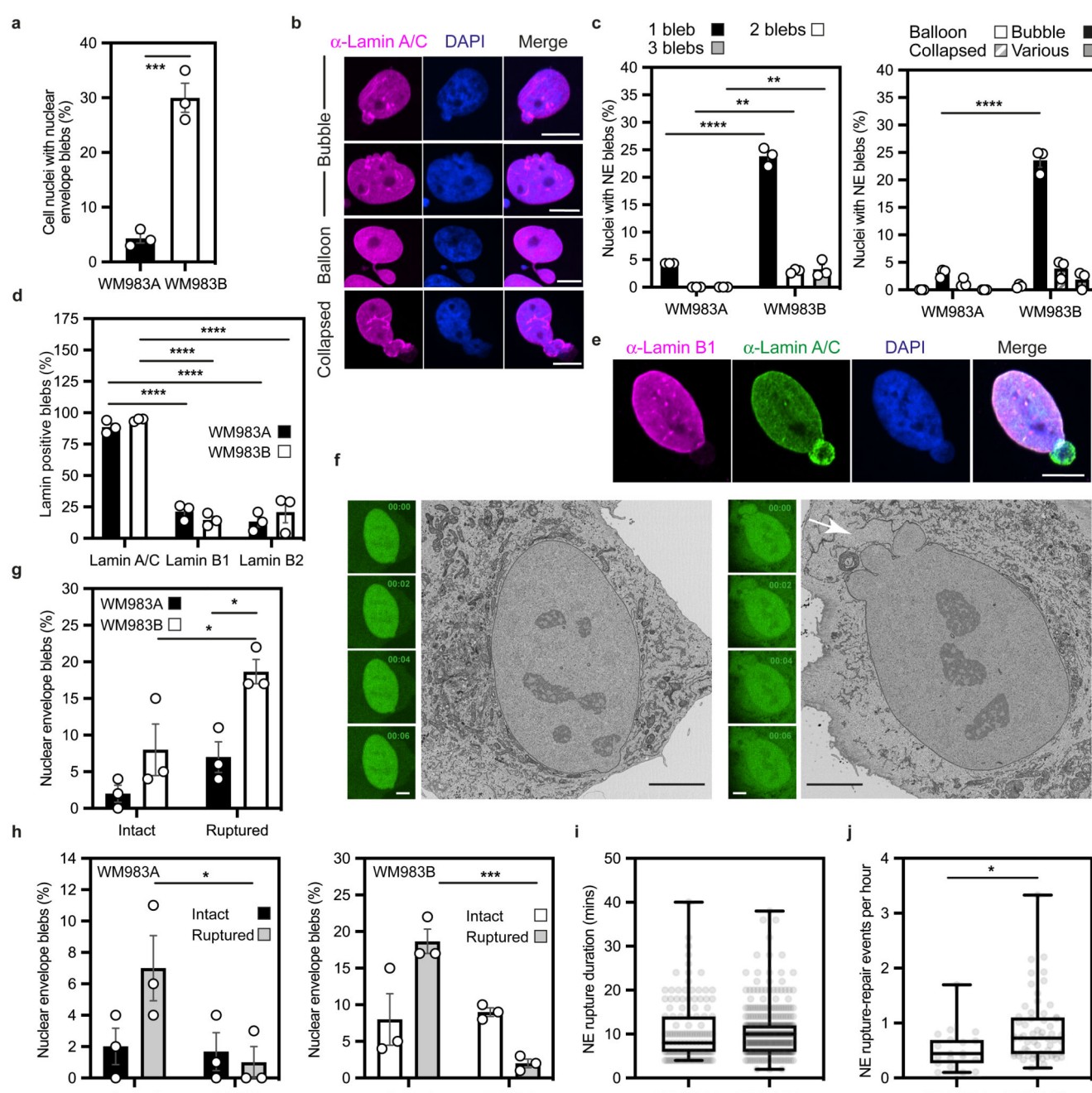

**Figure 2. The nuclear envelope of metastatic melanoma cells is highly dynamic**
(**a**) Percentage of primary melanoma WM983A and metastatic melanoma WM983B cells grown in 2D displaying nuclear envelope blebs. (**b**) Representative images of WM983B nuclei with nuclear envelope blebs of different shapes stained for Lamin A/C (magenta) and DNA (blue). Scale bars, 10 μm. (**c**) Percentage of nuclei with nuclear envelope blebs according to bleb number per nucleus and bleb shape in WM983A and WM983B cells. n= 434 and 561 cells, respectively. (**d**) Percentage of nuclear envelope blebs containing Lamin A/C, Lamin B1 or Lamin B2 in WM983A and WM983B cells. n= 605 and 510 cells,

respectively. **(e)** Representative images of a WM983B nucleus with a nuclear envelope bleb stained for Lamin A/C (green), Lamin B1 (magenta) and DNA (blue). Scale bar, 10 μm. **(f)** Representative live-cell image sequence of WM983B GFP-NLS nuclei with an intact nuclear envelope bleb (left) and a ruptured nuclear envelope bleb (right) with correlative SBF-SEM images of the same nuclei. The white arrow indicates the site of rupture of the nuclear membranes. Scale bar, 5 μm. **(g)** Percentage of intact and ruptured nuclear envelope blebs in WM983A and WM983B cells imaged over the course of 15 hours. n= 486 and 453 cells, respectively. **(h)** Percentage of intact and ruptured nuclear envelope blebs in WM983A (n= 486 and 522 cells, respectively) and WM983B (n= 453 and 560 cells, respectively) cells stably expressing GFP-NLS after treatment with GSK269962A and imaged over the course of 15 hours. **(i)** Duration of nuclear envelope rupture in WM983A and WM983B cells stably expressing GFP-NLS and imaged live (n.s. p=0.8389). **(j)** Nuclear envelope rupture events per hour in WM983A and WM983B stably expressing GFP-NLS and imaged live. n= 486 and 453 cells, respectively. In **a,c,d,g,h** bar charts show the mean and error bars represent S.E.M. from N = 3 independent experiments. In **i,j** horizontal lines show the median and whiskers show minimum and maximum range of values. p values calculated by two-way ANOVA (c,d,g,h), and two-tailed unpaired t-test (a,i,j); *p<0.05, **p<0.01, ***p<0.001, ****p<0.0001. Numerical data and exact p-values are available in the Source Data.

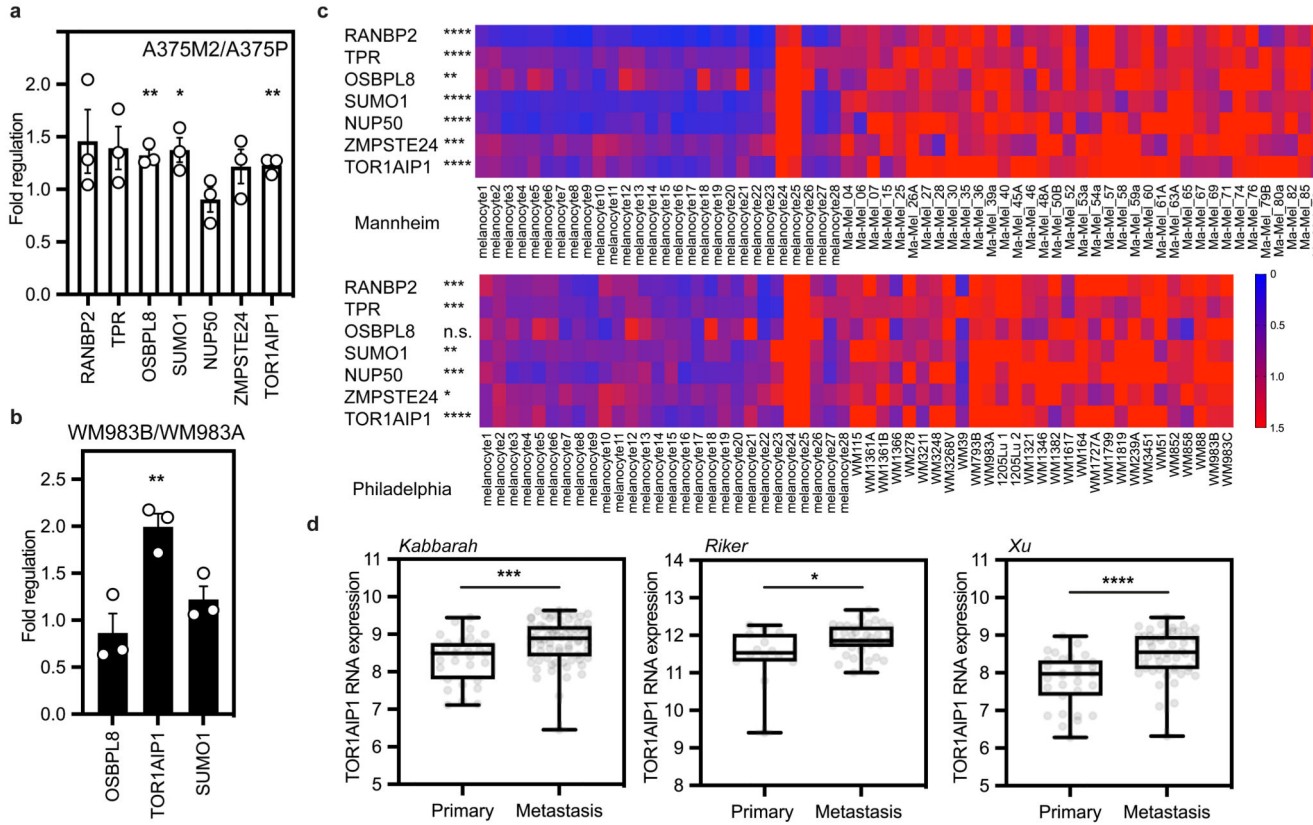

**Figure 3. *TOR1AIP1* is upregulated in metastatic melanoma cells**

**(a)** Fold regulation of candidate genes expression validated by RT-qPCR in A375M2 compared to A375P. **(b)** Fold regulation of *OSBPL8*, *TOR1AIP1* and *SUMO1* expression validated by RT-qPCR in metastatic melanoma WM983B cells compared to primary melanoma WM983A cells. **(c)** Heatmaps displaying fold change in expression of candidate genes in melanoma cell lines compared to melanocytes from Philadelphia and Mannheim datasets. **(d)** *TOR1AIP1* expression in primary tumours and metastasis in Kabbarah (n= 31 and 73, respectively), Riker (n= 14 and 40, respectively) and Xu (n= 31 and 52, respectively) melanoma patient datasets. Experimental data have been pooled from three individual experiments. In **a,b,** bar charts show the mean and error bars represent S.E.M. from N = 3 independent experiments. In **d** Horizontal lines show the median and whiskers show minimum and maximum range of values. P-values calculated by two-tailed unpaired t-test (a,b,c,d); *p<0.05, **p<0.01, ***p<0.001, ****p < 0.0001. Numerical data and exact p-values are available in the Source Data.

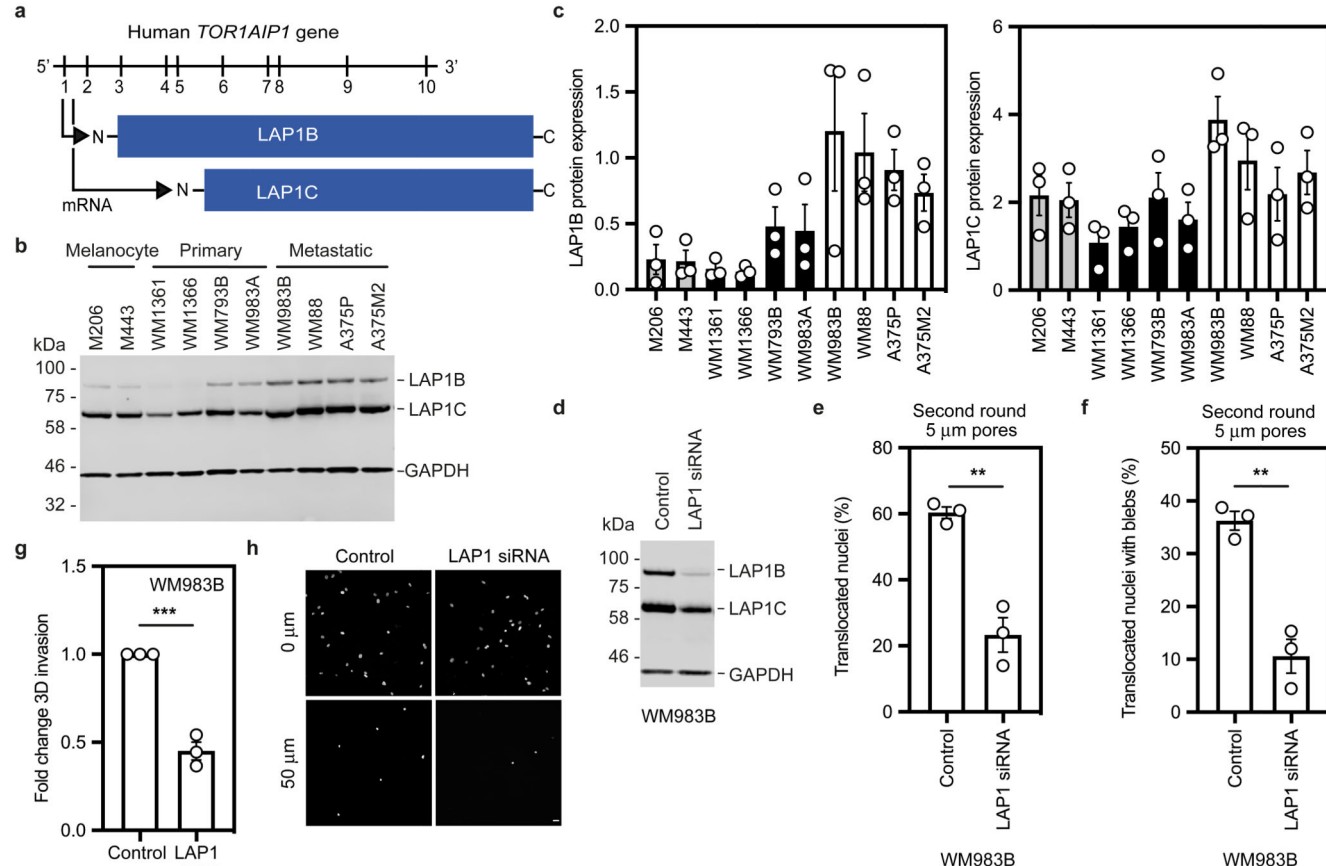

**Figure 4. LAP1 enables repeated constrained migration and invasion**

**(a)** Schematic of *TOR1AIP1* human gene transcription and translation. The gene is comprised of 10 exons and encodes a long protein isoform (584 aminoacids), LAP1B, and a short protein isoform (462 aminoacids), LAP1C, resulting from the use of an alternative translation initiation site at position 122. **(b)** Representative immunoblot of LAP1 expression levels in the indicated melanocytes, primary melanoma cells and metastatic melanoma cells. **(c)** Quantification of LAP1B and LAP1C protein expression in panel of cell lines in (b), normalised to GAPDH. **(d)** Representative immunoblot of LAP1 expression levels in WM983B cells upon transfection with pooled LAP1-targetting siRNA. **(e,f)** Percentage of WM983B cells that translocated their nuclei **(e)** and displayed nuclear envelope blebs **(f)** after a second round of transwell migration through 5-μm pores upon LAP1 depletion with siRNA pool. n= 872 and 705 cells, respectively. **(g,h)** 3D invasion index (number of invading cells at 50 μm / total number of cells) of WM983B cells in 3D collagen I matrices upon LAP1 depletion with siRNA pool. Representative images of WM983B cells stained for DNA at 0 μm and 50 μm into collagen upon LAP1 depletion with siRNA pool. Scale bars, 30 μm. n= 673 and 539 cells, respectively. Bar charts show the mean and error bars represent S.E.M. from N = 3 independent experiments. P-values calculated by two-tailed unpaired t-test (e,f,g); **p<0.01, ***p<0.001. Unprocessed western blots, numerical data and exact p-values are available in the Source Data.

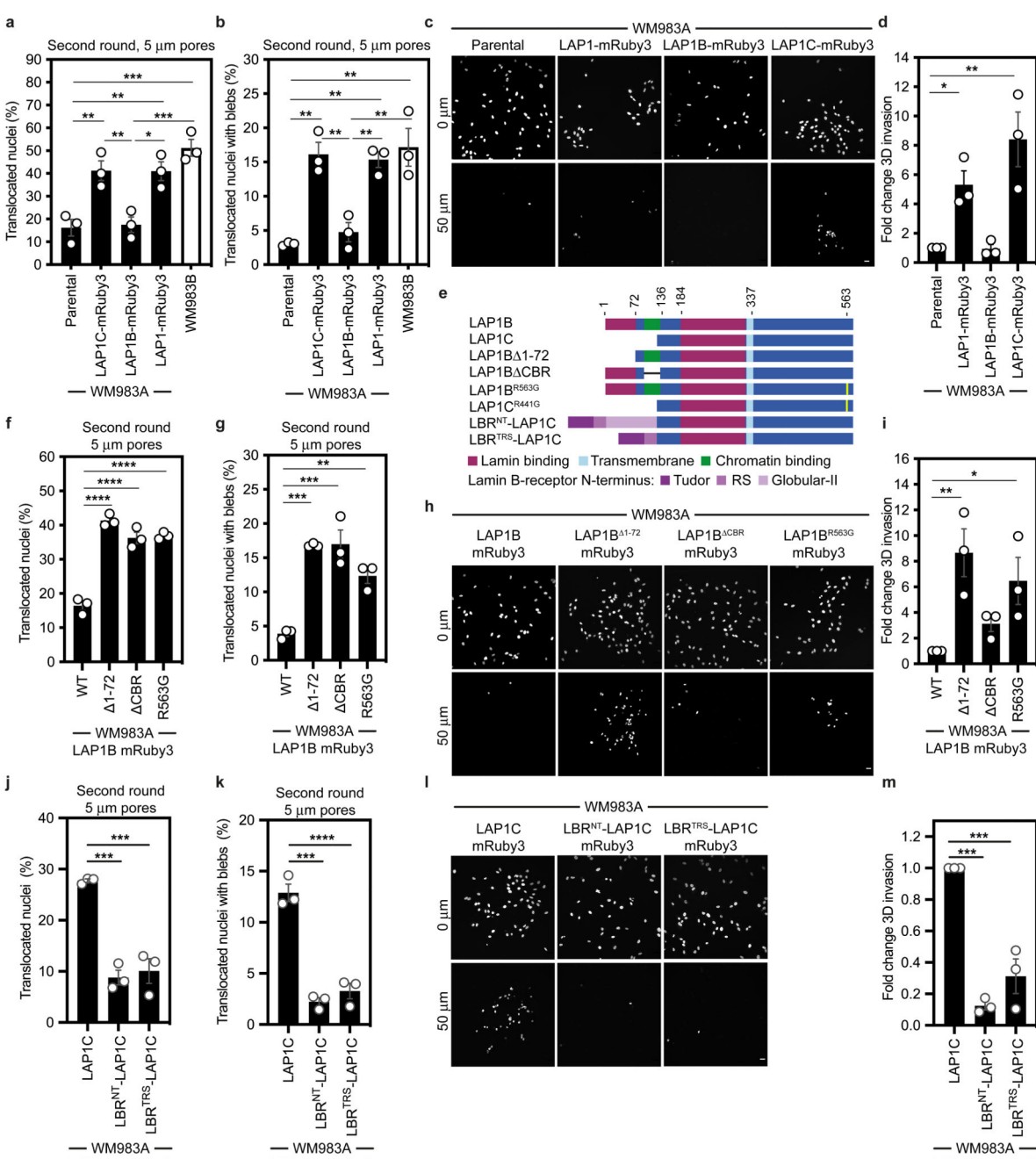

**Figure 5. LAP1C supports constrained migration and invasion**

**(a,b)** Percentage of primary melanoma WM983A cells stably expressing GFP-NLS alone or WM983A cells stably expressing GFP-NLS and LAP1C-mRuby3, LAP1B-mRuby3 or LAP1-mRuby3 and metastatic melanoma WM983B cells stably expressing GFP-NLS that translocated their nuclei **(a)** and displayed nuclear envelope blebs **(b)** after a second round of transwell migration through 5-μm pores. n= 668, 694, 592, 722, 633 cells, respectively. **(c,d)** Representative images **(c)** of WM983A GFP-NLS, WM983A GFP-NLS LAP1-mRuby3, WM983A GFP-NLS LAP1B-mRuby3 or WM983A GFP-NLS LAP1C-

mRuby3 cells invading into collagen I matrices and stained for DNA at Z=0 μm and Z=50 μm and **(d)** 3D invasion index from (c). Scale bars, 30 μm. n= 545, 609, 462, and 619 cells, respectively. **(e)** Schematic of LAP1B lacking the arginine finger (LAP1B$^{R563G}$), LAP1B lacking lamin-binding region 1-72 (LAP1B $^{1-72}$), LAP1B lacking its chromatin-binding region (LAP1B $^{CBR}$) and LAP1C and LBR-fusions to LAP1C's N-terminus. **(f,g)** Percentage of WM983A cells stably expressing GFP-NLS and LAP1B-mRuby3, LAP1B $^{1-72}$-mRuby3, LAP1B $^{CBR}$-mRuby3 or LAP1B$^{R563G}$-mRuby3 that translocated their nuclei **(f)** and displayed nuclear envelope blebs **(g)** after a second round of transwell migration through 5-μm pores. n= 518, 678, 545, 704 cells, respectively. **(h,i)** Representative images **(h)** of WM983A GFP-NLS LAP1B-mRuby3, WM983A GFP-NLS LAP1B $^{1-72}$-mRuby3, WM983A GFP-NLS LAP1B $^{CBR}$-mRuby3 and WM983A GFP-NLS LAP1B$^{R563G}$-mRuby3 cells invading into collagen I matrices and stained for DNA at Z=0 μm and Z=50 μm into collagen and **(i)** 3D invasion index for (h). Scale bars, 30 μm. n= 462. **(j,k)** Percentage of primary melanoma WM983A cells stably expressing GFP-NLS and LAP1C-mRuby3, LBR$^{NT}$-LAP1C-mRuby3 or LBR$^{TRS}$-LAP1C-mRuby3 that translocated their nuclei **(j)** and displayed nuclear envelope blebs **(k)** after a second round of transwell migration through 5-μm pores. n= 418, 402 and 385, respectively. **(l,m)** Representative images of WM983A GFP-NLS LAP1C-mRuby3, WM983A GFP-NLS LBR$^{NT}$-LAP1C-mRuby3 and WM983A GFP-NLS LBR$^{TRS}$-LAP1C-mRuby3 cells invading into collagen I matrices and stained for DNA at Z=0 μm and Z=50 μm into collagen and **(m)** 3D invasion index for (l), n= 918, 780 and 736, respectively. Scale bars, 30 μm. Bar charts show the mean and error bars represent S.E.M. from N = 3 independent experiements. P-values calculated by one-way ANOVA; *p<0.05, **p<0.01, ***p<0.001, ****p<0.0001. Numerical data and exact p-values are available in the Source Data.

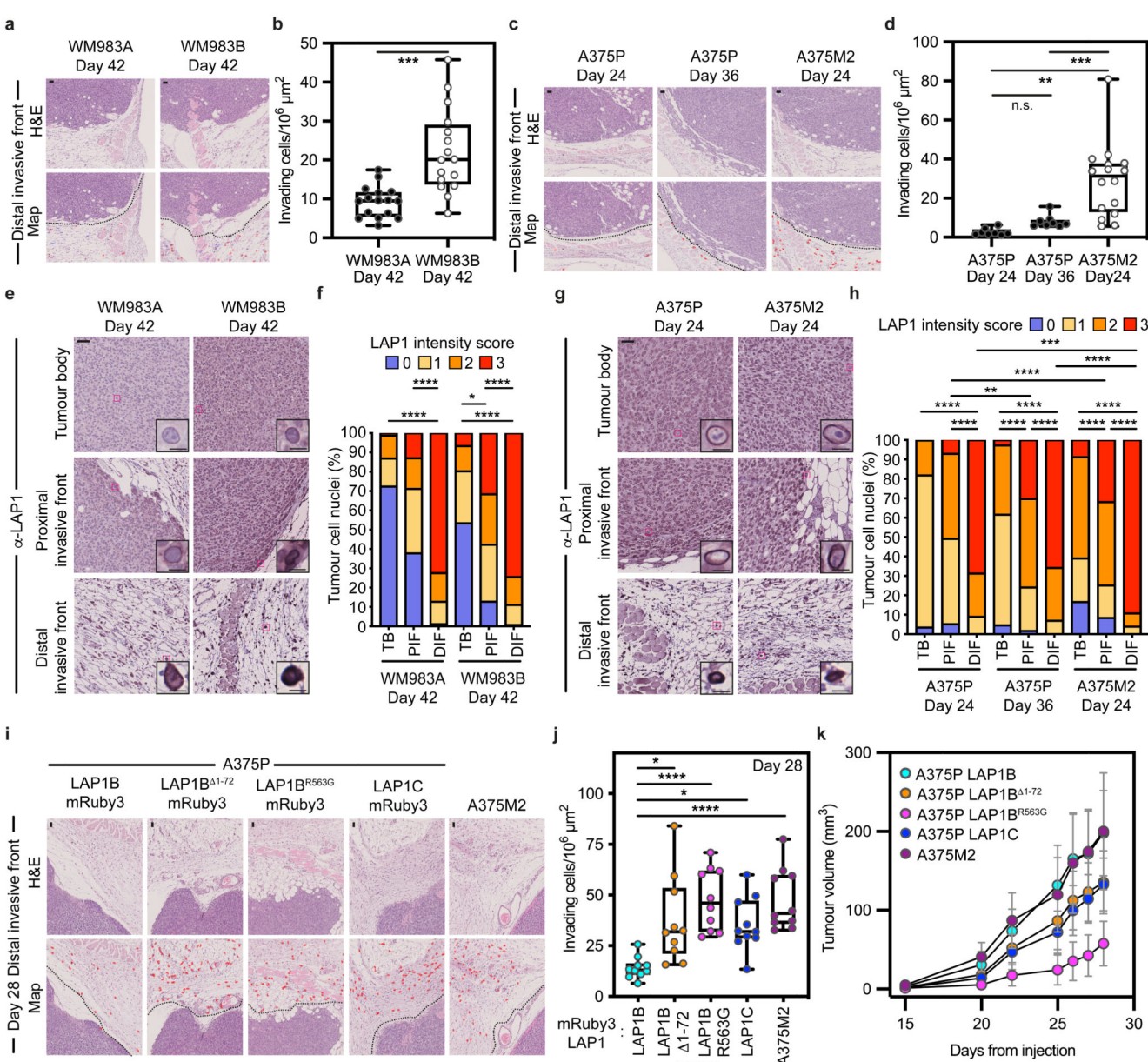

**Figure 6. LAP1C promotes invasion *in vivo***
**(a)** Representative images and QuPath mark-up (map) of hematoxilin (HE) staining and
**(b)** quantification of invading cells into the dermis from Day-42 WM983A and WM983B
tumours grown in NXG mice. Dashed lines represent the boundary between proximal
invasive front (PIF) and the distal invasive front (DIF). HE-positive tumour cells are in
red in the map. n= 16 and 16 tumours, respectively. **(c)** Representative images and QuPath
mark-up of HE staining and **(d)** quantification of invading cells into the dermis from Day 24
and Day 36 A375P tumours and Day 24 A375M2 tumours grown in NXG mice. n= 8, 8 and
16 tumours, respectively. **(e)** Representative images and **(f)** percentage of tumour cell nuclei
according to LAP1 intensity score in TB, PIF and DIF of Day 42 WM983A and WM983B
tumours. n= 16 and 15 tumours, respectively. **(g)** Representative images and **(h)** percentage

of tumour cell nuclei according to LAP1 intensity score in TB, PIF and DIF of Day 24 and Day 36 A375P tumours and Day 24 A375M2 tumours. n= 8, 8 and 16 tumours, respectively. **(i)** Representative and QuPath mark-up of HE staining, **(j)** quantification of invading cells into the dermis and **(k)** tumour growth curves of A375P GFP-NLS LAP1B-mRuby3, A375P GFP-NLS LAP1B $^{1-72}$-mRuby3, A375P GFP-NLS LAP1B$^{R563G}$-mRuby3, A375P GFP-NLS LAP1C-mRuby3 and A375M2 GFP-NLS tumours grown intradermally in NXG mice. n= 10 tumours per condition. In IHC panels, scale bars are 50 μm. In **b,d,j** horizontal lines show the median and whiskers show minimum and maximum range of values. In **k,** graphs show the mean and error bars represent S.D. P-values calculated by one-way ANOVA (d,j), two-way ANOVA (f,h) and two-tailed unpaired t-test (b). In f and h, significant p values shown are for LAP1 score 3. *p<0.05, **p<0.01, ***p<0.001, ****p<0.0001. Numerical data and exact p-values are available in the Source Data.

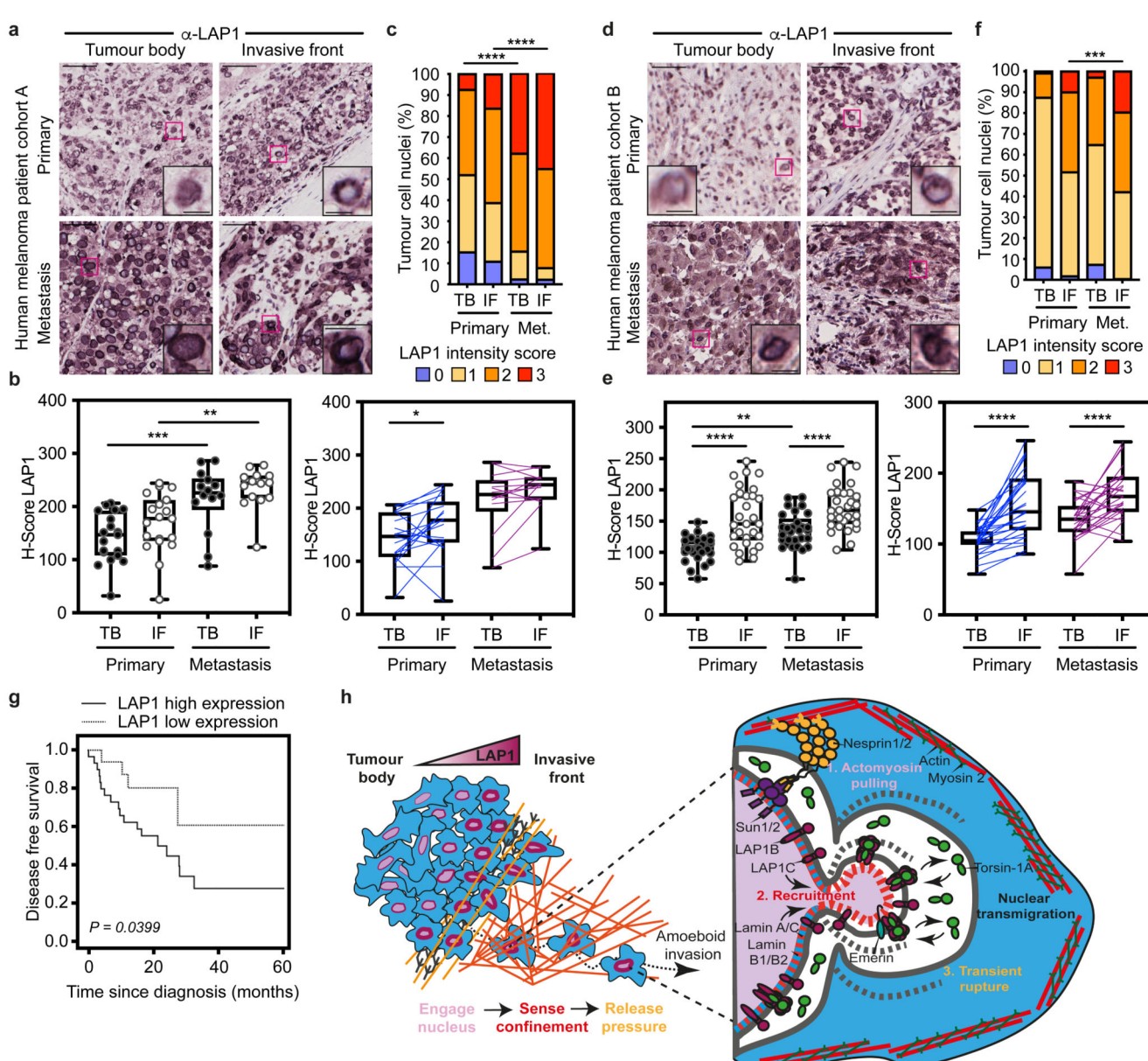

**Figure 7. LAP1 levels increase in human melanoma progression**

**(a)** Representative images of tumour body (TB) and invasive front (IF) of a primary tumour and a metastasis in human melanoma patient cohort A. Scale bars, 50 μm. The magnifications show representative cell nuclei. Scale bars, 10 μm. **(b)** H-score for LAP1 staining in TB and IF of primary tumours and metastases in cohort A by unpaired (left) or paired (right) analysis. **(c)** Percentage of tumour cell nuclei according to LAP1 intensity score in TB and IF of primary tumours and metastases in patient cohort A. Significant p values shown are for LAP1 score 3. n= 19 primary tumours and 14 metastases. **(d)** Representative images of TB and IF of a primary tumour and a metastasis in human melanoma patient cohort B. Scale bars, 50 μm. The magnifications show representative cell nuclei. Scale bars, 10 μm. **(e)** H-score for LAP1 staining in TB and IF of primary

tumours and metastases in patient cohort B by unpaired (left) or paired (right) analysis. **(f)** Percentage of tumour cell nuclei according to LAP1 intensity score in TB and IF of primary tumours and metastases in patient cohort B. Significant p values shown are for LAP1 score 3. n= 29 primary tumours and 29 metastases. **(g)** Kaplan–Meier survival curve of disease-free survival according to LAP1 expression in the IF from primary melanomas. LAP1 expression was categorized as low or high using the mean expression. n= 46 primary melanomas. **(h)** Summary model. LAP1 levels are elevated at the invasive front of melanoma tumours, which are enriched in cells displaying amoeboid features like a rounded cell morphology, high levels of Myosin II and nuclear envelope blebs. LAP1C, but not LAP1B, can localise to nuclear envelope blebs in a Lamin A/C dependent manner and promotes transit through physical constraints through its weaker N-terminal NE/lamina tethering allowing NE blebbing. In **b,e** Horizontal lines show the median and whiskers show minimum and maximum range of values. p values calculated by one-way ANOVA (b,e), two-way ANOVA (c,f), two-tailed unpaired and paired t-test (b,e) and (g) log-rank test; *p<0.05, **p<0.01, ***p<0.001, ****p<0.0001. Numerical data and exact p-values are available in the Source Data.

