## [Peer Review File · Nature cell biology]

Peer Review Information

Journal: Nature Cell Biology

Manuscript Title: LAP1 supports nuclear adaptability during constrained melanoma cell migration and invasion

Corresponding author name(s): Dr Jeremy Carlton

Editorial Notes:

Reviewer Comments & Decisions:

Decision Letter, initial version:
--

Dear Dr Carlton,

Thank you for submitting your manuscript, "LAP1 supports nuclear plasticity during constrained migration", to Nature Cell Biology. It has now been seen by 3 referees, who are experts in nuclear envelope (NE) biology (referee 1); NE, constrained migration in cancer (referee 2); and constrained migration in cancer, metastasis (referee 3). As you will see from their comments (attached below), they find this work of potential interest, but have raised substantial concerns, which in our view would need to be addressed with considerable revisions before we can consider publication in Nature Cell Biology.

As you may know, Nature Cell Biology editors discuss the referee reports in detail within the editorial team, including the chief editor, to identify key referee points that should be addressed with priority to

strengthen the core advance and central conclusions, as opposed to requests that are being beyond the scope of the current study. To guide the scope of the revisions, I have listed these points below. We are committed to providing a fair and constructive peer-review process, so please feel free to contact me if you would like to discuss any of the referee comments further. Our typical revision period is six months; however, please feel free to get in touch with me should you have any questions or anticipate delays or issues addressing the reviews.

In our view, for reconsideration at NCB, it would be essential to:

-- extend the mechanistic understanding of LAP1's contribution to NE blebbing, cell migration, and metastasis, with stronger links between these elements. All the reviewers commented on this aspect and we agree that this is an aspect of high interest to cell biologists that is key to the suitability of the work for NCB. The reviewers' comments also echo our own editorial discussions about the study. For instance, Rev#1 suggested testing the involvement of Torsins, which we agree would be an interesting and logical next step. The referees asked for further characterizations of the blebs, measurements of nuclear stiffness, NE integrity, and experiments establishing whether LAP1's effects on migration depend on nuclear blebbing in appropriate in vitro models.

-- enhance the understanding of the contributions of LAP1 and nuclear blebbing to metastasis, in vitro and in vivo

-- All other referee concerns pertaining to strengthening existing data, technical comments, providing controls, methodological details, clarifications and textual changes, should also be addressed.

-- Finally please pay close attention to our guidelines on statistical and methodological reporting (listed below) as failure to do so may delay the reconsideration of the revised manuscript. In particular please provide:

We would be happy to consider a revised manuscript that would satisfactorily address these points, unless a similar paper is published elsewhere, or is accepted for publication in Nature Cell Biology in the meantime.

- ensure that it conforms to our format instructions and publication policies (see below and <https://www.nature.com/nature/for-authors>).

- provide a point-by-point rebuttal to the full referee reports verbatim, as provided at the end of this letter.

- provide the completed Reporting Summary (found here <https://www.nature.com/documents/nr-reporting-summary.pdf>). This is essential for reconsideration of the manuscript will be available to editors and referees in the event of peer review. For more information see <http://www.nature.com/authors/policies/availability.html> or contact me.

When submitting the revised version of your manuscript, please pay close attention to our [Digital Image Integrity Guidelines](https://www.nature.com/nature-research/editorial-policies/image-integrity). and to the following points below:

This journal strongly supports public availability of data. Please place the data used in your paper into a public data repository, or alternatively, present the data as Supplementary Information. If data can only be shared on request, please explain why in your Data Availability Statement, and also in the correspondence with your editor. Please note that for some data types, deposition in a public repository is mandatory - more information on our data deposition policies and available repositories appears below.

[Redacted]

We hope that you will find our referees' comments and editorial guidance helpful. Please do not hesitate to contact me if there is anything you would like to discuss. Thank you again for considering NCB for your work.

Best wishes,

Melina

Melina Casadio, PhD
Senior Editor, Nature Cell Biology
ORCID ID: <https://orcid.org/0000-0003-2389-2243>

Reviewers' Comments:

Reviewer #1:

Remarks to the Author:

This is an interesting and well conducted study by Jung-Garcia et al exploring a role for LAP1 in modulating nuclear plasticity in a cancer context. Although it is well established that the nucleus undergoes major deformation, blebbing, and rupture, in the context of migration through constrictions (and much of the early data in the manuscript are largely confirmatory of these studies), there is still much to be understood regarding the underlying molecular mechanisms that control whether cells are more or less able to navigate these constrictions without losses in viability. There are several important advances in this work. The first is the explicit, and well-considered, link to melanoma with the potential of LAP1 levels being a cancer biomarker. The authors take advantage of melanoma cells derived from primary and metastatic lesions as a starting point to their experimental strategy. They demonstrate that the metastatic cells more frequently pass through membranes with narrow pores and then nicely tie this ability to the overexpression of the integral INM protein, LAP1. Unlike many other cancer cell studies that cherry-pick one overexpressed gene of hundreds, there is a clear demonstration that LAP1 is a standout among many genes upregulated and in fact direct staining of primary tumors reveals that the protein is found at high levels and specifically at the migratory edge. The authors also provide interesting genetic and in vivo physical interaction data that suggests a specific isoform of LAP1, LAP1C, which does not interact with the lamina or chromatin is more mobile, populates, and may drive, nuclear blebs and in fact acts in a dominant fashion to promote nuclear migration through constrictions in the non-metastatic cell type. Thus, the paper provides new mechanistic insight into nuclear plasticity in an important physiological (and pathological) context and thus the paper should be of broad appeal and appropriate for NCB pending some needed additions:

1) There was a justified focus on exploring the connections between LAP1 and nuclear factors, however, it is also well established that LAP1 binds to the AAA-ATPase Torsin through its luminal domain where it helps activate the ATPase in ways that have been linked to disrupting chromatin-LAP1 interactions (Luithle et al, 2020) and inducing nuclear envelope blebs (Laudermilch et al., 2016). Further, there is evidence that the LAP1-Torsin interaction contributes to cell migration by modulating LINC complex function (see Saunders et al., JCB, 2017). Thus, the relationship (or lack thereof) between LAP1 and Torsin in this study should be at least preliminarily investigated to more explicitly connect this work with more established roles of LAP1 in nuclear envelope biology. The suggestion is to overexpress a LAP1 mutant that cannot bind to Torsin and explore its impact on nuclear envelope blebbing and migration through constrictions.

2) The in vivo interaction data using the mitochondrial anchoring in Figure 6 is exceedingly difficult to interpret. First, a diagram of LAP1 with the "unique region" clearly labeled would be helpful. Second, there is the interpretation that relocalization of mitochondria to the nuclear envelope is suggestive of physical interactions between LAP1 fragments and lamins/chromatin. But as lamins/chromatin are inside the nucleus, there is a topological problem here unless the suggestion is that the mitochondria enter the nucleus? As this is highly unlikely, the interpretation of these experiments as reflecting physiological interactions is questionable. An orthogonal approach or a more explicit

description/interpretation of how the observed interactions occur would be helpful.

3) There is already published evidence that actomyosin contractility drives NE blebbing and nuclear ruptures. The Hatch and Hetzer (2016) article should be referenced and discussed in the context of ROCK inhibition.

4) Figure 5: A specificity control of knocking down another integral INM protein would strengthen this data. e.g. emerin or MAN1.

5) The use of the first and second round filters is useful but it is often challenging to understand whether or not the first round explicitly changes the cells (e.g. deforms the nucleus) and/or selects for a more migratory population for the second round. I don't have any explicit experimental suggestions but think that the reasoning and interpretation behind the two rounds of translocation could be better articulated.

6) Figure 7F: In some of the images it appears as if there is nuclear fluorescence of the Ruby construct raising the possibility that these constructs may be unstable. I don't think this changes the outcome of the experiment but an explanation for this result is necessary and could include a western blot to ensure the stability of the constructs.

Reviewer #2:

Remarks to the Author:

In this article, Jung-Garcia and colleagues investigate the difference between two cell lines derived from a melanoma from the same patient, one from the primary tumor, the other from a metastatic lesion. Using a sophisticated transwell assay in which the cells are recovered after a first passage then assayed for a second, they find that the cell line from the metastatic lesion is more effective in passing already at the first round, but even more at the second, and that this was associated with a strong phenotype of formation of blebs on the cells nucleus.

They found that nuclear blebs were present in a large number of nuclei on this cell line even when they had not passed through the transwells. Using published expression databases (on other melanoma cell lines), and further experiments and analysis, they identify and validate three genes upregulated in the more metastatic cell line and find that one of them, encoding for LAP1, is also upregulated in the more metastatic melanoma cell line derived from the patient. They then focus on this protein and find that it seems in general to be more expressed in invasive tumor front and that a higher expression correlates with a poor prognosis. A deeper investigation of the molecular and cell biology of this protein reveals that it has two isoforms with different binding to the nuclear lamina and chromatin and that both are upregulated in the more metastatic melanoma cells. Reduction of the level of expression of these isoforms in the metastatic cell line reduced its transwell migration capacity. The two isoforms showed different binding to Lamin A/C and Lamin B and consistently, different enrichment in the bleb membrane. Investigating the specific role of the two isoforms in promoting transwell migration, they find that the short isoform (which does not bind Lamin B) has a strong, dose-dependent effect on promoting nuclear blebbing and transwell migration when over-expressed, which the long isoform does not have. Over-expression of a shorter version of the long isoform, lacking lamin binding or chromatin binding domains, has the same effect as the short isoform in promoting nuclear blebbing and transwell migration.

It is hard to summarize much more, since the authors do not really provide a synthetic working model for their observations, which is a major concern.

Overall, the article contains a large number of very interesting experiments pointing to a role of LAP1 in modulating the capacity of cells to pass through small pores, with a potential impact on metastasis in melanoma cells. The authors also suggest that this is because this protein facilitates the formation

of nuclear blebs. The article is very interesting and the experiments convincing. I would recommend asking the authors for submission of a revised article with the following points clarified:

Major concerns:

The authors establish many correlative evidences but very few proof of causality, at various levels of the article.

The main aspects to investigate in more details are:

- a) whether nuclear blebs indeed help the passage of nuclei and thus of cells through small pores. Would there be another way to promote nuclear blebs, which would for example be independent of an increase in Myosin II activity, as it is a confounding factor here? For example by weakening the binding of the nuclear membranes to the lamina by depleting some specific proteins?
- b) how Lap1 short isoform promotes nuclear blebbing – and whether it is by promoting nuclear blebbing that it also promotes transwell migration.
- c) clarifying the function of Lap1/nuclear blebbing in the metastatic potential of melanoma (or other) cells – is there increased collagen matrix invasion (in an in vitro assay)? Or is it another step of the metastatic spread which is promoted by expression of Lap1 (extravasation, crossing basement membranes, ...?). This might go a bit too far for the scope of the article, but at least a discussion of this last point would be important.

With these important points clarified, the authors might be able to propose a more precise working model for how increased expression of Lap1 might promote metastasis, which is totally lacking at the moment. In their discussion they do not even discuss any sort of scenario – for example, is it possible that increased expression of the short Lap1 isoform, because it does not bind as strongly to lamina or chromatin, is weakening the binding of the nuclear inner membrane to the lamina or the chromatin, for example by displacing another protein, and thus makes it easier for the membrane to dissociate from the lamina and form a bleb?

Then once a bleb is formed, is the nucleus easier to deform, explaining why it passes more easily through the transwell. It is also not clear how to explain, from the data shown, how Lap1 increased expression in the more metastatic cell line derived from the patient, promotes the passage through the second transwell assay, while the less metastatic cells have a reduced rate of passage the second time (why this reduced rate of passage?).

There is a real need to provide more mechanistic explanations for the phenomena described in the article, in order to get to a working model that explains the observations. This would also strengthen the proof of the causal relationships suggested by the authors.

A minor concern, which might also help clarify some aspects: when the authors describe nuclear blebbing, they always seem to observe chromatin herniation inside the bleb. In the literature, it seems that there are at least two types of nuclear blebs: blebs that are just a separation of the nuclear membranes from the lamina, but the lamina is intact and there is no chromatin in these blebs, just nucleoplasm, and blebs that combine detachment of the membrane from the lamina and a rupture in the lamina, which allows chromatin to flow into the bleb. Depending on the type of effect that Lap1 short isoform has on perturbing the nuclear envelope, one or the other type of bleb might be more favored.

The blebs without chromatin inside should be easily visualized using a combination of NLS-GFP and a chromatin marker. They should be positive for GFP but not for the chromatin marker.

Reviewer #3:

Remarks to the Author:

In this manuscript, Jung-Garcia et al. determined that LAP1 is overexpressed in metastatic melanoma cells. Importantly, the authors investigated the role of two LAP1 isoforms (LAP1B and LAP1C) in melanoma cell migration through constricted pores. They determined that ectopic expression of LAP1C in WM983A melanoma cells derived from a primary tumor increases both nuclear envelope (NE) blebbing and migration efficiency. Intriguingly, ectopic expression of LAP1B lacking either the dominant lamin-binding or the CBR, but not full-length LAP1B, in WM983A melanoma cells also enhanced both NE blebbing and migration. Overall, these are interesting and novel observations. Unfortunately, several concerns and limitations have dampened the reviewer's enthusiasm for this manuscript as outlined below.

Concerns:

1. This is a rather phenomenological study as a mechanism of action for LAP1B and LAP1C isoforms is missing. For instance, it is unclear how LAP1C overexpression contributes to increased NE blebbing and transwell migration. Does LAP1C co-localization with lamin A/C affect the integrity of the nuclear envelope? Is the envelope more pliable with increased expression? What happens when the interaction between LAP1C and lamin A/C is disrupted? How do these results relate to the effect of actomyosin contractility described in Figure 1?
2. To prove that LAP1 isoforms confer nuclear plasticity, physical measurements (e.g., nuclear stiffness) are needed.
3. Extending the aforementioned interesting observations from the in vitro to the in vivo setting would greatly enhance the impact of this manuscript. Importantly, these studies would help differentiate if LAP1 is an important player in metastasis or a biomarker of metastatic cells.
4. The authors should employ alternative in vitro models such as 3D collagen gels.
5. LAP1C overexpression enhances the transwell migration of WM983A cells. Is the same true for A375P, other melanoma cells, and melanocytes? How do LAP1B mutants affect the transwell migration of these additional cell lines? In extension, does the localization of LAP1B change in response to $\Delta 1-72$ and ΔCBR mutations? How is 3D cell migration through confining pores affected in response to these mutants or LAP1C overexpression?
6. Some of the authors remarks are not supported by their data. For instance:
 - a. Lines 98-99: "metastatic melanoma WM983B cells were more effective at negotiating constraints than primary melanoma WM983A cells (Fig.1b). According to Fig. 1b, this is correct only for the pore size of 8 μm . There is no ss for the pore size of 5 μm and no difference for 3 μm . Along these lines, their next statement is also not supported by their data (Fig. 1c).
 - b. Lines: 343-344: "the effect of LAP1C promoting NE blebbing and constrained migration was concentration dependent (Fig.7c-g)". There is no difference between medium and high levels.
7. Lines 130-135: "ROCK1/2 inhibition did not reduce nuclear translocation but did reduce NE blebbing of WM983B cells during the first round of migration (Fig.1g, h). However, ROCK1/2 inhibition markedly impaired nuclear translocation and reduced NE blebbing after pore transit during the second round

(Fig. 1i, j), suggesting that passage through the first constraint activates a Rho-ROCK1/2-dependent migration programme for subsequent passages". The authors' statement is not necessarily correct. According to the authors, the second passage occurred through pores of 5 μm , whereas the first through 8 μm . The reviewer believes that 8 μm pores do not constitute a confining microenvironment, and that is why ROCK1/2 inhibition has little or no effect. The authors' statement will be supported only if they carry out both the first and second passages through 8 μm .

8. The authors should cite a relevant article with PMID: 31690619

9. There are several typos (e.g., the y-axis of Supplementary Fig. 1a,b, d, e,h j).

Methods should be written concisely, but should contain all elements necessary to allow interpretation and replication of the results. As a guideline, Methods sections typically do not exceed 3,000 words. The Methods should be divided into subsections listing reagents and techniques. When citing previous methods, accurate references should be provided and any alterations should be noted. Information must be provided about: antibody dilutions, company names, catalogue numbers and clone numbers for monoclonal antibodies; sequences of RNAi and cDNA probes/primers or company names and catalogue numbers if reagents are commercial; cell line names, sources and information on cell line identity and authentication. Animal studies and experiments involving human subjects must be reported in detail, identifying the committees approving the protocols. For studies involving human subjects/samples, a statement must be included confirming that informed consent was obtained. Statistical analyses and information on the reproducibility of experimental results should be provided in a section titled "Statistics and Reproducibility".

All Nature Cell Biology manuscripts submitted on or after March 21 2016 must include a Data availability statement as a separate section after Methods but before references, under the heading "Data Availability". For Springer Nature policies on data availability see <http://www.nature.com/authors/policies/availability.html>; for more information on this particular policy see <http://www.nature.com/authors/policies/data/data-availability-statements-data->

citations.pdf. The Data availability statement should include:

- Accession codes for primary datasets (generated during the study under consideration and designated as "primary accessions") and secondary datasets (published datasets reanalysed during the study under consideration, designated as "referenced accessions"). For primary accessions data should be made public to coincide with publication of the manuscript. A list of data types for which submission to community-endorsed public repositories is mandated (including sequence, structure, microarray, deep sequencing data) can be found here <http://www.nature.com/authors/policies/availability.html#data>.
- Unique identifiers (accession codes, DOIs or other unique persistent identifier) and hyperlinks for datasets deposited in an approved repository, but for which data deposition is not mandated (see here for details <http://www.nature.com/sdata/data-policies/repositories>).
- At a minimum, please include a statement confirming that all relevant data are available from the authors, and/or are included with the manuscript (e.g. as source data or supplementary information), listing which data are included (e.g. by figure panels and data types) and mentioning any restrictions on availability.
- If a dataset has a Digital Object Identifier (DOI) as its unique identifier, we strongly encourage including this in the Reference list and citing the dataset in the Methods.

We recommend that you upload the step-by-step protocols used in this manuscript to the Protocol Exchange. More details can found at www.nature.com/protocolexchange/about.

All imaging data should be accompanied by scale bars, which should be defined in the legend. Cropped images of gels/blots are acceptable, but need to be accompanied by size markers, and to retain visible background signal within the linear range (i.e. should not be saturated). The boundaries of panels with low background have to be demarked with black lines. Splicing of panels should only be considered if unavoidable, and must be clearly marked on the figure, and noted in the legend with a statement on whether the samples were obtained and processed simultaneously. Quantitative comparisons between samples on different gels/blots are discouraged; if this is unavoidable, it should only be performed for samples derived from the same experiment with gels/blots were processed in parallel, which needs to be stated in the legend.

Figures should be provided at approximately the size that they are to be printed at (single column is 86 mm, double column is 170 mm) and should not exceed an A4 page (8.5 x 11"). Reduction to the scale that will be used on the page is not necessary, but multi-panel figures should be sized so that

the whole figure can be reduced by the same amount at the smallest size at which essential details in each panel are visible. In the interest of our colour-blind readers we ask that you avoid using red and green for contrast in figures. Replacing red with magenta and green with turquoise are two possible colour-safe alternatives. Lines with widths of less than 1 point should be avoided. Sans serif typefaces, such as Helvetica (preferred) or Arial should be used. All text that forms part of a figure should be rewritable and removable.

The total number of Supplementary Figures (not including the “unprocessed scans” Supplementary Figure) should not exceed the number of main display items (figures and/or tables (see our Guide to Authors and March 2012 editorial <http://www.nature.com/ncb/authors/submit/index.html#suppinfo>; <http://www.nature.com/ncb/journal/v14/n3/index.html#ed>). No restrictions apply to Supplementary Tables or Videos, but we advise authors to be selective in including supplemental data.

GUIDELINES FOR EXPERIMENTAL AND STATISTICAL REPORTING

REPORTING REQUIREMENTS – We are trying to improve the quality of methods and statistics reporting in our papers. To that end, we are now asking authors to complete a reporting summary that collects information on experimental design and reagents. The Reporting Summary can be found here <https://www.nature.com/documents/nr-reporting-summary.pdf> If you would like to reference the guidance text as you complete the template, please access these flattened versions at <http://www.nature.com/authors/policies/availability.html>.

STATISTICS – Wherever statistics have been derived the legend needs to provide the n number (i.e. the sample size used to derive statistics) as a precise value (not a range), and define what this value represents. Error bars need to be defined in the legends (e.g. SD, SEM) together with a measure of centre (e.g. mean, median). Box plots need to be defined in terms of minima, maxima, centre, and

percentiles. Ranges are more appropriate than standard errors for small data sets. Wherever statistical significance has been derived, precise p values need to be provided and the statistical test used needs to be stated in the legend. Statistics such as error bars must not be derived from $n < 3$. For sample sizes of $n < 5$ please plot the individual data points rather than providing bar graphs. Deriving statistics from technical replicate samples, rather than biological replicates is strongly discouraged. Wherever statistical significance has been derived, precise p values need to be provided and the statistical test stated in the legend.

Author Rebuttal to Initial comments

Revision of NCB-C46392, Jung-Garcia et al., now retitled: *LAP1 supports nuclear adaptability during constrained migration and invasion*

We thank all the editors and reviewers for assessing our manuscript and for providing positive and useful comments. We addressed all issues raised through new experimental work and textual changes, and believe genuinely that they have improved our manuscript. In the following response, we supply the original reviewers' comments in roman type, with our response underneath each point in italic type. As well as indicating where textual and experimental additions have been incorporated into the text, we supply a series of figures for the reviewers to illustrate points that support our points, but that were beyond what was possible to incorporate into the manuscript. Our extensive revisions, and the reviewers' probing questioning allowed us to better report what we think LAP1 is playing and to better reflect this new data, we altered the title to: '*LAP1 supports nuclear adaptability during constrained migration and invasion*'. We thank you in advance for your consideration of our revised manuscript.

Editor's comments

In our view, for reconsideration at NCB, it would be essential to:

-- extend the mechanistic understanding of LAP1's contribution to NE blebbing, cell migration, and metastasis, with stronger links between these elements. All the reviewers commented on this aspect and we agree that this is an aspect of high interest to cell biologists that is key to the suitability of the work for NCB. The reviewers' comments also echo our own editorial discussions about the study. For instance, Rev#1 suggested **testing the involvement of Torsins**, which we agree would be an interesting and logical next step. The reviewers asked for further **characterizations of the blebs, measurements of nuclear stiffness, NE integrity, and experiments establishing whether LAP1's effects on migration depend on nuclear blebbing in appropriate in vitro models.**

In this revision, we have enhanced mechanistic understanding of LAP1's contribution to NE blebbing, constrained migration and metastasis by depleting orthogonal tethers between the NE and lamina, but strengthening tethers between the NE and lamina by fusing an alternate Lamin binding domain to LAP1B, and employing the suggested mutants known to impair Torsin activation. We propose now that the reason that LAP1 elevation enhances NE blebbing and constrained migration is that expression of the shorter isoform (LAP1C) allows decoupling of the NE from the underlying lamina, and additionally expose a role for Torsin activation in suppressing NE blebbing. We have created stronger links between these elements by performing experiments using appropriate in-vitro (both transwell migration and incorporating new 3D collagen invasion assays) and in-vivo work (using subcutaneous and orthotopic dermal invasion models) to better integrate our findings. We extended our IHC characterisation to extract prognostic value of our observations on disease-free survival in the cohort of human melanoma patients we analysed.

We performed further characterisation of NE stiffness and integrity and as we found no impact of our manipulations, we elected to present this data for reviewers in this rebuttal, but omitted it from the manuscript.

-- enhance the understanding of the contributions of LAP1 and nuclear blebbing to metastasis, in vitro and in vivo

We hope that the above points clarify the contributions of LAP1 and NE blebbing to metastasis, in vitro and in vivo.

-- **All other reviewer concerns** pertaining to strengthening existing data, technical comments, providing controls, methodological details, clarifications and textual changes, should also be addressed.

All other reviewer concerns have been addressed.

-- Finally please pay close attention to our guidelines on statistical and methodological reporting (listed below) as failure to do so may delay the reconsideration of the revised manuscript. In particular please provide:

We have checked our statistical and methodological reporting.

- a Supplementary Figure including **unprocessed images of all gels/blots** in the form of a multi-page pdf file. Please ensure that blots/gels are labelled and the sections presented in the figures are clearly indicated.

We have provided the relevant supplementary figure.

- a Supplementary Table including **all numerical source data in Excel format, with data for different figures provided as different sheets within a single Excel file**. The file should include source data giving rise to graphical representations and statistical descriptions in the paper and for all instances where the figures present representative experiments of multiple independent repeats, the source data of all repeats should be provided.

We have provided the relevant supplementary table.

Reviewers' comments:

Reviewer #1:

Remarks to the Author:

This is an interesting and well conducted study by Jung-Garcia et al exploring a role for LAP1 in modulating nuclear plasticity in a cancer context. Although it is well established that the nucleus undergoes major deformation, blebbing, and rupture, in the context of migration through constrictions

(and much of the early data in the manuscript are largely confirmatory of these studies), there is still much to be understood regarding the underlying molecular mechanisms that control whether cells are more or less able to navigate these constrictions without losses in viability. There are several important advances in this work. The first is the explicit, and well-considered, link to melanoma with the potential of LAP1 levels being a cancer biomarker. The authors take advantage of melanoma cells derived from primary and metastatic lesions as a starting point to their experimental strategy. They demonstrate that the metastatic cells more frequently pass through membranes with narrow pores and then nicely tie this ability to the overexpression of the integral INM protein, LAP1. Unlike many other cancer cell studies that cherry-pick one overexpressed gene of hundreds, there is a clear demonstration that LAP1 is a standout among many genes upregulated and in fact direct staining of primary tumors reveals that the protein is found at high levels and specifically at the migratory edge. The authors also provide interesting genetic and in vivo physical interaction data that suggests a specific isoform of LAP1, LAP1C, which does not interact with the lamina or chromatin is more mobile, populates, and may drive, nuclear blebs and in fact acts in a dominant fashion to promote nuclear migration through constrictions in the non-metastatic cell type. Thus, the paper provides new mechanistic insight into nuclear plasticity in an important physiological (and pathological) context and thus the paper should be of broad appeal and appropriate for NCB pending some needed additions:

1) There was a justified focus on exploring the connections between LAP1 and nuclear factors, however, it is also well established that LAP1 binds to the AAA-ATPase Torsin through its luminal domain where it helps activate the ATPase in ways that have been linked to disrupting chromatin-LAP1 interactions (Luithle et al, 2020) and inducing nuclear envelope blebs (Laudermilch et al., 2016). Further, there is evidence that the LAP1-Torsin interaction contributes to cell migration by modulating LINC complex function (see Saunders et al., JCB, 2017). **Thus, the relationship (or lack thereof) between LAP1 and Torsin in this study should be at least preliminarily investigated to more explicitly connect this work with more established roles of LAP1 in nuclear envelope biology.** The suggestion is to overexpress a LAP1 mutant that cannot bind to Torsin and explore its impact on nuclear envelope blebbing and migration through constrictions.

We thank the reviewer for raising this excellent point as it offered a great opportunity to incorporate some mechanistic data into the manuscript. As the reviewer notes, Torsin deletion/inactivation leads to increased NE blebbing (Laudermilch et al., 2016) and overexpression of LAP1B that is unable to activate Torsins (LAP1B^{R563G}) was unable to suppress NE abnormalities induced by Torsin overexpression (Luithle et al, 2020). Consistent with these data, we found that expressing LAP1B^{R563G}-mRuby3 in primary melanoma WM983A cells enhanced nuclear blebbing, constrained migration and invasion. We interpret these data as LAP1B-mediated Torsin activation does not support NE blebbing. These new data are presented in Figure 6f-k, Figure 7k-m and

Extended Data Figure 8i,j. Importantly, LAP1C^{R441G}-mRuby3 induced similar levels of NE blebbing, and could support similar levels of constrained migration, as LAP1C-mRuby3. LAP1C lacks the strong lamin-binding domain present in the N-terminus of LAP1B, and we suggest that this data suggests that LAP1 can couple the strength of NE/lamina interactions to the activation state of Torsin in the intermembrane space. These new data are presented in Extended Data Figure 7l,m. These data suggest that competition between isoforms may regulate the local activity of Torsin-1A at the nuclear envelope. We extended the reviewer's request to show that the biology observed in vitro was recapitulated in vivo and found that tumours expressing LAP1B^{R563G} are characterised by an invasion advantage similar to LAP1B^{A1-72}.

2) The in vivo interaction data using the mitochondrial anchoring in **Figure 6 is exceedingly difficult to interpret**. First, a **diagram** of LAP1 with the “unique region” clearly labeled would be helpful. Second, there is the interpretation that relocalization of mitochondria to the nuclear envelope is suggestive of physical interactions between LAP1 fragments and lamins/chromatin. But as lamins/chromatin are inside the nucleus, there is a **topological problem** here unless the suggestion is that the mitochondria enter the nucleus? As this is highly unlikely, the interpretation of these experiments as reflecting physiological interactions is questionable. An orthogonal approach **or** a more explicit description/interpretation of how the observed interactions occur would be helpful.

We apologise for the complexity of this figure and have made adjustments to the display and the text to make it more accessible. Specifically, we have renamed the ‘unique NT’ to more clearly reflect which amino acids from LAP1B were added and have incorporated a schematic of the amino terminal fragments of LAP1 isoforms (LAP1B^{NT}, LAP1B¹⁻¹²², LAP1C^{NT}) with the lamin-binding domains indicated. We have also re-written the text describing results of the mitochondrial retargeting assay as follows: “We found that mitochondria displaying HA-LAP1B^{NT} or HA-LAP1B¹⁻¹²², but not HA-LAP1C^{NT} were recruited to the nuclear periphery in cells expressing GFP-Lamin B1 or GFP-Lamin A/C (Fig. 5b,c). Indeed, a pool of mitochondria displaying HA-LAP1B^{NT} or HA-LAP1B¹⁻¹²² colocalised with the lamina and we speculate that interphase rupture or NEBD during M-phase allows nuclear entry of these organelles. We next examined the ability of mitochondria displaying HA-LAP1 N-termini to differentially recruit GFP-Lamins. We found that mitochondria displaying HA-LAP1B^{NT} and HA-LAP1B¹⁻¹²², but not HA-LAP1C^{NT}, could recruit GFP-Lamin B1, but not GFP-Lamin A/C (Fig. 5b,c), suggesting that the unique NT of LAP1B encodes a dominant lamin-binding domain that displays preference for B-type lamins.”

3) There is already published evidence that actomyosin contractility drives NE blebbing and nuclear ruptures. The Hatch and Hetzer (2016) **article** should be referenced and discussed in the context of ROCK inhibition.

We thank the reviewer for highlighting this and made appropriate references to Hatch and Hetzer (2016).

4) Figure 5: A specificity control of **knocking down another integral INM protein** would strengthen this data. e.g. emerin or MAN1.

We thank the reviewer for suggesting this. We have performed the experiment requested by knocking down Emerin in metastatic melanoma WM983B cells. We found that WM983B cells with reduced levels of Emerin show impaired constrained migration and nuclear blebbing in two-round transwell assays using sequentially 8- μ m pores and 5- μ m pores (Extended Data Fig. 7i-k). Whilst this may argue against the specific nature of the phenotype attributed to LAP1, we believe that this illustrates a more general point relating to how the strength of the tethers between NE and lamina controls NE bleb dynamics. Please see also the answer to Reviewer 2, point X for further exploration of this concept. We note that Emerin and LAP1 interact with each other (Shin et al, 2013) and extended our analysis to show that Emerin can localise with LAP1 to NE blebs (Extended Fig. 6d,e). As such, it is possible that the phenocopying between LAP1 and Emerin depletion relates to a functional interdependency. However, given our new findings relating to the role of the Torsin-interacting residues in LAP1's C-terminus in allowing NE bleb formation, constrained migration and invasion, we believe our mechanistic data point to an interplay between Torsin activation and the strength of NE/lamina tethers in NE blebbing.

5) The use of the first and second round filters is useful but it is often challenging to understand whether or not the first round explicitly changes the cells (e.g. deforms the nucleus) and/or selects for a more migratory population for the second round. I don't have any explicit experimental suggestions but think that the **reasoning and interpretation** behind the two rounds of translocation could be better articulated.

Thanks for the comments. We carried out additional experiments and readjusted the text to clarify the rationale behind the multi-round transwell assays.

To understand if the first round of migration primes the cells, we performed two-round transwell assays using sequentially 8- μ m pores and 8- μ m pores. We found that metastatic melanoma

WM983B cells migrate more efficiently and display enhanced nuclear blebbing than primary melanoma WM983A cells (Review Fig. 1), which is consistent with the migratory advantage of WM983B cells over WM983A cells observed passing sequentially 8- μm pores and 5- μm pores (Fig.1d-f and Extended Data Fig.1d-f). We concluded that the first round of migration primes metastatic melanoma cells for subsequent rounds. Please see also question 7 from Reviewer#3 to follow up on the priming mechanism.

To prove if the first round selects for a more migratory cell population, we performed three-round transwell assays using sequentially 8- μm pores, 8- μm pores and 5- μm pores. We found that metastatic melanoma WM983B cells retain a migratory advantage over primary melanoma WM983A cells, but the proportion of cells migrating and displaying nuclear blebs does not increase (Extended Data Fig.1g,h). We concluded that selection of a more migratory cell population does not occur in multi-round transwell assays. Instead, we suggest that metastatic melanoma WM983B cells have an adaptation mechanism to confinement that is absent or cannot get activated to the same extent in primary melanoma WM983A cells.

Review Figure 1. Two-round transwell assays using sequentially 8- μ m pores and 8- μ m pores. (a) Schematic of two-round transwell assays using sequentially 8- μ m pores and 8- μ m pores. (b) Percentage of primary melanoma WM983A cells and metastatic melanoma WM983B cells that translocated their nuclei and displayed nuclear blebs (c) after a second round of transwell migration. (d) Percentage of WM983A and WM983B cells that did not translocate their nuclei and displayed nuclear blebs (e) after a second round of transwell migration. n= 532 and 487, respectively. Experimental data have been pooled from three individual experiments. Graphs show the mean and error bars represent SEM. p values calculated by unpaired t-test; *p<0.05, **p<0.01.

6) Figure 7F: In some of the images it appears as if there is nuclear fluorescence of the Ruby construct raising the possibility that these constructs may be unstable. I don't think this changes the outcome of the experiment but an **explanation for this result is necessary** and could include a western blot to ensure the stability of the constructs.

We have investigated this point, but find no evidence of degradation in the western blot of LAP1C-mRuby3 levels and suspect that the absence of rim-like staining related to the plane of acquisition. After careful consideration and taking into account also the comments from the other reviewers and the need to save space, we decided to omit these data from the manuscript. We include them in this rebuttal for the reviewer's perusal in Review Fig. 2.

Review Figure 2. Effect of LAP1C expression on constrained migration. (a) FACS dot plot of WM983A cells stably expressing GFP-NLS and LAP1C-mRuby3 sorted according to levels of LAP1C-mRuby3 expression. (b) Representative immunoblot for endogenous and exogenous LAP1 expression levels in non-sorted WM983A GFP-NLS LAP1C-mRuby3 cells and WM983A cells stably expressing GFP-NLS and LAP1C-mRuby3 sorted according to levels of LAP1C-mRuby3 expression. (c) Representative pictures of non-sorted WM983A cells and WM983A cells stably expressing GFP-NLS (green) and LAP1C-mRuby3 (red) sorted according to levels of LAP1C-mRuby3 expression and stained for DNA (blue) after a second round of transwell migration through 5- μ m pores. Scale bars, 30 μ m. (d) Percentage of non-sorted WM983A cells and WM983A cells stably expressing GFP-NLS and LAP1C-mRuby3 sorted according to levels of LAP1C-mRuby3 expression that translocated their nuclei and displayed nuclear envelope blebs (e) after a second round of transwell migration. $n = 664, 601, 462, 531$, respectively. Experimental data have been pooled from three individual experiments. Graphs show the mean and error bars represent SEM. p values calculated by one-way ANOVA; $*p < 0.05$.

Reviewer #2:

Remarks to the Author:

In this article, Jung-Garcia and colleagues investigate the difference between two cell lines derived from a melanoma from the same patient, one from the primary tumor, the other from a metastatic lesion. Using a sophisticated transwell assay in which the cells are recovered after a first passage then assayed for a second, they find that the cell line from the metastatic lesion is more effective in passing already at the first round, but even more at the second, and that this was associated with a strong phenotype of formation of blebs on the cells nucleus.

They found that nuclear blebs were present in a large number of nuclei on this cell line even when they had not passed through the transwells. Using published expression databases (on other melanoma cell lines), and further experiments and analysis, they identify and validate three genes upregulated in the more metastatic cell line and find that one of them, encoding for LAP1, is also upregulated in the more metastatic melanoma cell line derived from the patient. They then focus on this protein and find that it seems in general to be more expressed in invasive tumor front and that a higher expression correlates with a poor prognosis. A deeper investigation of the molecular and cell biology of this protein reveals that it has two isoforms with different binding to the nuclear lamina and chromatin and that both are upregulated in the more metastatic melanoma cells. Reduction of the level of expression of these isoforms in the metastatic cell line reduced its transwell migration capacity. The two isoforms showed different binding to Lamin A/C and Lamin B and consistently, different enrichment in the bleb membrane. Investigating the specific role of the two isoforms in promoting transwell migration, they find that the short isoform (which does not bind Lamin B) has a strong, dose-dependent effect on promoting nuclear blebbing and transwell migration when over-expressed, which the long isoform does not have. Over-expression of a shorter version of the long isoform, lacking lamin binding or chromatin binding domains, has the same effect as the short isoform in promoting nuclear blebbing and transwell migration.

It is hard to summarize much more, since the authors do not really provide a synthetic working model for their observations, which is a major concern.

Overall, the article contains a large number of very interesting experiments pointing to a role of LAP1 in modulating the capacity of cells to pass through small pores, with a potential impact on metastasis in melanoma cells. The authors also suggest that this is because this protein facilitates the formation of nuclear blebs. The article is very interesting and the experiments convincing. I would recommend asking the authors for submission of a revised article with the following points clarified:

Major concerns:

The authors **establish many correlative evidences but very few proof of causality**, at various levels of

the article.

The main aspects to investigate in more details are:

a) whether nuclear blebs indeed help the passage of nuclei and thus of cells through small pores. Would there be **another way to promote nuclear blebs, which would for example be independent of an increase in Myosin II** activity, as it is a confounding factor here? For example by weakening the binding of the nuclear membranes to the lamina by depleting some specific proteins? b) **how Lap1 short isoform promotes nuclear blebbing** – and whether it is by promoting nuclear blebbing that it also promotes transwell migration. c) clarifying the function of Lap1/nuclear blebbing in **the metastatic potential of melanoma** (or other) cells – is there increased **collagen matrix invasion** (in an in vitro assay)? Or is it another step of the metastatic spread which is promoted by expression of Lap1 (extravasation, crossing basement membranes, ...?). This might go a bit too far for the scope of the article, but at least a **discussion** of this last point would be important.

We thank the reviewer for raising these points, and agree that understanding how NE blebs facilitate constrained migration is a challenging experimental problem. In this revision, we looked at experimental ways to enhance mechanistic understanding of this process, and have separated our response into the reviewer's four major points:

a) whether nuclear blebs indeed help the passage of nuclei and thus of cells through small pores. Would there be **another way to promote nuclear blebs, which would for example be independent of an increase in Myosin II** activity, as it is a confounding factor here? For example by weakening the binding of the nuclear membranes to the lamina by depleting some specific proteins?

We found that like LAP1, Emerin also localised to NE blebs (Extended Fig. 6d,e). We found that Emerin-depleted WM983B cells show impaired NE blebbing and constrained migration in two-round transwell assays using sequentially 8- μ m pores and 5- μ m pores. We concluded that weakening the binding of the NE to the lamina contributes to the generation of NE blebs and the ability to perform constrained migration. These new results are presented in Extended Data Figure 7i-k and address the reviewer's first point, that promoting NE blebs independently of actomyosin contractility enhances NE blebbing and constrained migration.

b) **how Lap1 short isoform promotes nuclear blebbing** – and whether it is by promoting nuclear blebbing that it also promotes transwell migration.

We extended these results by generating versions of LAP1C that were more strongly tethered to the nuclear lamina. We fused the whole (LBR^{NT}-LAP1C-mRuby3) or part (LBR^{TRS}-LAP1C-mRuby3) of the Lamin B Receptor (LBR) N-terminus to the nucleoplasmic domain of LAP1C to assess the effect of strengthening LAP1-mediated nuclear envelope/lamina tethering in nuclear blebbing, constrained migration and invasion. We found that unlike LAP1C-mRuby3, cells expressing LBR^{NT}-LAP1C-mRuby3 or LBR^{TRS}-LAP1C-mRuby3 in WM983A cells could no longer enhance NE blebbing or migration in two-round transwell assays using sequentially 8- μ m pores and 5- μ m pores and could no longer enhance invasion into collagen I. We concluded that expression of LAP1C supports nuclear blebbing, constrained migration and invasion through allowing the generation of weaker NE/lamina tethers. These new results are presented in (Extended Data Figure 7f-h,n,o) and address the reviewer's second point, by showing that the short isoform of LAP1C can promote blebbing by decoupling the NE from the nuclear lamina.

LAP1 contains an arginine finger (R563) on its carboxy terminus that enables the ER- and NE-luminal AAA-ATPase Torsin-1A to hydrolyse ATP and become active (Brown et al 2014; Sosa et al 2014; Zhan et al, 2013). We found that expressing LAP1B^{R563G}-mRuby3 in primary melanoma WM983A cells enhanced NE blebbing, constrained migration and invasion. We interpret these data as LAP1B-mediated Torsin activation acts to suppress NE blebbing. These new data are presented in Figure 6f-k, Figure 7k-m and Extended Data Figure 8i,j. Importantly, LAP1C^{R441G}-mRuby3 induced similar levels of NE blebbing, and could support similar levels of constrained migration, as LAP1C-mRuby3. LAP1C lacks the strong lamin-binding domain present in the N-terminus of LAP1B, and we suggest that this data suggests that LAP1 can couple the strength of NE/lamina interactions to the activation state of Torsin in the intermembrane space. These new data are presented in Extended Figure 7l,m. These data suggest that competition between isoforms may regulate the local activity of Torsin-1A at the nuclear envelope. We extended the reviewer's request to show that the biology observed in vitro was recapitulated in vivo and found that tumours expressing LAP1B^{R563G} are characterised by an invasion advantage similar to LAP1B ^{Δ 1-72}.

c) clarifying the function of Lap1/nuclear blebbing in **the metastatic potential of melanoma** (or other) cells – is there increased **collagen matrix invasion** (in an in vitro assay)? Or is it another step of the metastatic spread which is promoted by expression of Lap1 (extravasation, crossing basement membranes, ...?). This might go a bit too far for the scope of the article, but at least a **discussion** of this last point would be important.

We found that expressing LAP1C-mRuby3, but not LAP1B-mRuby3, in primary melanoma WM983A cells promoted invasion into 3D collagen I matrices. We concluded that expression of LAP1C supports the ability of melanoma cells to invade. These new data are presented in Figure 6c,d.

We allowed WM983A cells expressing wild-type or mutant versions of LAP1B (LAP1B-mRuby3, LAP1B^{Δ1-72}-mRuby3, LAP1B^{ΔCBR}-mRuby3, LAP1B^{R563G}-mRuby3) or LAP1C (LAP1C-mRuby3, LBR^{NT}-LAP1C-mRuby3, LBR^{TRS}-LAP1C-mRuby3) to invade in 3D collagen I matrices. We found that expression of LAP1C-mRuby3, LAP1B^{Δ1-72}-mRuby3 or LAP1B^{R563G}-mRuby3 enhanced invasion of WM983A cells into collagen I, allowing us to relate the degree of NE blebbing observed to the invasive potential of these melanoma cells. These new data are presented in Figure 6j,k and Extended Data Figure 7n,o.

We extended these *in vitro* invasion assays to the *in vivo* context. We used orthotopic melanoma models where WM983A or WM983B and A375P or A375M2 were injected into the dermis of NSG mice to examine LAP1's contribution to initial local invasion into the dermis as part of the metastatic cascade in melanoma. We found that the metastatic lines invaded more into the dermis than their less or non-metastatic counterparts. Moreover, A375M2 were not only more invasive but also grew much faster *in vivo*, highlighting the aggressiveness of this model (Fig. 7a-f and Extended Data Fig. 8a-d). Supporting our data using subcutaneous melanoma tumours presented in the original submission, we used these orthotopic models, followed by immunohistochemistry and digital pathology methods to show that LAP1 expression was higher at the proximal invasive front (PIF) compared to the tumour body (TB) and higher again at the distal invasive front (DIF) compared to the PIF of these tumours (Fig. 7g-j). Moreover, WM983B and A375M2 tumours presented a higher proportion of cancer cells expressing very high levels of LAP1 compared to their counterparts WM983A and A375P respectively (Fig. 7g-j). Tumours grown in severe combined immunodeficient (SCID) mice after subcutaneous injection retained a similar LAP1 expression pattern (Extended Data Fig. 8e-h). These data provide further evidence that differential LAP1 expression in orthotopic models of melanoma is associated with invasive behaviour.

We next looked to apply our mechanistic understanding of LAP1-dependent NE bleb formation and local invasion *in vivo*. We generated A375P cells bearing versions of LAP1-mRuby3 and examined tumour growth and invasion into the dermis after intradermal injection, and compared to A375M2, our model of aggressive disease. Relative to LAP1B-mRuby3, we found that expression of LAP1C-mRuby3, LAP1B^{Δ1-72}-mRuby3 or LAP1B^{R563G}-mRuby3 in A375P cells all increased local invasion (Fig. 7k,l and Extended Data Fig. 8i,j). Invasion was assessed at endpoint (day 28). Increased invasion was accompanied by increased tumour growth in A375P LAP1B^{Δ1-72}-mRuby3 or

LAP1C-mRuby3 but not in LAP1B^{R563G}-mRuby3 expressing tumours (Fig. 7m), suggesting that in vivo, there is some poorly understood control of proliferation by the LAP1B-Torsin interaction. No differences in proliferation were observed in vitro for any of the cell lines (Extended Data Fig. 8k), suggesting that these cancer cells establish different interactions with the tumour microenvironment for their differential growth in vivo. We concluded that LAP1C supports tumour invasion both in vitro and in vivo.

Lastly, we integrated survival analysis into our observations of LAP1 expression in tissue microarrays from two human melanoma patient cohorts (cohort A including 19 primary tumours and 14 metastases and cohort B with a total of 29 primary tumours and their matched metastases) (Extended Data Tables 13,14). Importantly, higher LAP1 expression in the IF confers shorter disease-free survival (Fig. 8g) indicating that LAP1 levels are linked to worse prognosis. These results suggest that LAP1 could be a prognostic marker in melanoma.

With these important points clarified, the authors might be able to propose a **more precise working model** for how increased expression of Lap1 might promote metastasis, which is totally lacking at the moment. In their discussion they do not even discuss any sort of scenario – for example, is it possible that increased expression of the short Lap1 isoform, because it does not bind as strongly to lamina or chromatin, is weakening the binding of the nuclear inner membrane to the lamina or the chromatin, for example by displacing another protein, and thus makes it easier for the membrane to dissociate from the lamina and form a bleb? Then once a bleb is formed, is the **nucleus easier to deform**, explaining why it passes more easily through the transwell.

We apologise for this oversight. I think it is fair to say that our mechanistic understanding of this process has been significantly enhanced by the experiments suggested in revision. Our model aligns well with the reviewer's hypothesis, and I think is now borne out by experimental data. As shown by depleting orthogonal tethers, identifying the restriction imposed on NE blebbing by the strong lamin-binding domain at the N-terminus of LAP1B and using gain-of-tethering chimaeras, we believe that elevating expression of the short isoform of LAP1(LAP1C) leads to NE/lamina uncoupling and the formation of a bleb. We note that we were unable to detect any proteins that were displaced from blebs by the presence of LAP1C, and whilst we speculate that a salt-and-pepper localisation of differential LAP1 isoforms may selectively weaken NE/lamina interactions in 'hot-spots', we have not been able to provide experimental proof of this. Once a bleb is formed and ruptures, it is possible that this allows a reduction in intranuclear pressure and an increase in deformability allowing transwell passage. However, we should caution that nuclear mechanics are

complex and as this manuscript has not investigated nuclear biophysics (although please see our AFM data in the response to Reviewer 3), we wanted to limit our speculation in this area.

It is also not clear how to explain, from the data shown, how Lap1 increased expression in the more metastatic cell line derived from the patient, promotes the passage through the second transwell assay, while the less metastatic cells have a reduced rate of passage the second time (**why this reduced rate of passage?**).

We apologise that this wasn't clear. In Figure 1, we discovered that these subsequent passages displayed differential requirements for ROCK-mediated actomyosin contractility. ROCK inhibition impeded translocation of the 2nd, but not the 1st, round of transwell migration. The biogenesis of NE blebs is critically dependent upon ROCK1/2 and we suggest that passage through the 1st constraint activates a ROCK-dependent migration programme for subsequent rounds. We believe that elevated LAP1 expression (particularly LAP1C), in the more metastatic cell line renders its nucleus more pliant to the effects of elevated actomyosin contractility resulting in a more deformable and 'blebby' nucleus that we believe licenses migration through constraints.

We have shown in this manuscript that LAP1B does not support NE blebbing. This restriction of NE blebbing required the strong lamin-binding domain, pointing to the NE/lamina tethering described above, but it also required the ability of LAP1B to activate Torsin through R563. Given Torsin's ability to control the LINC complex (Laudermilch et al, 2016), an alternate possibility is that the impaired migration ability of melanoma cells from the primary tumour relates to differential control of the LINC complex and altered balance of nucleo-cytoskeletal forces. We look forward to unpicking this regulation in future manuscripts.

There is a real **need to provide more mechanistic explanations** for the phenomena described in the article, in order to get to a working model that explains the observations. This would also strengthen the proof of the causal relationships suggested by the authors.

We hope that the additional experiments referred to in this rebuttal have provided more mechanistic explanations for the phenomena we described, and that the reviewer is satisfied with the working model (Figure 8h) that we described above.

A minor concern, which might also help clarify some aspects: when the authors describe nuclear blebbing, they always seem to observe chromatin herniation inside the bleb. In the literature, it seems that there are at least **two types of nuclear blebs**: blebs that are just a separation of the nuclear membranes from the lamina, but the lamina is intact and there is no chromatin in these blebs, just nucleoplasm, and blebs that combine detachment of the membrane from the lamina and a rupture in the lamina, which allows chromatin to flow into the bleb. Depending on the type of effect that Lap1 short isoform has on perturbing the nuclear envelope, one or the other type of bleb might be more favored. The blebs without chromatin inside should be easily visualized using a combination of NLS-GFP and a chromatin marker. They should be positive for GFP but not for the chromatin marker.

Thanks for noting this. Understanding NE bleb dynamics is complex. We characterised nuclear envelope bleb composition by immunofluorescence in primary melanoma WM983A cells and found that nuclear envelope blebs contain nucleoplasm, chromatin and Emerin but are deficient in nuclear pore complexes (Extended Data Fig. 6d,e). This might suggest that the chromatin herniates and pushes against/extrudes the lamina and membrane. However, bleb formation is a dynamic process, and our initial imaging suggests that the NE bleb occurs first, with lamina and chromatin following. We will require microscopy with spatial and temporal resolution beyond what is currently achievable to investigate this further.

Reviewer #3:

Remarks to the Author:

In this manuscript, Jung-Garcia et al. determined that LAP1 is overexpressed in metastatic melanoma cells. Importantly, the authors investigated the role of two LAP1 isoforms (LAP1B and LAP1C) in melanoma cell migration through constricted pores. They determined that ectopic expression of LAP1C in WM983A melanoma cells derived from a primary tumor increases both nuclear envelope (NE) blebbing and migration efficiency. Intriguingly, ectopic expression of LAP1B lacking either the dominant lamin-binding or the CBR, but not full-length LAP1B, in WM983A melanoma cells also enhanced both NE blebbing and migration. Overall, these are interesting and novel observations. Unfortunately, several concerns and limitations have dampened the reviewer's enthusiasm for this manuscript as outlined below.

Concerns:

1. This is a rather phenomenological study as a **mechanism of action for LAP1B and LAP1C isoforms is missing**. For instance, (A) it is **unclear how LAP1C overexpression contributes to increased NE blebbing and transwell migration**. (B) **Does LAP1C co-localization with lamin A/C affect the integrity** of the nuclear envelope? (C) **Is the envelope more pliable with increased expression?** (D) What happens when

the interaction between LAP1C and lamin A/C is disrupted? (E) How do these results relate to the effect of actomyosin contractility described in Figure 1?

We thank the reviewer for the assessment of our manuscript and agree that addressing these concerns would strengthen our manuscript. Here follows our response to the major points A-D.

(A) It is unclear how LAP1C overexpression contributes to increased NE blebbing and transwell migration

The mechanism by which LAP1C contributes to NE blebbing and transwell migration was raised by all reviewers. As shown by depleting orthogonal tethers (Emerin), identifying the restriction imposed on NE blebbing by the strong lamin-binding domain at the N-terminus of LAP1B and using gain-of-tethering chimaeras, we believe that elevating expression of the short isoform of LAP1 leads to NE/lamina uncoupling and the formation of a NE bleb. To avoid restating the same information again, we respectfully refer the reviewer to our answers to Reviewer 2, Points A-C in which the details of this mechanism have been discussed.

(B) Does LAP1C co-localization with lamin A/C affect the integrity of the nuclear envelope?

In our original submission, we analysed nuclear envelope repair kinetics and nuclear envelope rupture-repair frequency in primary melanoma WM983A and metastatic melanoma (WM983B) cells stably expressing GFP-NLS. We looked to extend these data here and transduced WM983A cells with vectors encoding both LAP1 isoforms (LAP1-mRuby3), LAP1B-mRuby3 or LAP1C-mRuby3. We found that expressing LAP1-mRuby3, LAP1B-mRuby3 or LAP1C-mRuby3 did not influence the average time for nuclear envelope repair (Review Fig. 3a). Interestingly, expressing LAP1B-mRuby3 induced a higher rate of nuclear envelope rupture events (Review Fig. 3b). We concluded that neither LAP1B nor LAP1C contribute towards nuclear envelope repair but LAP1B might enhance nuclear envelope fragility, perhaps through its ability to communicate with Torsin and the LINC complex. Whilst these data are interesting, in the interests of space, we decided not to include them in the resubmitted manuscript and provide them here for review. A future manuscript will investigate the connections between LAP1, Torsin and the LINC complex and we would like to build on these findings here.

Review Figure 3. Effect of LAP1 expression on nuclear envelope rupture and repair. (a) Duration of nuclear envelope repair in primary melanoma WM983A cells and WM983A expressing both LAP1 isoforms (LAP1), LAP1B or LAP1C and nuclear envelope rupture-repair events per hour (b) over the course of 15 hours. $n = 486, 575, 656$ and 651 , respectively. Experimental data have been pooled from three individual experiments. Horizontal lines show the median and whiskers show minimum and maximum range of values. p values calculated by one-way ANOVA. $*p < 0.05$.

(C) Is the envelope more pliable with increased expression?

We thank the reviewer for suggesting these interesting experiments. We used atomic force microscopy in primary melanoma WM983A cells stably expressing GFP-NLS and both LAP1 isoforms (LAP1-mRuby3), LAP1B-mRuby3 or LAP1C-mRuby3, and metastatic melanoma WM983B cells stably expressing GFP-NLS. In these assays, the atomic force microscopy probe (round) targets a supranuclear region of the cell and the measured stiffness comprises that of the plasma membrane, the actin cortex, the cytosol, and the nucleus and offers a whole-cell measurement of nuclear stiffness. We found that WM983A GFP-NLS, WM983A GFP-NLS LAP1-mRuby3, LAP1B-mRuby3 or LAP1C-mRuby3 showed reduced whole-cell nuclear stiffness compared with WM983B GFP-NLS but no differences between individual LAP1 isoforms was observed (Review Fig. 4). We concluded that LAP1 does not influence whole-cell nuclear stiffness of primary melanoma cells. We considered including these data in the manuscript, but in the interests of space, elected to present them only in this rebuttal.

Review Figure 4. Effect of LAP1 expression on whole-cell nuclear stiffness. Whole-cell nuclear stiffness of primary melanoma WM983A cells stably expressing GFP-NLS and both LAP1 isoforms (LAP1-mRuby3), LAP1B-mRuby3 or LAP1C-mRuby3 and metastatic melanoma WM983B cells stably expressing GFP-NLS. n= 60 in all conditions. Experimental data have been pooled from three individual experiments. Graph shows the mean and error bars represent SD. p values calculated by one-way ANOVA. *p<0.05, **p<0.01, ***p<0.001.

(D) What happens when the interaction between LAP1C and lamin A/C is disrupted?

We generated a version of LAP1 lacking the 1-72 and 184-337 lamin-binding regions (LAP1^{ALB}-mRuby3) to assess the effect of completely disrupting the LAP1-lamina interaction. LAP1^{ALB}-mRuby3 expressed in primary melanoma WM983A cells, was shifted to the ER, but it did not influence migration or nuclear blebbing in two-round transwell assays using sequentially 8- μ m pores and 5- μ m pores (Review Fig. 5). Interpretation of this mutant is complex, as the relocalisation to the ER likely disturbs many aspects of LAP1-dependent nuclear biology. We found that in these cells, NE blebbing and transwell migration occurred at similar rates to that observed when LAP1C was expressed, but a mechanistic explanation for this requires too many assumptions about the underlying biology and we decided not to include this in the manuscript, but to present the data for the reviewer in this rebuttal.

Review Figure 5. Effect of disrupting LAP1-lamina interaction. (a) Representative pictures of primary melanoma WM983A cells expressing LAP1-mRuby3 or LAP1 Δ LB-mRuby3. Scale bars, 30 μ m. (b) Percentage of WM983A cells expressing LAP1-mRuby3 or LAP1 Δ LB-mRuby3 that translocated their nuclei (c) and displayed nuclear envelope blebs (c) after a second round of transwell migration. n = 744 and 732, respectively. Experimental data have been pooled from three individual experiments. Graph shows the mean and error bars represent SEM. p values calculated by unpaired t-test. n.s.: not significant.

2. To prove that LAP1 isoforms confer nuclear plasticity, **physical measurements (e.g., nuclear stiffness) are needed.**

We thank the reviewer for this point and realise that we were a little too blasé with our phraseology. As described above, we measured whole cell nuclear stiffness by atomic force microscopy at supranuclear regions in primary melanoma WM983A cells stably expressing GFP-

NLS, WM983A stably expressing GFP-NLS and both LAP1 isoforms (LAP1-mRuby3), LAP1B-mRuby3 or LAP1C-mRuby3, and metastatic melanoma WM983B cells stably expressing GFP-NLS. We found that whilst WM983A GFP-NLS, WM983A GFP-NLS LAP1-mRuby3, LAP1B-mRuby3 or LAP1C-mRuby3 showed reduced whole cell nuclear stiffness compared with WM983B GFP-NLS (Review Fig.3), there were no differences between these lines. We concluded that LAP1 does not influence whole cell nuclear stiffness of primary melanoma cells. Given the absence of phenotype of LAP1 on nuclear stiffness, we have been careful not to overinterpret our data and have veered away from making assumptions of the underlying biophysics. In recognition that we did not present data regarding nuclear mechanobiology, we also updated the manuscript title, from “LAP1 regulates nuclear plasticity to enable constrained migration” to “LAP1 supports nuclear adaptability during constrained migration and invasion” to better reflect these and all our new results.

3. Extending the aforementioned interesting observations from the **in vitro to the in vivo** setting would greatly enhance the impact of this manuscript. Importantly, these studies would help differentiate if LAP1 is an important player in metastasis or a biomarker of metastatic cells.

This was a great idea and we thank the reviewer for the suggestions which we believe have really strengthened our manuscript, allowing us to integrate the mechanistic information gleaned in vitro on LAP1-dependent nuclear envelope remodelling to physiologically relevant in vivo models of melanoma progression.

We used orthotopic melanoma models where WM983A or WM983B and A375P or A375M2 were injected into the dermis of NSG mice to examine LAP1's contribution to initial local invasion into the dermis as part of the metastatic cascade in melanoma. We found that the metastatic lines invaded more into the dermis than their less or non-metastatic counterparts. Moreover, A375M2 were not only more invasive but also grew much faster in vivo, highlighting the aggressiveness of this model (Fig. 7a-f and Extended Data Fig. 8a-d). Supporting our subcutaneous tumour data presented in the original submission we used immunohistochemistry and digital pathology methods to show that LAP1 expression was higher at the proximal invasive front (PIF) compared to the tumour body (TB) and higher again at the distal invasive front (DIF) compared to the PIF of these tumours (Fig. 7g-j). Moreover, WM983B and A375M2 tumours presented a higher proportion of cancer cells expressing very high levels of LAP1 compared to their counterparts WM983A and A375P respectively (Fig. 7h,j). Tumours grown in severe combined immunodeficient (SCID) mice after subcutaneous injection retained a similar LAP1 expression pattern (Extended Data Fig. 8e-h), but due to the nature of the injections, a distal invasive front was not possible to obtain. These new data provide further evidence of how LAP1 expression correlates with invasive behaviour in orthotopic models of melanoma.

We next looked to apply our mechanistic understanding of LAP1-dependent NE bleb formation to invasion *in vivo*. We generated A375P cells bearing versions of LAP1-mRuby3 and examined local invasion into the dermis after intradermal injection, and compared to A375M2, our model of aggressive disease. Relative to LAP1B-mRuby3, we found that expression of LAP1C-mRuby3, LAP1B^{Δ1-72}-mRuby3 or LAP1B^{R563G}-mRuby3 in A375P cells all increased local invasion (Fig. 7k,l and Extended Data Fig. 8i,j). Invasion was assessed at endpoint (day 28). Increased invasion was accompanied by increased tumour growth in A375P LAP1B^{Δ1-72}-mRuby3 or LAP1C-mRuby3 but not in LAP1B^{R563G}-mRuby3 expressing tumours (Fig. 7m), suggesting that *in vivo*, there is some poorly understood control of proliferation by the LAP1B-Torsin interaction. No differences in proliferation were observed *in vitro* (Extended Data Fig. 8k), suggesting that these cancer cells establish different interactions with the tumour microenvironment for their differential growth *in vivo*. We concluded that LAP1C supports tumour invasion both *in vitro* and *in vivo*.

Lastly, we integrated survival analysis into our observations of LAP1 expression in tissue microarrays from two human melanoma patient cohorts (cohort A including 19 primary tumours and 14 metastases and cohort B with a total of 29 primary tumours and their matched metastases) (Extended Data Tables 13,14). Importantly, higher LAP1 expression in the IF confers shorter disease-free survival (Fig. 8g) indicating that LAP1 levels are linked to worse prognosis. These results suggest that LAP1 could be a prognostic marker in melanoma.

4. The authors should employ alternative *in vitro* models such as 3D collagen gels.

We thank the reviewer for this suggestion and found that the *in vitro* invasion assays performed in 3D-collagen gels were a nice bridge between our microscopy, transwell migration assays and *in vivo* work.

We found that expressing LAP1C-mRuby3, but not LAP1B-mRuby3, in primary melanoma WM983A cells promotes invasion into collagen I (Fig. 6c, d). We allowed WM983A cells expressing wild-type or mutant versions of LAP1B (LAP1B-mRuby3, LAP1B^{Δ1-72}-mRuby3, LAP1B^{ΔCBR}-mRuby3, LAP1B^{R563G}-mRuby3) or LAP1C (LAP1C-mRuby3, LBR^{NT}-LAP1C-mRuby3, LBR^{TRS}-LAP1C-mRuby3) to invade in 3D collagen I matrices. We found that expression of LAP1C-mRuby3, LAP1B^{Δ1-72}-mRuby3 or LAP1B^{R563G}-mRuby3 enhanced invasion of WM983A cells into collagen I (Fig. 6j,k and Extended Data Fig. 7n,o). These data paralleled well the transwell migration data, and similarly paralleled the *in vivo* data described in point 4.

5. LAP1C overexpression enhances the transwell migration of WM983A cells. Is the same true for A375P, other melanoma cells, and melanocytes? How do LAP1B mutants affect the transwell migration of these additional cell lines? In extension, does the localization of LAP1B change in response to $\Delta 1-72$ and ΔCBR mutations? How is 3D cell migration through confining pores affected in response to these mutants or LAP1C overexpression?

The reviewer essentially requested that we repeat almost all of our in vitro findings with an alternate cell line pair. Although this was a lot of work, the benefit of this was three-fold. Firstly, it allowed us to validate our findings in an additional system. Secondly, it allowed us to better link the transcriptomics (performed originally in the A375 pair and validated in the WM983 pair) to our functional studies. Lastly, the A375 model turned out to be a robust model of in vivo growth and dermal invasion, which we described in point 4 and used in in vivo assays. We should note that we did not perform migration assays with melanocytes because they grow much more slowly, are hard to transfect/transduce and do not migrate or invade compared with melanoma cells using serum as chemoattractant (eg, Fig 2g, h from Orgaz et al, 2009). To address Reviewer 3's concerns, we generated A375P cells stably expressing GFP-NLS and both LAP1 isoforms (LAP1-mRuby3), LAP1B-mRuby3 or LAP1C-mRuby3 and challenged them to two-round transwells assays using sequentially 8- μm pores and 5- μm pores. Just as in WM983A cells, we found that expressing LAP1-mRuby3 or LAP1C-mRuby3 but not LAP1B-mRuby3 enhanced nuclear blebbing and migration of A375P cells (Extended Data Fig. 7b,c). We concluded that expression of LAP1C-mRuby3 in different melanoma cell lines (WM983A and A375P) enhances nuclear blebbing and constrained migration.

We next generated A375P cells stably expressing GFP-NLS and LAP1B $\Delta 1-72$ -mRuby3, LAP1B ΔCBR -mRuby3 or LAP1B R^{563G} -mRuby3 and challenged them to two-round transwells assays using sequentially 8- μm pores and 5- μm pores. We found that expressing LAP1B $\Delta 1-72$ -mRuby3 or LAP1B R^{563G} -mRuby3 but not of LAP1B ΔCBR -mRuby3 enhanced nuclear blebbing and migration of A375P cells (Fig. 6h,i). We concluded that expression of LAP1B $\Delta 1-72$ -mRuby3 or LAP1B R^{563G} -mRuby3 in different melanoma cell lines (WM983A and A375P) enhances nuclear blebbing and constrained migration.

The reviewer also requested that we assessed the effect of LAP1B mutants and LAP1C overexpression in 3D collagen I matrices. We allowed WM983A cells expressing wild-type or mutant versions of LAP1B (LAP1B-mRuby3, LAP1B $\Delta 1-72$ -mRuby3, LAP1B ΔCBR -mRuby3, LAP1B R^{563G} -mRuby3) or LAP1C (LAP1C-mRuby3, LBR NT -LAP1C-mRuby3, LBR TR5 -LAP1C-mRuby3) to invade in 3D collagen I matrices. We found that expression of LAP1C-mRuby3, LAP1B $\Delta 1-72$ -mRuby3 or

LAP1B^{RS63G}-mRuby3 enhanced invasion of WM983A cells into collagen I (Fig. 6c,d,j,k and Extended Data Fig. 7n,o. We thank the reviewer once again for this suggestion that provided a nice bridge between our microscopy, transwell migration assays and in vivo assays.

6. Some of the authors remarks are not supported by their data. For instance:

a. Lines 98-99: “metastatic melanoma WM983B cells were more effective at negotiating constraints than primary melanoma WM983A cells (Fig.1b). According to Fig. 1b, this is correct only for the pore size of 8 μm . There is no ss for the pore size of 5 μm and no difference for 3 μm . Along these lines, their next statement is also not supported by their data (Fig. 1c).

We thank the reviewer for the careful reading and assessment of our manuscript. We repeated this experiment, reanalysed and replotted the data and rephrased the two statements as follows: “We found that during the first round of migration, whilst decreasing pore size impaired migration, WM983B cells were more effective at negotiating 8- μm and 5- μm pores than WM983A cells (Fig. 1b and Extended Data Fig. 1a) and up to 10% nuclei displayed at least one NE bleb before and after pore transit (Fig. 1c and Extended Data Fig. 1b,c).”

b. Lines: 343-344: “the effect of LAP1C promoting NE blebbing and constrained migration was concentration dependent (Fig.7c-g)”. There is no difference between medium and high levels.

We thank the reviewer for the careful reading and assessment of our manuscript. After consideration about the relative merits of this figure, we decided to omit these data from the manuscript and to move these data to Review Fig. 2.

7. Lines 130-135: “ROCK1/2 inhibition did not reduce nuclear translocation but did reduce NE blebbing of WM983B cells during the first round of migration (Fig.1g, h). However, ROCK1/2 inhibition markedly impaired nuclear translocation and reduced NE blebbing after pore transit during the second round (Fig.1i, j), suggesting that passage through the first constraint activates a Rho-ROCK1/2-dependent migration programme for subsequent passages”. The authors’ statement is not necessarily correct. According to the authors, the second passage occurred through pores of 5 μm , whereas the first through 8 μm . **The reviewer believes that 8 μm pores do not constitute a confining microenvironment, and that is why ROCK1/2 inhibition has little or no effect.** The authors’ statement will be supported only if they carry out both the first and second passages through 8 μm .

We treated metastatic melanoma WM983B cells with ROCK1/2 inhibitor (ROCKi) GSK269962A and challenged them to two-round transwell assays using sequentially 8- μ m and 8- μ m pores. We found that migration and nuclear blebbing of WM983B cells were reduced upon ROCKi treatment after a second round (Extended Data Fig. 2f,g). The statement highlighted by the reviewer was rephrased as follows: "We confirmed that MLC2 activity was reduced after ROCK1/2 inhibition (Extended Data Fig. 2d,e) and found that ROCK1/2 inhibition did not reduce nuclear translocation (Fig. 1g) but did reduce NE blebbing (Fig. 1h) of WM983B cells during the first round of migration through 8- μ m pores. However, ROCK1/2 inhibition markedly impaired nuclear translocation and reduced NE blebbing during the second round through 8- μ m (Extended Data Fig. 2f,g) and 5- μ m pores (Fig. 1i,j). We suggest that passage through the first constraint activates a Rho-ROCK1/2-dependent migration programme for subsequent rounds." We hope that this rewording is satisfactory.

8. The authors should cite a relevant article with PMID: 31690619

We made appropriate references to Mistriotis et al (2019) article in the introduction and discussion.

9. There are several typos (e.g., the y-axis of Supplementary Fig. 1a,b, d, e,h j).

We revised Extended Data Fig. 1 and have checked the manuscript again for typographical errors.

We reiterate our thanks for your consideration of these points,

Best wishes,

Jez Carlton & Victoria Sanz-Moreno

Decision Letter, first revision:

Our ref: NCB-C46392A

24th August 2022

Dear Dr. Carlton,

Thank you for submitting your revised manuscript "LAP1 supports nuclear adaptability during constrained migration and invasion" (NCB-C46392A) and thank you very much for your patience with the re-review process. The revision has now been seen by the original referees and their comments are below. The reviewers find that the paper has improved in revision, and therefore we'll be happy in principle to publish it in Nature Cell Biology, pending minor revisions to satisfy Rev#3's final comment (see below) and to comply with our editorial and formatting guidelines. We welcome the addition of the data requested by Rev#3 if you have them but will leave it to you to decide whether to provide them or not, as we do not find them strictly essential to support the core conclusions.

We are now performing detailed checks on your paper and will send you a checklist detailing our editorial and formatting requirements in about 1-2 weeks. Please do not upload the final materials and make any revisions until you receive this additional information from us.

Thank you again for your interest in Nature Cell Biology. Please do not hesitate to contact me if you have any questions.

Sincerely,

Melina

Melina Casadio, PhD
Senior Editor, Nature Cell Biology
ORCID ID: <https://orcid.org/0000-0003-2389-2243>

Reviewer #1 (Remarks to the Author):

I was quite enthusiastic about this manuscript and the authors did an excellent job addressing my, and the other reviewer's, concerns. The additional data and discussion have really strengthened the manuscript. I'm very supportive of publication.

Reviewer #2 (Remarks to the Author):

In their revised manuscript and their point by point answer, Jung-Garcia and colleague avec addressed all the concerns I had - they made a very substantial effort on the experimental side, with a lot of novel and insightfull results. They also provided more mechanistic explanations and proposed a working model to summarize their findings. This clarifies the major concern I had and provides some basis to go beyond purely correlative evidences, which will be important for futur studies. The article is very interesting as it proposes conceptual advances for the role of nuclear blebs in the context of migration of metastatic cells. It should appeal to a large readership. I recommend accepting it for publication in NCB.

Reviewer #3 (Remarks to the Author):

The revised manuscript is markedly improved and worthy of publication in Nature Cell Biology provided that the authors address one remaining point.

Lines 124-126: "We suggest that passage through the first constraint activates a Rho-ROCK1/2-dependent migration programme for subsequent rounds".

This reviewer is unclear why this program is critical during the second translocation through 8 μm pores when it appears to be dispensable in the first round? Could the authors show that RhoA or pMLC2 activity is higher in the 2nd than the 1st round?

Minor points:

1. Typos are still present in the y-axes of Extended Data Fig. 1.
2. Figure 5d-f: t1/2 and mobile fraction quantification could be presented for these data.

Decision Letter, final checks:

Our ref: NCB-C46392A

6th September 2022

Dear Dr. Carlton,

Thank you for your patience as we've prepared the guidelines for final submission of your Nature Cell Biology manuscript, "LAP1 supports nuclear adaptability during constrained migration and invasion" (NCB-C46392A). Please carefully follow the step-by-step instructions provided in the attached file, and add a response in each row of the table to indicate the changes that you have made. Please also check and comment on any additional marked-up edits we have proposed within the text. Ensuring that each point is addressed will help to ensure that your revised manuscript can be swiftly handed over to our production team.

We would like to start working on your revised paper, with all of the requested files and forms, as soon as possible (preferably within one week). Please get in contact with us if you anticipate delays.

In recognition of the time and expertise our reviewers provide to Nature Cell Biology's editorial process, we would like to formally acknowledge their contribution to the external peer review of your manuscript entitled "LAP1 supports nuclear adaptability during constrained migration and invasion". For those reviewers who give their assent, we will be publishing their names alongside the published article.

Nature Cell Biology offers a Transparent Peer Review option for new original research manuscripts submitted after December 1st, 2019. As part of this initiative, we encourage our authors to support increased transparency into the peer review process by agreeing to have the reviewer comments, author rebuttal letters, and editorial decision letters published as a Supplementary item. When you submit your final files please clearly state in your cover letter whether or not you would like to participate in this initiative. Please note that failure to state your preference will result in delays in accepting your manuscript for publication.

Cover suggestions

As you prepare your final files we encourage you to consider whether you have any images or illustrations that may be appropriate for use on the cover of Nature Cell Biology.

Nature Cell Biology has now transitioned to a unified Rights Collection system which will allow our Author Services team to quickly and easily collect the rights and permissions required to publish your work. Approximately 10 days after your paper is formally accepted, you will receive an email in providing you with a link to complete the grant of rights. If your paper is eligible for Open Access, our Author Services team will also be in touch regarding any additional information that may be required to arrange payment for your article.

Please note that *Nature Cell Biology* is a Transformative Journal (TJ). Authors may publish their research with us through the traditional subscription access route or make their paper immediately open access through payment of an article-processing charge (APC). Authors will not be required to make a final decision about access to their article until it has been accepted. Find out more about Transformative Journals

Please use the following link for uploading these materials:
[Redacted]

Best regards,

Nyx Hills
Staff
Nature Cell Biology

On behalf of

Melina Casadio, PhD
Senior Editor, Nature Cell Biology
ORCID ID: <https://orcid.org/0000-0003-2389-2243>

Reviewer #1:

Remarks to the Author:

I was quite enthusiastic about this manuscript and the authors did an excellent job addressing my, and the other reviewer's, concerns. The additional data and discussion have really strengthened the

manuscript. I'm very supportive of publication.

Reviewer #2:

Remarks to the Author:

In their revised manuscript and their point by point answer, Jung-Garcia and colleague avec addressed all the concerns I had - they made a very substantial effort on the experimental side, with a lot of novel and insightfull results. They also provided more mechanistic explanations and proposed a working model to summarize their findings. THis clarifies the major concern I had and provides some basis to go beyond purely correlative evidences, which will be important for futur studies. The article is very interesting as it proposes conceptual advances for the role of nuclear blebs in the context of migration of metastatic cells. It should appeal to a large readership. I recommend accepting it for publication in NCB.

Reviewer #3:

Remarks to the Author:

The revised manuscript is markedly improved and worthy of publication in Nature Cell Biology provided that the authors address one remaining point.

Lines 124-126: "We suggest that passage through the first constraint activates a Rho-ROCK1/2-dependent migration programme for subsequent rounds".

This reviewer is unclear why this program is critical during the second translocation through 8 μm pores when it appears to be dispensable in the first round? Could the authors show that RhoA or pMLC2 activity is higher in the 2nd than the 1st round?

Minor points:

1. Typos are still present in the y-axes of Extended Data Fig. 1.
2. Figure 5d-f: $t_{1/2}$ and mobile fraction quantification could be presented for these data.

Author Rebuttal, first revision:

In response to reviewers' comments:

Reviewer 1 and Reviewer 2 raised no further concerns.

Reviewer 3 noted: *'The revised manuscript is markedly improved and worthy of publication in Nature Cell Biology provided that the authors address one remaining point. Lines 124-126: "We suggest that*

passage through the first constraint activates a Rho-ROCK1/2-dependent migration programme for subsequent rounds”.

This reviewer is unclear why this program is critical during the second translocation through 8 μm pores when it appears to be dispensable in the first round? Could the authors show that RhoA or pMLC2 activity is higher in the 2nd than the 1st round?’

Minor points:

- 1. Typos are still present in the y-axes of Extended Data Fig. 1.*
- 2. Figure 5d-f: $t_{1/2}$ and mobile fraction quantification could be presented for these data.*

Biochemical analysis of cells recovered from multiple passes through transwells is challenging, however we wanted to address this point as we felt it had the potential to make a strong mechanistic contribution to our understanding of how passage through constraints could reprogramme a cell’s subsequent migratory ability. We also note that Reviewer 3 suggested that an 8 μm pore size was an insufficient challenge to restrict the cell’s nucleus. Whilst we agree with this, we point out that passage of this constraint will still require substantial cytoskeletal rearrangement. We compared levels of pMLC2 in primary WM983A and metastatic WM983B cells grown in 2D or recovered after one round of passage through 8 μm pores – our 1st challenge. Whilst the metastatic WM983B cells display higher levels of actomyosin contractility than the primary WM983A cells, after pore transit, levels of phosphorylated MLC2 in WM983B cells were enhanced further, whereas levels of pMLC2 in WM983A cells did not change. Given that the second round of transwell migration and the formation of NE blebs is critically dependent upon pMLC and actomyosin contractility, we feel that these data expose an important difference between primary and metastatic melanoma cells, in that the act of squeezing through a gap elevates actomyosin contractility in metastatic, but not primary melanoma cells. We think that this may explain how physical constraints can prime metastatic cells to reprogram their migratory mode to a more contractile state that can facilitate subsequent nuclear remodelling. We thank the reviewer for suggesting experiments along these lines and have incorporated these data into Figure 1 as Fig. 1k.

Reviewer 3 also noted typographical errors in the Y-axis of ED1A, ED1B, ED1D & ED1E (‘Unstranlocated’). These have been corrected to ‘Untranslocated’ in new ED1A, ED1B, ED1E & ED1F.

Reviewer 3 suggested providing $T_{1/2}$ and mobile fraction quantification for the LAP1 mobility data in Figure 5E. To calculate these parameters, the FRAP recovery curves need to plateau. Unfortunately, recovery of LAP1B-GFP is so slow that we don’t reach plateau in the timeframe of the experiment. We

have used EasyFRAP (<https://easyfrap.vynet.upatras.gr/>) to attempt curve fitting, as described for LAP1-GFP FRAP data in PMID:33320087. We are able to report mobile fractions of 0.64 ± 0.23 and 0.71 ± 0.28 for LAP1C-mRuby in the main nucleus and bleb respectively and $T_{1/2}$ of 63 ± 59 seconds and 56 ± 51 seconds for LAP1C-mRuby in the main nucleus and bleb respectively. We were unable to fit curves for LAP1B-mRuby3 datasets as the recovery was so slow. You will note from the size of the error that the data we could obtain for LAP1C-mRuby3 are very noisy, and that is because the averaged values for $T_{1/2}$ and mobile fraction are calculated from extrapolations of the recovery data beyond the timeframe of recording. We note that Luthile et al., in PMID:33320087 could only provide $T_{1/2}$ and mobile fraction data for LAP1-GFP in Lamin-depleted cells, in which INM protein mobility is greatly increased. Rather than incorporate this extrapolated data into the manuscript, we would like to retain our qualitative interpretation of these data, suggesting only that LAP1B is less mobile in the INM than LAP1C.

Final Decision Letter:

Dear Dr Carlton,

I am pleased to inform you that your manuscript, "LAP1 supports nuclear adaptability during constrained melanoma cell migration and invasion", has now been accepted for publication in Nature Cell Biology. Congratulations on this exciting study!

Please note that *Nature Cell Biology* is a Transformative Journal (TJ). Authors may publish their research with us through the traditional subscription access route or make their paper immediately open access through payment of an article-processing charge (APC). Authors will not be required to make a final decision about access to their article until it has been accepted. Find out more about Transformative Journals

If you have not already done so, we strongly recommend that you upload the step-by-step protocols used in this manuscript to the Protocol Exchange (www.nature.com/protocolexchange), an open online resource established by Nature Protocols that allows researchers to share their detailed experimental know-how. All uploaded protocols are made freely available, assigned DOIs for ease of citation and are fully searchable through nature.com. Protocols and Nature Portfolio journal papers in which they are used can be linked to one another, and this link is clearly and prominently visible in the online versions of both papers. Authors who performed the specific experiments can act as primary authors for the Protocol as they will be best placed to share the methodology details, but the Corresponding Author of the present research paper should be included as one of the authors. By uploading your Protocols to Protocol Exchange, you are enabling researchers to more readily reproduce or adapt the methodology you use, as well as increasing the visibility of your protocols and papers. You can also establish a dedicated page to collect your lab Protocols. Further information can be found at www.nature.com/protocolexchange/about

With kind regards,

Melina

Melina Casadio, PhD
Senior Editor, Nature Cell Biology
ORCID ID: <https://orcid.org/0000-0003-2389-2243>